# Foreshock-induced slip transients set mainshock nucleation timing

Barnaby Fryer[1✉], Dmitry Garagash[2], Mathias Lebihain[3] & François Passelègue[1]

Foreshocks are sometimes observed before earthquakes[1–13], yet their role in controlling rupture nucleation remains unclear[1,11,14]. Classical models often assume that nucleation arises from slow, quasi-static slip governed primarily by fault weakening[15–21], typically neglecting impulsive precursory events. Here we show, using laboratory experiments and a rate-and-state-based Griffith-like rupture framework[22], that foreshocks, when they occur at the onset of or during nucleation, can fundamentally regulate earthquake initiation. We find that the slip burst induced by foreshocks imparts a transient sliding velocity, $V_{min}$, whose magnitude is set by foreshock size and which robustly predicts both nucleation duration and spatial length. Larger foreshocks generate higher $V_{min}$ and trigger a more rapid transition to dynamic rupture, whereas smaller foreshocks produce long-duration quasi-static growth and very small impulses lead to ruptures entirely arresting. Extending our theoretical framework to tectonic faults, we show that foreshock and associated slow-slip sequences preceding natural earthquakes seem to follow the same scaling. These observations allow us to constrain realistic characteristic nucleation slip distances of 0.3–3.0 mm, orders of magnitude smaller than those inferred for dynamic rupture[23]. Our results demonstrate that foreshock-induced transients set the timing and potential detectability of earthquake nucleation[24].

Understanding the nucleation of frictional ruptures is a central challenge in earthquake mechanics[18]. Beyond earthquake physics, nucleation processes are generic to frictional instabilities. Nucleation refers to the early processes that initiate fault rupture, marking the transition from a stable, locked fault to an unstable, dynamically slipping one. Coseismic rupture is theoretically expected to be preceded by quasi-static slip, growing in space and time until it reaches a critical size that may trigger dynamic failure[15–21]. This behaviour is observed in both rate-and-state[16,20,22,25] and slip-weakening models[15,17,19,26]. These foundational studies suggest that the nucleation length, a critical measure of the size of the slipping zone before dynamic rupture, is primarily governed by the fracture energy of the fault interface, $G_c$, and the weakening rate within the nucleation region[19]. However, the physical parameters that control the duration and detectability of this stable phase remain poorly understood[25,27]. As a result, it is still unclear under which conditions geophysical observations might reveal precursory signals. Some earthquakes seem to be preceded by a prolonged nucleation stage[2–4,6,8,24], potentially accompanied by foreshocks[1,3,4,6,8,28], whereas many others exhibit no detectable dynamic precursor activity at all. These contrasting observations highlight the need for further investigation into the various physical mechanisms that may govern earthquake nucleation. Key open questions include: (1) whether all earthquakes exhibit quasi-static nucleation; (2) how foreshocks affect nucleation; and (3) whether the slip distance governing dynamic rupture also applies to nucleation.

We report laboratory experiments that, for the first time to our knowledge, directly capture how foreshocks control the nucleation of earthquakes. A small initial slip event sets the transient sliding velocity, which in turn dictates the duration and scale of the nucleation phase before dynamic rupture. Our results are supported by a theoretical framework based on a rate-dependent Griffith-like equation of motion (EoM), formulated using rate-and-state friction. Finally, we extend this theoretical framework to natural earthquakes preceded by foreshocks, enabling constraint of the characteristic nucleation parameters of large earthquakes, challenging standard estimates.

## Two decades of nucleation duration

We conducted experiments using a biaxial loading apparatus (Fig. 1a; see Methods for details). Three representative nucleation phases from a single experiment are shown in Fig. 1b–e. Despite having nearly identical initial stress conditions (Fig. 1b), the duration of the nucleation phase varied by more than an order of magnitude, ranging from 0.6 to 12.7 ms (Fig. 1c–e), and reached maximum values of about 80 ms across the experimental suite, highlighting the intrinsic complexity of nucleation dynamics.

Note that the three selected representative nucleation phases are observed to initiate with a foreshock (Fig. 1c–e and Extended Data Figs. 1g–i and 2a–c). In many cases, further foreshocks can be clearly identified before the onset of the main rupture (Extended Data Fig. 1g,h). The initiation of nucleation with a distinct foreshock is a consistent feature across the experimental dataset. The primary exceptions

[1]Université Côte d'Azur, CNRS, Observatoire de la Côte d'Azur, IRD, Géoazur, Sophia Antipolis, France. [2]Department of Civil and Resource Engineering, Dalhousie University, Halifax, Nova Scotia, Canada. [3]Navier, ENPC, Institut Polytechnique de Paris, Université Gustave Eiffel, CNRS, Marne-la-Vallée, France. ✉e-mail: Barnaby.fryer@geoazur.unice.fr

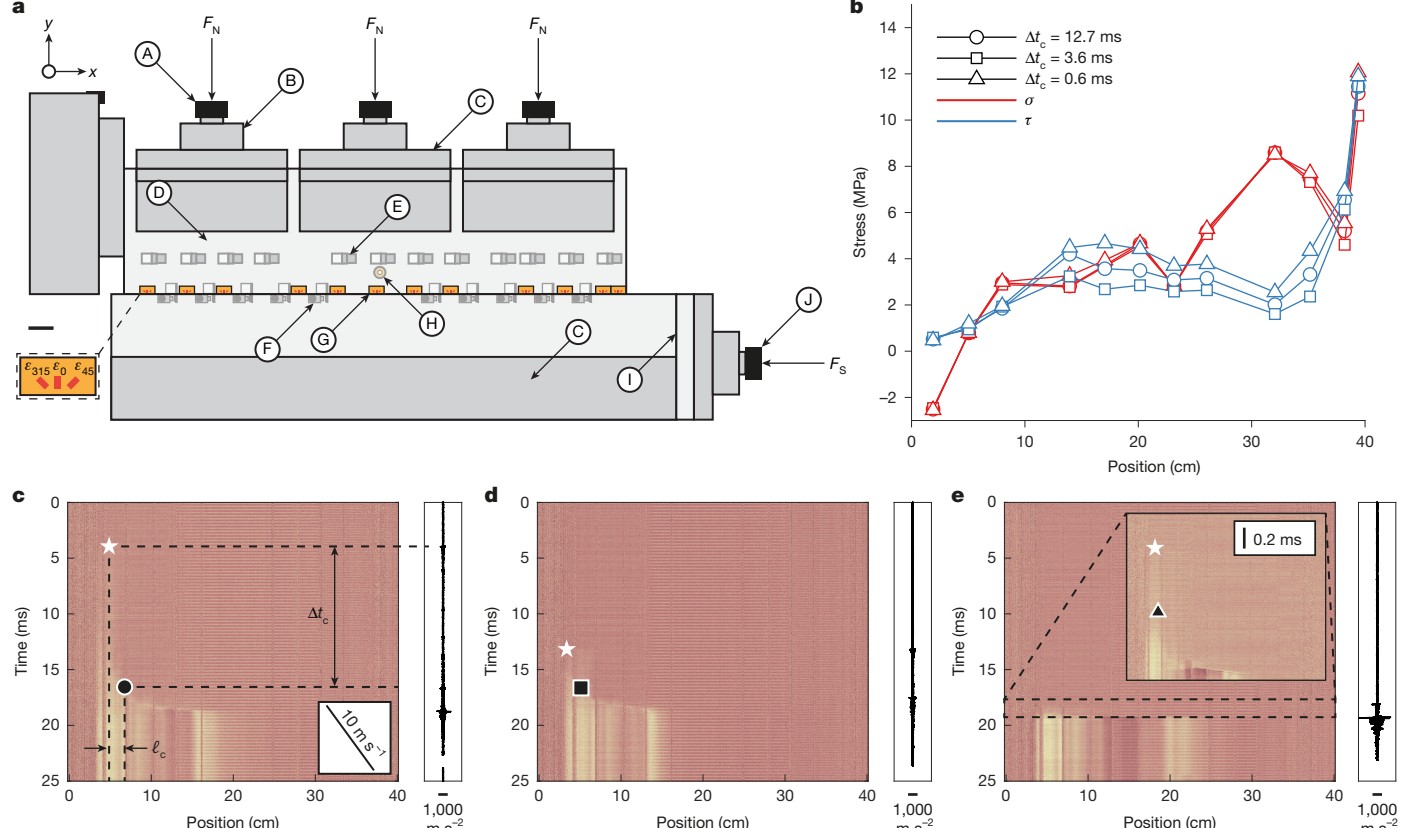

**Fig. 1 | Large variations in nucleation duration despite similar stress states.** **a**, The experimental set-up of frictional stick-slip experiments. Two PMMA blocks (D) with lengths, heights and widths of $(x, y, z) = (40, 10, 1)$ cm and $(x, y, z) = (45, 10, 1.8)$ cm are contained in a steel sample holder (C). A normal load, $F_N$, is first applied by three vertical hydraulic pistons (A), with its nominal value given by an analogue gauge and value in force measured by a load cell (B). Then, a shear load, $F_S$, is applied using a fourth piston (J) by a hydraulic pump at a constant injection rate. A spacer can be adjusted to allow for varied stress states (I). Twelve strain-gauge rosettes are located at $y = 3$ mm (G). Ten displacement sensors measure local slip (F). Twelve horizontally oriented accelerometers are located at $y = 30$ mm (E). One piezoelectric sensor is used to trigger a high-speed camera and synchronize all recording systems (H). The use of the high-speed camera with a polarized light source and the photoelastic properties of PMMA allow for the tracking of the propagating crack tip[37,38] (Methods). $\varepsilon_{315}$, $\varepsilon_0$, and $\varepsilon_{45}$ denote the strains oriented 315°, 0°, and 45° clockwise from the $y$ direction. Scale bar, 20 mm. **b**–**e**, Three example events, which are the 6th, 2nd and 9th dynamic events, respectively, recorded from the same experiment at a nominal normal stress of 150 bar but with different nucleation durations (12.7, 3.6 and 0.6 ms). **b**, The stress profiles 0.5 s before the dynamic events. **c**–**e**, The corresponding videograms of the dynamic events, which visualize rupture progression. The acceleration recorded by the left-most accelerometer is shown to the right of the videogram. Nucleation is taken to begin with a foreshock recorded by the accelerometers (white star) and to be achieved when rupture velocity reaches 10 m s$^{-1}$ (black shape corresponding to specific event), allowing for the calculation of nucleation duration, $\Delta t_c$, and length, $\ell_c$.

are events in which no discernible nucleation phase is present; in these instances, the initial foreshock evolves directly into dynamic rupture, without the intermediate stage of detectable quasi-static slip-front propagation (Extended Data Fig. 3).

## Weakening rate and nucleation duration

As slip, $\delta$, initiates within the nucleating patch, the shear stress, $\tau$, begins to decrease, reflecting fault weakening. Initially, the slip-weakening rate, $\frac{d\tau}{d\delta}$, is steep, meaning that a substantial stress drop occurs over a small amount of slip. However, the weakening rate progressively decreases as slip continues, eventually approaching a quasi-steady residual shear stress level (Fig. 2). Previous studies have shown that sharper weakening rates result in smaller nucleation lengths[17,19]. These studies generally indicate that weakening rate has a similar effect on nucleation duration, which further depends on, for example, loading rate[29]. Yet, the example events show similar weakening rates despite an order of magnitude difference in nucleation duration (Fig. 2).

Moreover, the two orders of magnitude variation in nucleation durations observed across the full experimental suite cannot be easily attributed to differences in weakening rate. In fact, the initial weakening seems to be insensitive to pressure, that is, non-frictional, as the drop in strength is essentially constant despite substantial differences in normal stress for many events.

## Foreshocks and the onset of sliding

Although the stress criticality does not seem to vary greatly between cases of slow and rapid nucleation (Fig. 1b), the initial impulse imparted to the nucleating patch does (Extended Data Fig. 4). The initial foreshock generates a burst of sliding velocity along the fault. This rapidly decays to a minimum transient sliding velocity, $V_{min}$ (Fig. 3a,b). The value of $V_{min}$ is found to be positively correlated with the size of the foreshock, such that larger foreshocks produce higher transient sliding velocities (Fig. 3b,c). This relationship suggests that the initial energy imparted by the foreshock plays a key role in controlling the evolution and duration of the nucleation phase.

Effectively, a clear correlation emerges between nucleation duration and foreshock size (Fig. 4a): events with longer nucleation phases are typically preceded by smaller foreshocks and accumulate greater amounts of quasi-static slip before transitioning to dynamic rupture. This trend is not only evident in the present experimental dataset but

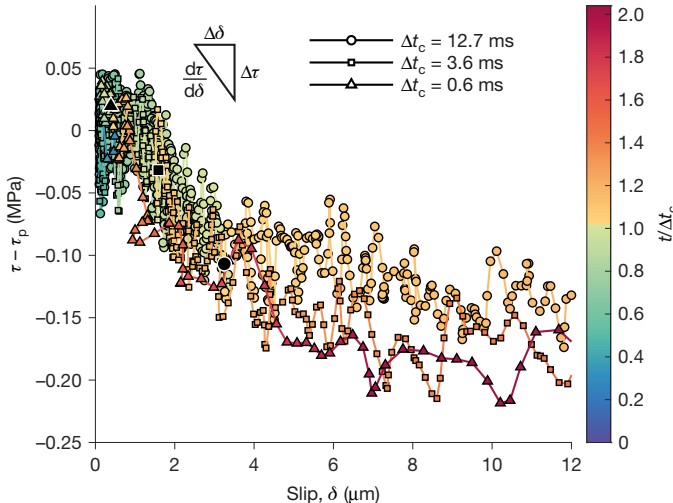

**Fig. 2 | Comparable weakening rates for an order of magnitude difference in nucleation duration.** The weakening (change of shear stress) as a function of slip within the nucleating patch. The figure corresponds to the same three events shown in Fig. 1. These three events occurred at similar normal (0.8, 0.8 and 0.9 MPa) and peak shear (1.1, 1.1 and 1.4 MPa) stresses. The colour bar indicates the time normalized by the nucleation duration and a black shape corresponding to the event indicates the moment nucleation is considered to be achieved. The data correspond to a position of 5.2 cm, the approximate position of the foreshocks as seen on the videograms in Fig. 1c–e. The data of the two nearest strain gauges (5.1 and 8.0 cm) and displacement sensors (3.5 and 6.5 cm) are linearly interpolated to this position. The data are subjected to a low-pass filter of 10 kHz. The data are zeroed to the rupture front as seen by the high-speed camera (Fig. 1a), resulting in slight mismatch, evident in particular for small values of slip.

also seems consistent with observations from natural earthquakes (Fig. 4a). A general pattern of nucleation behaviour can thus be identified (Fig. 3a,b). The sequence begins with a foreshock that triggers a burst in sliding velocity locally along the fault and crack expansion. This sliding velocity then decays to a minimum, that is, the so-called transient sliding velocity $V_{min}$, corresponding to a quasi-static slip phase, before both sliding velocity and rupture velocity accelerate exponentially (Extended Data Fig. 2), ultimately leading to dynamic rupture onset.

Although natural earthquake rupture may not always be triggered by discrete foreshocks, larger foreshocks lead to higher transient sliding velocities, $V_{min}$, both during and immediately following such precursory activity. This trend is clearly evident in our experimental data, in which larger foreshocks correlate with higher values of $V_{min}$ (Fig. 3c).

## Equation of motion

We now seek to reconcile our experimental observations with a state-of-the-art rupture model, grounded in rate-and-state friction. Garagash[22] developed an EoM linking rate-and-state parameters and loading conditions to describe the nucleation and propagation of frictional ruptures. It reads:

$$K_{bg}(\Delta f_0; \ell, v_r) + K_{fs}(\Delta T; \ell) = K_c(v_r). \tag{1}$$

Here all terms are implicit functions of the frictional parameters ($a$, $b$, $L$), in which $a$, $b$ and $L$ and are the direct effect, evolution effect and state-evolution slip distance, respectively. $K_{bg}$ is the stress intensity factor (SIF) owing to the background stress. It depends on the initial overstress $\Delta f_0$ at ambient sliding velocity $V_0$, the crack half-length $\ell$ and rupture velocity $v_r$, following rate-and-state friction. $K_{fs}$ denotes the SIF contribution from the foreshock and depends on $\ell$ and a localized

Coulomb force of magnitude $\Delta T \propto \mu' \delta_a$, in which $\mu'$ is the apparent shear modulus and $\delta_a$ is the asperity slip. Ultimately, the two SIF contributions must balance the fault toughness $K_c$, which increases with $v_r$, owing to the rate-dependent frictional response. The exact form of the SIFs is not required to qualitatively match experimental observations, as long as $K_{bg}$ increases with $\ell$ but $K_{fs}$ decreases with $\ell$. This defines a 'well' in the energy available to drive the rupture at some intermediate crack length[30,31], which, when balanced with $K_c(v_r)$, sets a transient minimum in the rupture propagation and sliding velocities (Methods).

Nucleation dynamics are therefore controlled by two parameters, $\Delta f_0$ and $\Delta T$, which scale the aforementioned dependence of $K_{bg}$ and $K_{fs}$ on the crack length, respectively. As shown in Extended Data Fig. 6, the EoM predicts three possible nucleation scenarios: (1) a transient deceleration followed by arrest without the nucleation of dynamic rupture; (2) deceleration to $V_{min}$ followed by a prolonged quasi-static phase during which both slip and rupture velocity accelerate, eventually transitioning to dynamic rupture; or (3) near-immediate acceleration into dynamic rupture, without a quasi-static phase. All three behaviours are observed in our experimental dataset (Extended Data Fig. 7, Fig. 3a and Extended Data Fig. 3) and can be rationalized using the EoM. Indeed, following equation (1), rupture initiation is initially driven by the impulsive force $\Delta T$. As the crack grows, the foreshock contribution $K_{fs}$ decreases with increasing $\ell$, promoting rupture deceleration. However, the background contribution $K_{bg}$ increases with $\ell$ and tends to accelerate the crack. Crack growth, in turn, raises the sliding velocity $V$, which increases the transient stress drop, owing to the rate-weakening nature of the interface, and feeds $K_{bg}$. This feedback can sustain or arrest rupture depending on foreshock magnitude, represented by $\Delta T$, as it defines early-stage sliding velocity and thus the transient strength of the interface. When $\Delta T$ is too small, the increase in $K_{bg}$ with increasing crack half-length $\ell$ is insufficient to offset the associated decrease in $K_{fs}$ and rupture arrests (regime 1). By contrast, a large $\Delta T$ generates high transient sliding velocities, leading to rapid growth of $K_{bg}$ and immediate transition to dynamic rupture (regime 3). Between these extreme cases, a moderate $\Delta T$ induces a slower but self-sustained increase in $K_{bg}$, reinforced by the positive feedback of $\ell$ and $V$, resulting in delayed dynamic rupture (regime 2). Here the overstress $\Delta f_0$ primarily sets the baseline value of the transient stress drop and thus controls the threshold values of $\Delta T$ that determine the transition between nucleation regimes.

Moreover, the EoM offers an explanation for the observed variability of the nucleation length $\ell_c$ and the lack of a clear correlation with normal stress $\sigma$ (Fig. 4b), which is otherwise expected in nucleation models[19,20,32] and reported in some experimental studies[27,33,34], neither of which generally consider impulses. The predicted nucleation size of the EoM for ageing frictional faults is bounded by the maximum value $\ell_\infty \approx \frac{1}{\pi} \frac{1}{(1-a/b)^2} \frac{\mu' L}{b\sigma}$ (ref. 20). This terminal value is, however, approached by nucleation transients driven by a vanishingly small impulse on a neutrally or slightly understressed fault (Extended Data Fig. 8). Otherwise, both the overstress and the impulse imparted by the foreshock act to materially reduce the nucleation length compared with $\ell_\infty$ by as much as an order of magnitude (Extended Data Fig. 8).

In summary, the EoM offers a physical explanation for the observed rupture behaviour during nucleation. The advancing crack tip draws energy from both the inherent stress state along the fault and the stress concentration (or redistribution) generated by the foreshock. However, these two contributions are not necessarily readily constrainable in both laboratory and natural settings. Instead, as these two components are the only contributors to the energy balance of the EoM, their combination results in one unique nucleation behaviour or crack development path (Fig. 3b). This unique crack development path is characterized by the minimum transient sliding velocity $V_{min}$, which then, as a single parameter, dictates nucleation behaviour (Fig. 4c and Extended Data Fig. 9a).

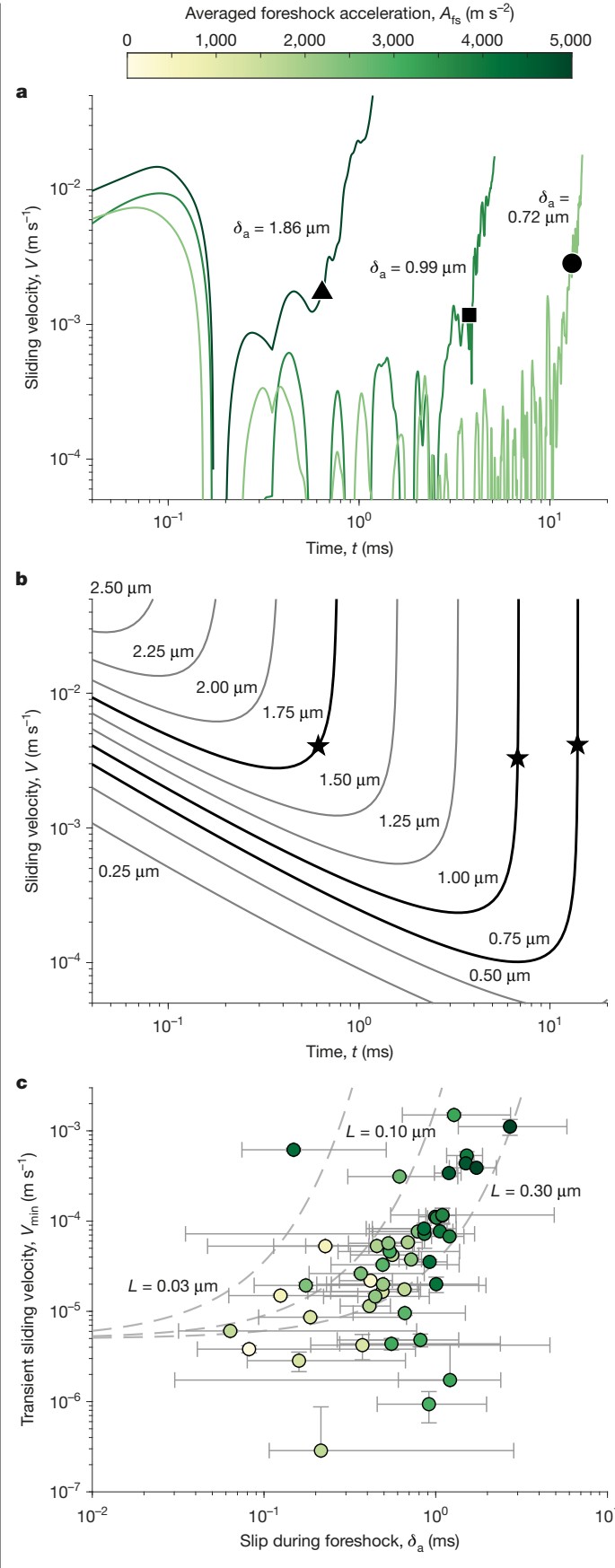

**Fig. 3 | Foreshock size predicts the transient sliding velocity of the nucleating patch. a**, The sliding velocity of the three events shown in Fig. 2. The black symbols show $v_r = 10$ m s$^{-1}$. Note the initial sliding velocity due to the foreshock, its reduction after the foreshock and the reacceleration up to dynamic rupture propagation. The foreshocks of these three events have estimated slips, $\delta_a$, of 0.72, 0.99 and 1.86 μm (Extended Data Fig. 2e) in order of increasing foreshock size, $A_{fs}$ (Methods). **b**, EoM simulation runs, characterized by an asperity slip, $\delta_a$; $\Delta f_0 = 0$. The simulations with $\delta_a$ most closely corresponding to the experimental events in **a** are shown with dark lines with $v_r = 10$ m s$^{-1}$ indicated by stars (Extended Data Fig. 5). **c**, After the foreshock, the slip evolves exponentially with time (Extended Data Fig. 2). This assumed evolution is fit and its derivative is evaluated at the foreshock, yielding the initial minimum sliding velocity of the patch at the onset of nucleation. The slip accrued during the foreshock, $\delta_a$, is taken as the jump in slip at the moment of the foreshock (Extended Data Fig. 2). Larger foreshocks yield larger $V_{min}$. This result is predicted by the EoM, considering $a/b = 0.55$, state-evolution slip distances, $L = 0.03$, 0.10 and 0.30 μm, and ambient fault sliding velocity, $V_0 = 10^{-6}$ m s$^{-1}$. The dashed lines represent EoM predictions for the case with non-negative overstress. Events with negative overstress are predicted to fall below these lines. For **a** and **c**, the foreshock amplitude is given by the colour bar. The error of $\delta_a$ is estimated as the difference between the maximum and mean displacement recorded within 0.5 ms after the foreshock, with the lower bound limited to 50% of the measured value. The error of $V_{min}$ is estimated by propagating the 95% confidence interval of the exponential fit to the slip–time data.

in our experiments but also in larger-scale (about 1 m) rock friction experiments without a distinguishable foreshock and for quasi-statically slipping patches of natural earthquakes, in which the onset of nucleation is identified by a foreshock. These foreshocks provide an empirical basis for estimating nucleation duration, $\Delta t_c$. Larger impulsive forces (that is, larger foreshocks) result in higher transient minimum sliding velocities $V_{min}$ (Fig. 3b,c) and shorter nucleation duration $\Delta t_c$ and nucleation length $\ell_c$ (Fig. 4c). At 'large' sliding velocities $V_{min}$, $\Delta t_c$ scales as the inverse of $V_{min}$, $\Delta t_c \propto \frac{L}{V_{min}}$. However, a progressive shift from this inverse power-law behaviour to a constant nucleation duration $\Delta t_c \cong \Delta t_0$ is predicted at the transition between regimes 1 and 2, when $V_{min}$ approaches the ambient sliding velocity $V_0$ (Extended Data Fig. 9c). The relationship between $\Delta t_c$ and $V_{min}$ is quantitatively predicted by the EoM using the frictional properties of polymethylmethacrylate (PMMA) and realistic laboratory values for the state-evolution slip distance $L$ (equation (17)).

Notably, the observed dependence of nucleation duration on transient sliding velocity seems to extend to quasi-static slip patches preceding certain natural earthquakes (Fig. 4d). The slope of this relationship is consistent between our laboratory experiments and natural events, but the natural earthquake data are offset by several orders of magnitude. This offset implies that, for a given transient velocity, natural faults exhibit much longer nucleation durations than those observed in the laboratory. This shift is attributable to differences in frictional properties, specifically much larger state-evolution slip distances and smaller values of the rate-weakening coefficient $b - a$ for natural faults, which may depend on fault lithology[35]. Furthermore, part of the discrepancy may arise from the uncertainty in estimating transient sliding velocities for natural earthquakes; our values, based on indirect constraints and the assumption of exponentially increasing slip, probably overestimate the $V_{min}$ during nucleation. Indeed, these data contain notable uncertainty because transient sliding velocities for natural earthquakes must be inferred indirectly from either geodetic slip inversions or by assuming small repeater earthquakes recover accrued aseismic slip deficit (Extended Data Fig. 10). Nevertheless, variations in the $a/b$ ratio arising from possible differences in rheology and/or variations of state-evolution slip distance $L$, possibly related to fault maturity[36], are sufficient to explain the discrepancies observed among the different earthquakes analysed (Fig. 4d).

Our experimental results provide new insight into the physics of earthquake nucleation and allow us to revisit several key questions

## Transient slip rate controls nucleation

The nucleation length of the experimental events is positively correlated with the nucleation duration (Fig. 4b). This trend holds not only

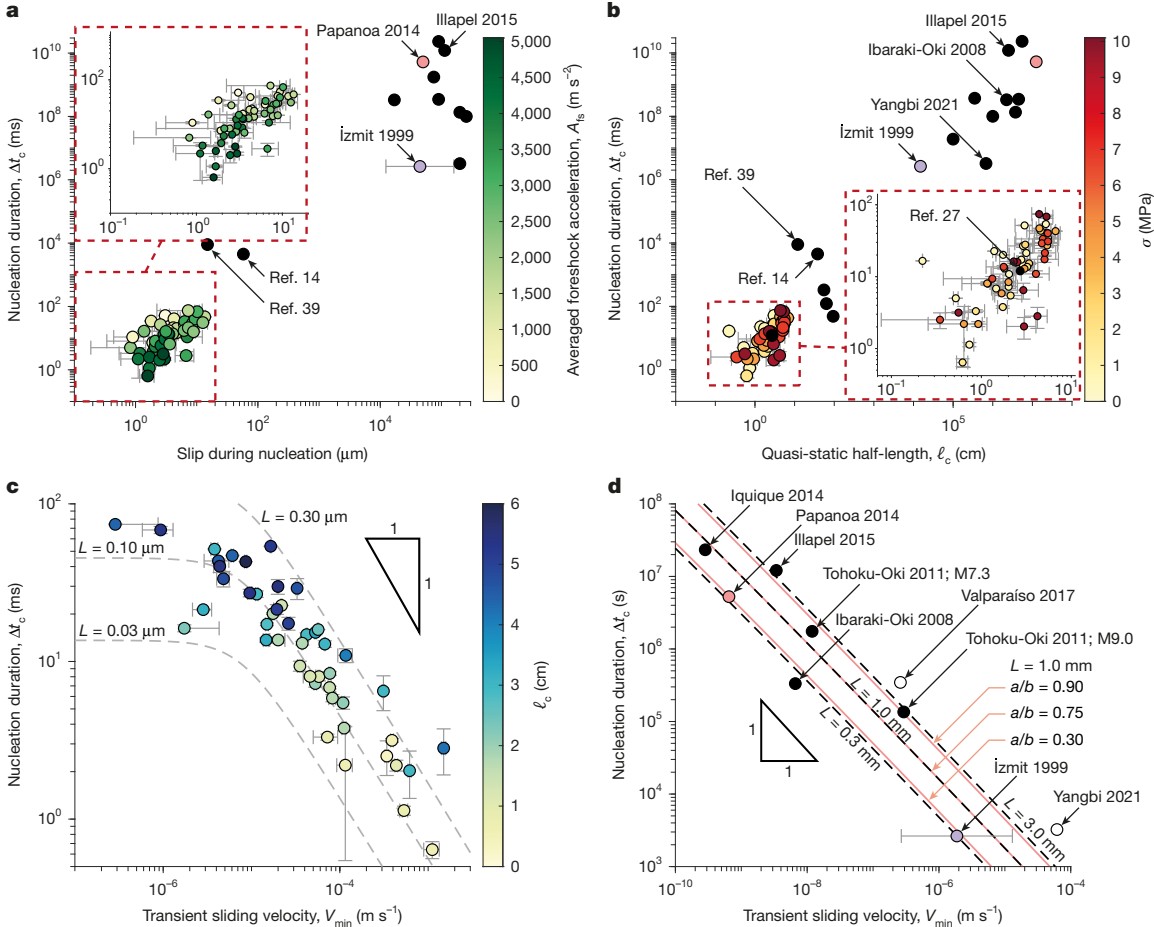

**Fig. 4 | Transient sliding velocity predicts nucleation duration and length. a**, Prolonged nucleation phases accrue more slip. Laboratory[14,39] and natural[1,2,4–6,9–11,40] earthquakes are included with black dots. The foreshock amplitude is given by the colour bar. Slip-during-nucleation error is estimated by evaluating the slip within ±0.05 ms of $v_r$ reaching 10 m s[-1]. **b**, The nucleation duration, $\Delta t_c$, as a function of the nucleation length, $\ell_c$. The error of both $\ell_c$ and $\Delta t_c$ was found by reassessing both quantities with $v_r = 20$ m s[-1]. Natural[1,2,4–6,9–11,13,40–42] and laboratory[14,27,39,43,44] earthquakes are plotted with black dots. The local value of normal stress is given by the colour bar. In **a** and **b**, the laboratory events from the literature are not preceded by a distinguishable foreshock. **c**, The nucleation duration as a function of $V_{min}$. Predictions from the EoM are shown by dashed grey lines based on state-evolution ageing law slip distances, $L$, of 0.03, 0.10 and

0.30 μm, $a/b = 0.55$ and an ambient sliding velocity of $10^{-6}$ m s[-1]. The colour bar indicates the quasi-static half-length at nucleation. **d**, The estimated duration and transient sliding velocity of the nucleation phases of natural earthquakes[1,2,4,6,9–11,13,40]. These data are estimated from inverted GNSS data or repeater earthquakes. However, Papanoa 2014 (pink) was a slow-slip event not triggered by a foreshock[40]. İzmit 1999 (purple) is a debated case whose repeaters may or may not be linked to aseismic slip[1,11]. Time series data for Yangbi 2021 (ref. 13) and Valparaíso 2017 (ref. 6) (white) are not available such that a simple average is used to estimate $V_{min}$. EoM fits in dashed lines for state-evolution ageing law slip distances of 0.3, 1.0 and 3.0 mm using $a/b = 0.75$ and an ambient sliding velocity of $10^{-12}$ m s[-1]. Fits with $L = 1.0$ mm and $a/b = 0.30$, 0.75 and 0.90 are shown in pink.

about the onset of seismic rupture. (1) Are all earthquakes preceded by a quasi-static nucleation phase? Although most of the dynamic events in these experiments exhibit a quasi-static nucleation phase, observable quasi-static nucleation is not a prerequisite for dynamic rupture. Given an appropriate overstress, a sufficient impulse (provided here by a foreshock) can merge into dynamic rupture without an accelerating quasi-static slip phase. Similarly, although in our laboratory experiments all dynamic events exhibiting observable nucleation phases are initiated by seismic foreshocks, the impulse that initiates quasi-static nucleation can be supplied by any phenomenon causing localized acceleration, such as tectonic loading or other aseismic processes[20]. (2) How does the occurrence of a foreshock influence the nucleation of a large earthquake? On rate-weakening faults, foreshocks initiate a positive feedback loop between the sliding velocity and the stress drop, causing $K_{bg}$ to eventually grow to the dynamic range of the rate-dependent fault toughness $K_c(v_r)$ with $v_r \approx c_s$. This drives the growth of the aseismically slipping patch and causes main-shock nucleation in sufficiently stressed conditions, making rate-weakening interfaces sensitive to foreshocks, especially when compared with

rate-independent frameworks, and providing a mechanism compatible with a rate-dependent cascade up model[14], even in cases in which foreshocks are the dominant driving mechanism of the rupture tip (that is, cascade-like). Larger foreshocks shrink the timescale and reduce the stress criticality required for nucleation. The largest foreshocks can themselves merge into a mainshock under sufficiently stressed conditions. Yet, observations of quasi-static nucleation phases in foreshock to main-shock sequences are relatively rare in nature. This is probably because the impulses of most foreshocks are too small to lead to nucleation after the initial slowdown (regime 1), too large to not directly merge into a mainshock (regime 3) or because quasi-static phases are simply too difficult to identify in observations[24]. (3) Does the characteristic slip distance representative of dynamic rupture also describe nucleation? These results allow for the inference of reasonable ranges (between 0.3 and 3.0 mm) of state-evolution slip distances, $L$, of real earthquakes during their nucleation phases (Fig. 4d). These are a notable departure from the larger values of $L$ inferred for dynamic rupture; for example, the state-evolution distance of 0.8 m used for modelling Tohoku-Oki[23].

Taken together, our results demonstrate that nucleation is a dynamic, rate-dependent process that can manifest in several forms and whose characteristics hold important implications for understanding the conditions under which rupture initiates and evolves. Beyond earthquakes, our results highlight a fundamental mechanism of frictional failure that may also govern the stability of engineered and tribological interfaces, as well as landslides and icequakes.

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

## Methods

### Experimental set-up

A biaxial apparatus, CrackDyn, was used to perform the experiments (Fig. 1a). The apparatus housed two PMMA plates ($40 \times 10 \times 1$ cm and $45 \times 10 \times 1.8$ cm). Experiments were conducted on PMMA rather than natural rocks because it provides three main advantages for scaling laboratory observations to natural fault systems. First, owing to its lower elastic stiffness compared with rocks, the state-evolution slip distance, $L$, and cohesive zone size, $X_c$, are smaller in PMMA. As a result, a laboratory-scale PMMA fault is dynamically representative of a much larger natural fault, whereas a laboratory rock fault essentially remains of the same scale as its natural counterpart. Second, the reduced elastic properties and fracture energy, $G_c$, of PMMA ensure that nucleation lengths, $\ell_c$ and $\ell_\infty$, remain smaller than the total fault dimension in the experiment, which is rarely the case in rocks, in which nucleation patches can be comparable with or larger than the laboratory fault. Finally, the birefringent properties of PMMA allow for photoelastic visualization, enabling direct observation of nucleation growth and subsequent rupture propagation using high-speed imaging.

The plates were characterized by a static Young's modulus, $E$, of 3 GPa and a Poisson's ratio, $v$, of 0.35. A normal load was applied by three vertical pistons by means of steel sample holders. The pistons were supplied by a Enerpac P141 hydraulic pump. Similarly, a shearing load was applied by a single horizontal piston. This piston was supplied by a Top Industrie PMHP-35-1000 hydraulic pump. The force applied by each piston was recorded by a Scaime K13 load cell at 500 Hz. Local strains were recorded by 13 350-ohm strain-gauge rosettes (Micro-Measurements C5K-06-S5198-350-33F) located 3 mm from the sample–sample interface. The strain gauges recorded at 2 MHz and were amplified by a factor of 10 by Elsys SGA-2 MK2 amplifiers. Ten PHILTEC model D100-E2H2PQTS displacement sensors were fixed across the sample–sample interface and recorded the local displacement with a recording frequency of 2 MHz. Thirteen Brüel & Kjær Type 8309 accelerometers were glued horizontally approximately 30 mm from the sample–sample interface and recorded at 2 MHz. These signals were amplified by a NEXUS conditioning amplifier 2692. Finally, an EFFILUX EFFI-LINE3-WTR-600-000-POL-PWR-C light source was used to shine linearly polarized light through the sample. This light then passed through a second linear polarizer before reaching a Phantom TMX 6410 high-speed camera, which recorded $1,280 \times 32$ pixels at 1 MHz. Considering that PMMA is a birefringent material, this allowed for the use of photoelasticity to track changes in stress across the sample–sample interface[27,37,38,45]. A piezoelectric sensor attached to one of the PMMA samples was used as a trigger for an oscilloscope (Picoscope 4224A), which generated a TLL-like signal that triggered the camera and allowed for the synchronization of the other acquisition systems.

### Experimental approach

Experiments were begun by applying 100, 150, 200, 250 or 300 bar nominal normal stress (that is, the pressure indicated on the analogue gauge of the pump supplying the vertical pistons). Next, the shear stress was increased by setting a constant flow rate, 3 cm³ min⁻¹, on the pump supplying the horizontal piston. These conditions were kept constant until enough slip had been accumulated such that the displacement sensors were out of range (approximately 0.5 mm). A $1.3 \times 0.3 \times 10$-cm stopper was placed between the lower-most PMMA sample and the horizontal piston. By adjusting this stopper before the experiment such that it was in either a raised or a lowered position, the local loading conditions could be altered such that a wider variety of events could be produced. Each of the five normal stresses were tested with both stopper positions, resulting in ten total experiments and 94 dynamic events.

### Data treatment

As the strain gauges were set up in a quarter-bridge configuration, the strain, $\varepsilon$, could be found for the $i$th strain gauge as,

$$\varepsilon_i = \frac{-4U_i}{U_{ex}\left(J_f J_{amp}\left(1 + \frac{2U_i}{U_{ex}J_{amp}}\right)\right)}, \tag{2}$$

in which $U_i$ is the voltage reading of the individual strain gauge, $U_{ex}$ is the excitation voltage, $J_{amp}$ is the amplification gain and $J_f$ is the gauge factor of the strain gauge.

In this three-strain-gauge rosette, the strain gauges were oriented at 45° from one another, such that the principal strains were then found as

$$\varepsilon_{xx} = \varepsilon_{315} + \varepsilon_{45} - \varepsilon_v, \ \varepsilon_{yy} = \varepsilon_v, \ \varepsilon_{xy} = \frac{\varepsilon_{45} - \varepsilon_{315}}{2}, \tag{3}$$

in which $\varepsilon_v$ is the vertical direction and $\varepsilon_{315}$ and $\varepsilon_{45}$ are oriented 315° and 45° clockwise from vertical (Fig. 1a). Here $x$ refers to the horizontal direction and $y$ refers to the vertical direction.

These strains were then converted to stress, $\sigma$, considering Hooke's law in plane stress conditions,

$$\begin{bmatrix} \sigma_{xx} \\ \sigma_{yy} \\ \tau \end{bmatrix} = \frac{E}{(1+v)(1-2v)} \begin{bmatrix} 1-v & v & 0 \\ v & 1-v & 0 \\ 0 & 0 & 1-2v \end{bmatrix} \begin{bmatrix} \varepsilon_{xx} \\ \varepsilon_{yy} \\ \varepsilon_{xy} \end{bmatrix}. \tag{4}$$

The values of stress were filtered with a low-pass filter at 10 kHz.

The camera recorded greyscale images in a $1,280 \times 32$-pixel matrix in which the values ranged from zero (black) to a maximum value (white) of light intensity. Taking the first 20 frames as reference, the future frames were compared with this reference to detect changes in stress during rupture propagation, made possible by the stress-induced birefringence of PMMA and the use of linearly polarized light[37,38]. One of the 32 rows (corresponding to the 1,280 pixels closest to the sample–sample interface) was isolated and used for this analysis. The readings of the accelerometers were converted to accelerations using the individually calibrated sensitivities of each sensor. Finally, the readings of the displacement sensors were converted from voltage to gap distance using the calibration provided by PHILTEC. They were then filtered with a low-pass filter at 10 kHz. The nucleation duration was then defined as the interval between the foreshock recorded by the accelerometers and the transition to dynamic rupture, marked by rupture velocities exceeding 10 m s⁻¹ as recorded by high-speed imaging.

### Foreshocks and nucleation length scaling

In the present experimental dataset, the onset of nucleation is systematically associated with a foreshock. This feature is specific to this series of experiments and probably reflects the heterogeneous stress distribution along the fault, which varies between PMMA plates. In other laboratory studies, nucleation has been observed to occur without any identifiable foreshock. In such cases, the influence of foreshocks on the nucleation process cannot be investigated and the nucleation size is, to a first order, defined by its theoretical asymptotic value, $\ell_\infty$ (ref. 20), for understressed faults. The lack of foreshock initiating the quasi-static nucleation phase in other bare-rock-surface experiments can probably be explained by the scaling parameter $\frac{\Delta T}{\mu' L}$, in which $\Delta T \approx \mu' \delta_a$, such that the scaling parameter reads $\frac{\delta_a}{L}$. Because $L$ in PMMA (about 0.1 μm) is one to two orders of magnitude smaller than typical values in bare-surface rock samples (1–10 μm (ref. 46)), achieving a comparable influence of a foreshock in rock would require foreshock slips that are 10–100 times larger, or on the order of tens to hundreds of microns. As shown, for example, in ref. 14, such slip amplitudes are in many cases similar to those of the mainshock, suggesting that the foreshocks observed in

those samples are probably much smaller and therefore contribute less to the nucleation dynamics. However, on larger natural faults, the absolute slip accumulated during foreshocks can be substantially greater, potentially enabling foreshocks to exert a dynamically notable perturbation to the EoM. Note, however, that the EoM still describes cases with vanishingly small foreshocks. The limiting case of $\Delta T \to 0$ results in $\ell_c \to \ell_\infty$ for the ageing law (Extended Data Fig. 8). Furthermore, it should be noted that $\Delta T$ need not be supplied by a seismic foreshock.

The coexistence of foreshocks and a larger nucleation front can be rationalized by considering the scale dependence of nucleation. At the scale of several micron-sized asperities, local stress concentrations lead to much higher effective normal stresses than the macroscopically applied value. Because the theoretical nucleation size scales inversely with normal stress[20],

$$\ell_\infty \approx \frac{1}{\pi} \frac{1}{(1-a/b)^2} \frac{\mu' L}{b\sigma} = \frac{1}{\pi} \frac{1}{(1-a/b)^2} \ell_b, \tag{5}$$

in which $\ell_b$ is the elasto-frictional length scale, these local stress amplifications reduce the critical nucleation length at the asperity scale, enabling small slip instabilities that manifest as foreshocks. At the same time, the macroscopic nucleation process spans a much larger portion of the fault, controlled by the bulk stress state and the effective average $\sigma$, and can therefore extend over dimensions much greater than the asperity scale. This multiscale interplay explains how foreshocks, driven by localized asperity failure, can coexist with a larger quasi-static nucleation front that evolves towards dynamic rupture, as recently highlighted in the homogenized regime[26].

**Transient sliding velocity**
Although the onset of stable slip is systematically triggered by a foreshock, the associated sliding velocity is too small to be found as an instantaneous measurement. Instead, slip is seen to increase exponentially with time (Extended Data Fig. 2) and can fit with an exponential function. The derivative of this function is found just after the foreshock to provide the transient sliding velocity. A similar procedure is applied to the data digitized for natural earthquakes (Extended Data Fig. 10 and Extended Data Table 1). Note that, in the case of data for natural earthquakes, these data rely on either inversion of the fault slip based on geodetic data or the supposition that the slip during repeater earthquakes is representative of the aseismic slip of the patch[47]. Both methods are subject to several assumptions and potential errors. Illapel 2015 exhibited repeater earthquakes beginning approximately 140 days before the mainshock[9]. The cumulative slip of these repeaters was used to infer aseismic slip, assumed to be representative of quasi-static slip associated with nucleation. Iquique 2014 exhibited repeater earthquakes starting around 270 days before the mainshock[4]. These repeaters were used to infer aseismic slip, which was assumed here to be associated with the nucleation of the mainshock. Alternatively, we could use GNSS data[3], which showed substantial movement beginning approximately 17 days before the mainshock to invert for aseismic fault slip. Note that a $M_w$ 6.7 earthquake occurred at the onset of this aseismic slip. However, a time series of this data is not available, such that a calculation of $V_{min}$ is not possible. Papanoa 2014 exhibited a slow-slip event that began two months before a mainshock[40]. Although GNSS data can be used to infer the slip velocity at the onset of this period, the notion of $V_{min}$ is unclear, as no known foreshock is associated with the start of the event. Ibaraki-Oki 2008 exhibited repeater earthquakes in the approximately 4 days leading up to the mainshock, with this sequence accelerating substantially approximately 12 h before the mainshock[10]. These repeater events were used to estimate aseismic slip assumed to be associated with nucleation. Tohoku-Oki 2011 was preceded by repeating earthquakes, beginning approximately 23 days before the mainshock[2]. Then, a magnitude 7.3 foreshock occurred approximately 2 days before the mainshock. The repeater earthquakes were used to infer aseismic slip. It is considered that the first foreshocks initiated nucleation, resulting in a slip velocity peak followed by a slip velocity minimum (Extended Data Fig. 11). Next, the M7.3 probably had a strong influence on the nucleation dynamics, resulting in another slip velocity peak and a new slip velocity minimum (Extended Data Fig. 11). It is possible that the initial foreshocks are linked to the nucleation of the M7.3, which itself governs the nucleation of the M9.0; however, note that other authors consider that the associated pre-slip occurred away from the mainshock rupture zone[48]. Here the initial foreshocks are considered to initiate the nucleation of the M7.3 event, which itself is considered to initiate the nucleation of the M9.0 mainshock. İzmit 1999 exhibited a 44-min-long foreshock sequence[1,11]. However, the exact nature of this sequence is controversial and it is unclear whether there was a precursory aseismic driving process. Here the assumption is made that these are repeating earthquakes representative of aseismic slip, but it should be emphasized that this may not be the case. Valparaíso 2017 was observed to be preceded by repeater-type seismicity and trenchward movement observed in GNSS data beginning 2 days before the mainshock[6]. Although fault slip can be inverted from the GNSS data, the full time series is not available. Instead, a simple average of fault slip over the time interval was used to provide an upper-bound estimate of $V_{min}$. Yangbi 2021 was associated with migrating foreshocks and repeating earthquakes, with a rapid migration beginning 30 min before the mainshock[13]. During this period, GNSS data allows for the inversion of aseismic slip, assumed to be associated with nucleation. Unfortunately, the time series of this data is not available, so a simple average is again used to provide an upper-bound estimate of $V_{min}$.

**Foreshock magnitude**
The foreshocks at the origin of all observable nucleation phases in these experiments are located using error minimization based on arrival times. This localization allows for the assessment of the event magnitude by means of an averaged acceleration, $A_{fs}$, found as[49],

$$A_{fs} = \sqrt{\frac{1}{N_a} \sum_{i=1}^{N_a} \left( \frac{r_i}{0.01} A_i^{max} \right)^2} \tag{6}$$

in which $N_a$ is the number of accelerometers, $r$ is the distance from the sensor to the event location (normalized by a reference sphere of 0.01 m) and $A_i^{max}$ is the maximum acceleration measured by the sensor during the event.

**Unified framework for earthquake nucleation from laboratory to nature**
The EoM of the crack originates from the Griffith's energy balance, which together with a set of simplifying assumptions, can allow for the solving of the crack front propagation[50]. Recently, the EoM framework has been applied to crack-like slip transients on rate-and-state frictional faults[20,22,51]. Here we follow the EoM treatment by Garagash[22], who considers a plane-strain or anti-plane shear crack of half-length $\ell$, propagating at instantaneous velocity $v_r$ along a planar 1D fault, characterized by the ambient sliding velocity $V_0$ and 'overstress', the difference between the initial stress (ratio) $f_0 = \tau_0/\sigma$ and the steady-state level of friction at the ambient sliding velocity, $\Delta f_0 = f_0 - f_{ss}(V_0)$. The fault is embedded in an infinite elastic medium and its rate-and-state-dependent resistance to sliding[32] is governed by three parameters: the direct effect coefficient $a$, the state-evolution coefficient $b$ and the characteristic slip distance $L$ over which the state of the sliding surface is renewed (not to be confused with the apparent slip-weakening distance of the rupture, $\delta_c$). Evolution of the state is often modelled in the framework of either ageing[16] or slip[52] laws, which manifest in different apparent weakening behaviours of strength with incurred slip in a transient (for example, behind a propagating rupture front). Specifically, a transient governed by the ageing law corresponds to a rate of weakening with slip ($W = -\frac{d\tau}{d\delta}$), which is approximately independent of the strength

of the transient[22,53], $W \approx b\sigma/L$, whereas for the slip law, the weakening rate increases with the transient strength, $W \approx \Delta f_p \sigma/L$, in which $\Delta f_p$ is the maximum departure of friction from the steady state during the transient (that is, 'friction breakdown'). Conversely, the apparent slip-weakening distance $\delta_c$ required to 'break' the strength by amount $\Delta f_p \sigma$ from the peak to the steady-state 'residual' value at the above values of the weakening rate $W$ is an increasing function of $\Delta f_p$, $\delta_c = \left(\frac{\Delta f_p}{2b}\right)L$, for the ageing law, whereas it is invariant, $\delta_c \approx L$, for the slip law. Considering that friction experiments have shown the relevance of the ageing law for PMMA[54], we use the ageing law for the rest of the analysis. However, previous studies have indicated that the slip law may be more appropriate for natural fault material[55], conversely to PMMA. Note that the model presents a qualitatively similar behaviour using the slip law (Extended Data Fig. 6). For consistency and simplicity, we extend our results to natural earthquakes using the ageing law rather than the slip law (Fig. 4d).

**EoM framework.** The EoM considers the balance between the energy release rate $G = \overline{K}^2/(2\overline{\mu})$ into the propagating crack tip with the fracture energy of the frictional breakdown process $G_c$. Here $\overline{K} = k(v_r)K$ is the dynamic SIF obtained from the static value $K$ multiplied by the wave-mediated dynamic prefactor $k(v_r)$ and $\overline{\mu} = g(v_r)\mu'$ is the apparent dynamic shear modulus formed from the static value $\mu' = \mu$ and dynamic prefactor $g(v_r)$. These dynamic factors are mostly inconsequential during the nucleation phase characterized by $v_r \ll c_s$, with $k \approx g \approx 1$, and full expressions[22] are omitted here for simplicity. The fracture energy of a rate-and-state fault can be defined as the breakdown work $G_c = \int \Delta f \sigma d\delta$ expended near the rupture front in dissipating the strength excess over the steady-state value, $\Delta f = f - f_{ss}(V)$, from its peak value $\Delta f_p$, attained at the front, to zero some distance behind the front over some slip distance $\delta$ relatable to the state-evolution slip distance $L$. Because the peak friction, which is approximately $a\ln v_r$, and the peak friction breakdown, $\Delta f_p \approx b\ln v_r$, are increasing with increasing rupture velocity by the direct and state-evolution effects, respectively, so is the fracture energy $G_c$. Contrary to the energy release rate $G$, which materially depends on $v_r$ only in the seismic range, the dependence of the fracture energy $G_c$ on $v_r$ is persistent over the entire range of rupture speeds, from quasi-static to dynamic. For the ageing law used here, Garagash[22] gives

$$G_c(v_r) = -\text{Li}_2\left(1 - \exp\left(\frac{\Delta f_p(v_r)}{b}\right)\right)b\sigma L \tag{7}$$

in which $\text{Li}_2$ is the dilog function. This expression further simplifies to $G_c \approx (1/2)\left(\frac{\Delta f_p}{b}\right)^2 b\sigma L$ for transients with $\Delta f_p/b \gg 1$. The dependence on rupture speed originates from that of the peak friction breakdown $\Delta f_p(v_r)$, given implicitly by equation (2.17) in ref. 22. Here we use an explicit expression,

$$\frac{\Delta f_p(v_r)}{b} \approx -\frac{2}{3}\text{ProductLog}_{-1}\left(-\frac{3}{2}\left(\frac{1}{\kappa_0 g(v_r)}\frac{v_r}{\overline{v}_0}\right)^{-3/2}\right) \tag{8}$$

in which $\overline{v}_0 = \exp\left(-\frac{\Delta f_0}{b}\right)\frac{\mu'}{b\sigma}V_0$ is the characteristic rupture velocity embodying initial fault conditions (ambient sliding velocity $V_0$ and initial overstress $\Delta f_0 = f_0 - f_{ss}(V_0)$), $\kappa_0 \approx 0.838$ and $\text{ProductLog}_{-1}(x)$ denotes the −1 branch of the Lambert productlog function providing the real-valued solution of $ne^n = m$ for $-1/e < m < 0$. The above explicit inversion of Garagash's equation (2.17) for $\Delta f_p$ is exact for the slip law case and is an excellent approximation for the ageing law case considered here. Indeed, Garagash[22] observes that $\Delta f_p(v_r)$ is essentially indistinguishable for the two laws when plotted in his Fig. 4a.

Reasonable (zeroth-order) approximations, $\Delta f_p \approx b\ln(v_r/\overline{v}_0)$ and $G_c = (1/2)(\ln(v_r/\overline{v}_0))^2 b\sigma L$, allow us to readily glance at a simplified (logarithm-based) relation of the peak friction breakdown and the associated fracture energy on the rupture speed across the entire range of slip transients, from slow to fast.

The crack energy balance $G = G_c$ can be further recast as $K = K_c$, in which the static SIF $K$ matches the effective fault toughness $K_c = (\sqrt{g}/k)\sqrt{2\mu'G_c}$, with the prefactor $(\sqrt{g}/k)(v_r)$ encapsulating the wave-mediated effects and $G_c(v_r)$ the quasi-static-to-dynamic range of the fracture energy.

The SIF $K = K_{bg} + K_{fs}$ consists of the contributions from the background stress, $K_{bg}$, and the foreshock, $K_{fs}$, which can be approximated by the solutions for a bi-wing crack of half-length $\ell$ loaded by an effective uniformly distributed stress drop $\Delta\tau_{eff}$ and a localized, hypocentral point force $\Delta T$, respectively,

$$K_{bg} = \Delta\tau_{eff}\sqrt{\pi\ell}, \quad K_{fs} = \frac{\Delta T}{\sqrt{\pi\ell}}. \tag{9}$$

We note that an edge-crack geometry and corresponding slightly modified expressions for the SIF components can be alternatively used to describe the experimental ruptures, which start proximally to one of the experimental fault ends. However, such a modification is not expected to greatly improve the prediction of an already much simplified model. To close-form the EoM, we now inform the SIFs with expressions for the effective stress drop $\Delta\tau_{eff}$ and hypocentral force $\Delta T$. The hypocentral (foreshock) force $\Delta T$, by a dimensional argument, is

$$\Delta T = C\mu'\delta_a, \tag{10}$$

in which $\delta_a$ is the foreshock 'asperity' slip and $C \approx 0.3$ is a numerical coefficient, which will be used as a fitting parameter. The effective (uniform) stress drop $\Delta\tau_{eff}$ is the energetically equivalent measure of the actual, spatially varying stress drop along the crack defined as,

$$\Delta\tau(V) = (f_0 - f_{ss}(V))\sigma, \tag{11}$$

the drop from the initial stress $f_0\sigma$ to the 'residual' stress $f_{ss}(V)\sigma$ set by the frictional steady-state

$$f_{ss}(V) = (f_0 - \Delta f_0) + (a - b)\ln\left(\frac{V}{V_0}\right), \tag{12}$$

as parametrized here in terms of the fault initial state: $f_0$, $V_0$ and overstress $\Delta f_0 = f_0 - f_{ss}(V_0)$. The spatial distribution of sliding velocity $V = V(x)$ along the crack then defines the spatial distribution of the stress drop. The energetically equivalent $\Delta\tau_{eff}$ can be found from matching its SIF contribution (that is, $K_{bg}$) to that of $\Delta\tau(V(x))$, that is,

$$K_{bg} = \frac{\sqrt{\pi\ell}}{\pi}\int_{-\ell}^{\ell}\frac{\Delta\tau(V(x))}{\sqrt{\ell^2 - x^2}}dx = \Delta\tau_{eff}\sqrt{\pi\ell}. \tag{13}$$

Alternatively, we rewrite

$$\Delta\tau_{eff} = \Delta\tau(V_{eff}) = \frac{1}{\pi}\int_{-\ell}^{\ell}\frac{\Delta\tau(V(x))}{\sqrt{\ell^2 - x^2}}dx, \tag{14}$$

in which the effective sliding velocity $V_{eff}$ is implicitly defined by the second equality when the sliding velocity $V(x)$ is known. Approximating the latter from the solution for the slip on the propagating crack loaded by $\Delta\tau_{eff}$ and $\Delta T$ (Section 3(b) of ref. 22) and solving for $V_{eff}$ ultimately results in

$$V_{eff} = \frac{4K_c}{(g/k)\mu'\sqrt{\pi\ell}}v_r. \tag{15}$$

Finally, the EoM

$$\Delta\tau(V_{\mathrm{eff}})\sqrt{\pi\ell} + \frac{\Delta T}{\sqrt{\pi\ell}} = K_{\mathrm{c}}(v_{\mathrm{r}}) \qquad (16)$$

together with the above expressions for $K_{\mathrm{c}}(v_{\mathrm{r}})$ and $V_{\mathrm{eff}}(v_{\mathrm{r}}, \ell)$ can be solved for $v_{\mathrm{r}}$ as a function of crack length $\ell$. The relation between these two parameters can be integrated to solve for time. Evolution of the fracture toughness, fracture energy and sliding velocity also result.

In the main text, we simplify equation (16) by introducing $K_{\mathrm{bg}} = \Delta\tau(V_{\mathrm{eff}})\sqrt{\pi\ell}$. $K_{\mathrm{bg}}$ is increasing with crack half-length $\ell$ and scales with the rate-dependent transient stress drop $\Delta\tau$—a function of the initial fault overstress parameter $\Delta f_0$ defined as the distance between the background stress ratio $f_0$ from the steady-state friction value $f_{\mathrm{ss}}(V_0)$ at the ambient sliding velocity $V_0$—and transient rupture velocity $v_{\mathrm{r}}$. In our experiments, the fault is probably neutrally or mildly understressed ($\Delta f_0 \gtreqless 0$; Fig. 3c and Extended Data Fig. 12). Furthermore, we introduce $K_{\mathrm{fs}} = \Delta T / \sqrt{\pi\ell}$, which denotes the SIF contribution from the foreshock. It decreases with the crack half-length $\ell$ and scales with a localized hypocentral Coulomb force of magnitude $\Delta T \propto \mu'\delta_{\mathrm{a}}$, generated by the impulse associated with the foreshock slip, $\delta_{\mathrm{a}}$, that initiates the nucleation process.

We note that the classical nucleation crack half-length $\ell_\infty$ (ref. 20) (equation (5)) emerges from the EoM in a formal asymptotic limit of diverging, but not yet seismic, sliding velocity and rupture speed. Indeed, in this case, the SIF, dominated by the stress drop contribution, $K \approx K_{\mathrm{bg}} \approx (b-a)\sigma\sqrt{\pi\ell} \times \ln(V_{\mathrm{eff}}/V_0)$, balances the asymptotic expression for the fault toughness $K_{\mathrm{c}} \approx \sqrt{\mu'b\sigma L} \times \ln(v_{\mathrm{r}}/\bar{v}_0)$. Given the proportionality of the sliding velocity and rupture speed (equation (15)), the two logarithmically diverging terms in the above are identical to the leading order, resulting in $(b-a)\sigma\sqrt{\pi\ell} \approx \sqrt{\mu'b\sigma L}$, with the solution $\ell = \ell_\infty$. Conditions under which the theoretical nucleation length $\ell_\infty$ describes the actual nucleation process depend on the validity of the assumption of an unbounded sliding velocity at nucleation, which allows for the neglecting of finite contributions of the foreshock $\Delta T$ and understress/overstress $\Delta f_0$ to the SIF, as well as the neglecting of the difference in the scaling factors under the logarithms in expressions for $K_{\mathrm{bg}}$ and $K_{\mathrm{c}}$ in the above asymptotic analysis. In reality, these various contributions would have to be evaluated and compared at a finite rupture speed and sliding velocity of the nucleation process (a fraction of seismic values) to validate the applicability of the asymptotic solution.

Illustrative examples of EoM solution for the sliding velocity histories resembling the experimentally observed, foreshock-mediated rupture nucleation regimes are shown in Extended Data Fig. 6 and Fig. 3b using the parametrization explained in the following. See Extended Data Table 2 for a tabulation of the main variables.

**EoM solution in parametric space.** We use a suite of several simulations of the EoM to investigate the dependence of the foreshock-mediated nucleation process: foreshock, transient slow slip, acceleration to mainshock, as encapsulated by the nucleation time $\Delta t_c$, nucleation length $\ell_c$ and the transient minimum sliding velocity $V_{\mathrm{eff,min}}$, on the foreshock force $\Delta T$, initial fault conditions ($\Delta f_0$ and $V_0$) and a set of fault elasto-frictional constitutive parameters. We use a normalized EoM formulation that allows for the reduction of the parametric space to a non-dimensional foreshock force, $\frac{\Delta T}{\mu' L}$, initial overstress, $\frac{\Delta f_0}{b}$, ambient sliding velocity, $\frac{V_0}{(b\sigma/\mu')c_{\mathrm{s}}}$, and a single constitute parameter, the ratio of the direct to state coefficients, $a/b$. We present results for the PMMA-like value $a/b \approx 0.55$ (ref. 56) while also performing a limited suite of solutions for the rock-like value $a/b \approx 0.75$ (ref. 35) to further validate the emerging correlations discussed below.

**Nucleation length, nucleation time, transient minimum velocity and inversion of fault-state-evolution distance.** The EoM suite of solutions for the normalized nucleation length versus nucleation

time is shown in Extended Data Fig. 8 for a wide variety of the non-dimensional loading parameters foreshock force $\frac{\Delta T}{\mu' L}$ (implicit), initial overstress $\frac{\Delta f_0}{b}$ and nondimensional ambient sliding velocity $\frac{V_0}{(b\sigma/\mu')c_{\mathrm{s}}}$. Length and time to nucleation decrease with both increasing foreshock impulse and increasing fault initial overstress. The theoretical nucleation length $\ell_\infty$ (refs. 20,57) serves as the upper bound, which is approached asymptotically on understressed ($\Delta f_0 < 0$) faults when the foreshock force decreases to the $\Delta f_0$-dependent threshold value for the transient arrest (and as time to nucleation diverges).

Performed EoM simulations show a collapse of time-to-nucleation $\Delta t_c$ versus minimum effective sliding velocity, $V_{\mathrm{eff,min}}$, for a wide variety of the non-dimensional loading parameters, $\frac{\Delta T}{\mu' L}$ and $\frac{\Delta f_0}{b}$, and ambient sliding velocity $\frac{V_0}{(b\sigma/\mu')c_{\mathrm{s}}}$ (Extended Data Fig. 9a). This collapse manifests two asymptotic behaviours in which $\Delta t_c \approx L/V_0$ is a constant for transients with very small minimum sliding velocity, comparable with the initial fault value, that is, when $V_{\mathrm{eff,min}} \approx V_0$, and $\Delta t_c \approx (L/V_{\mathrm{eff,min}})$ $\ln(V_{\mathrm{eff,min}}/V_0)$ when $V_{\mathrm{eff,min}} \gg V_0$. Essentially, to the first order, $\Delta t_c$ is inverse in $V_{\mathrm{eff,min}}$ with the low-velocity cut-off at about $V_0$. Small departure from the collapsed curve on Extended Data Fig. 9a corresponds to the solutions with appreciable understress $\Delta f_0 < 0$. This second-order behaviour is more apparent when we remove the dominant (inverse in $V_{\mathrm{eff,min}}$) behaviour when plotting the product $\Delta t_c V_{\mathrm{eff,min}}$ in Extended Data Fig. 9b.

The overall relation between $\Delta t_c$ and $V_{\mathrm{eff,min}}$ (Extended Data Fig. 9a) can be well modelled by

$$\Delta t_c = \frac{1}{\pi(1 - a/b)} \frac{L}{V_{\mathrm{eff,min}}} \times \mathcal{D}\left(\frac{V_{\mathrm{eff,min}}}{V_0}\right) \qquad (17)$$

with the transition function

$$\mathcal{D}(\mathcal{V}) = \left( (C_0\mathcal{V})^{-\alpha} + \left(C_1\ln\frac{\mathcal{V}}{\mathcal{V}_1}\right)^{-\alpha} \right)^{-1/\alpha} \qquad (18)$$

between the small-velocity, $\mathcal{D} \approx C_0\mathcal{V}$, and large-velocity, $\mathcal{D} \approx C_1\ln\left(\frac{\mathcal{V}}{\mathcal{V}_1}\right)$, asymptotes, and fitting constants $C_0$, $C_1$, $\mathcal{V}_1$ and $\alpha$ found as 0.64174, 1.33245, 0.01332 and 2.15680, respectively. The asymptotes follow by considering the time required to achieve the maximum stable crack length, $\ell_\infty$, at the minimum rupture velocity, $v_{\mathrm{r,min}}$, itself dependent on $V_{\mathrm{eff,min}}$ by equation (15).

To perform the fit of the EoM empirical relation of the time-to-nucleation $\Delta t_c$ and the average minimum sliding velocity $V_{\mathrm{eff,min}}$ (equation (17)) to the correlation of experimental events $\Delta t_c$ with the minimum value of the sliding velocity $V_{\mathrm{min}}$ measured at a fixed point on the fault, we need to relate the pointwise value $V_{\mathrm{min}}$ to the crack-average one $V_{\mathrm{eff,min}}$. To accomplish this, we use the EoM approximate solution for the sliding velocity distribution[22] $V(x) = (V_{\mathrm{eff}}/2)(1 - (x/\ell)^2)^{-1/2}$ and model the sliding velocity measured at a fixed location on the fault proximal to the foreshock as the hypocentral value of the EoM distribution. In other words, we use the $V_{\mathrm{eff,min}}/2$ of the model as the proxy for the experimental $V_{\mathrm{min}}$.

Taking $a = 0.016$ and $b = 0.029$ for the PMMA laboratory fault[56], equation (17) was fit to the laboratory data, yielding a fault ambient sliding velocity $V_0 = 1.35 \times 10^{-6}$ m s$^{-1}$ and the state-evolution slip distance of the interface $L = 0.19$ μm comparable with approximately 0.4 μm inferred from velocity-stepping experiments on a PMMA fault[56].

Similarly, in Fig. 4d, equation (17) was used to prepare possible fits for the natural earthquake data assuming granite friction[35] with $a = 0.014$ and $b = 0.019$ and natural fault ambient sliding velocity $V_0 = 10^{-12}$ m s$^{-1}$. We note that the transient (minimum) sliding velocity values inferred for the natural earthquakes in our dataset (Fig. 4d) are on the order of or exceed the typical plate convergence rate, that is, $V_{\mathrm{min}} \gtrsim 10^{-9}$ m s$^{-1}$. Thus, any suitable choice of the ambient sliding velocity assuming an initially 'locked' fault, that is, $V_0 \ll 10^{-9}$ m s$^{-1}$, will not have a substantial effect on the EoM fit for the natural events.

**Transient minimum velocity versus foreshock force versus foreshock slip.** The EoM empirically predicts a collapse of $V_{eff,min}$ and $\Delta T$ only in the case of large overstress (Extended Data Fig. 12). In those cases, we find $\frac{V_{eff,min}}{V_0} \approx 10 \times e^{\frac{\Delta T}{0.5l\mu'L}}$, in which $\Delta T = C\mu'\delta_a$. We use this relation together with the fitting constant $C = 0.3$ to produce the upper-bound dashed curves in Fig. 3c for different values of $L$ bracketing the best-fit value of 0.19 μm. Indeed, the EoM does not indicate that there is a one-to-one relationship between $V_{eff,min}$ and the hypocentral force, $\Delta T$. Instead, $V_{eff,min}$ also depends on the overstress. Rather, $V_{eff,min}$ and its experimental proxy $V_{min}$ encapsulate the loading conditions on the fault from both prestress and the foreshock (asperity slip) and cannot be reduced to one or the other.

**EoM modelling of experimental sliding velocity history.** The model parameters used to produce the experiment-reflecting results for transient sliding velocity evolution of Fig. 3b are the same as used above for PMMA with the addition of the shear wave speed taken as 1,345 m s$^{-1}$, the apparent shear modulus taken as 1.24 GPa and the normal stress taken as 0.8333 MPa (the average of the normal stresses given in Fig. 2). The asperity slip $\delta_a$ is converted into an equivalent hypocentral Coulomb force $\Delta T$ using a single asperity equivalent point force, $\Delta T = C\mu'\delta_a$ with $C = 0.3$.

## Data availability

Data are available at https://doi.org/10.5281/zenodo.17185894 (ref. 58). Source data are provided with this paper.

## Code availability

The code is available in the online repository at https://doi.org/10.5281/zenodo.17185894 (ref. 58). All information and details necessary to reproduce the theoretical development are described in Methods.

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

**Acknowledgements** B.F. and F.P. acknowledge support from the European Union (ERC Starting Grant HOPE number 101041966). C. Noël, F. Paglialunga and J. Ambre are thanked for their help in preparing the sample and set-up. X. Cui, J.-P. Ampuero and F. Cappa are thanked for helpful discussions.

**Author contributions** B.F. performed the experimental measurements. B.F. and F.P. performed the experimental data analysis. F.P., M.L. and B.F. designed the experiments. D.G. led the derivation of the theory, with contributions from M.L. and other authors. M.L., D.G. and B.F. developed the numerical codes. D.G. and M.L. produced the numerical results. B.F. and F.P. incorporated the natural earthquakes. All authors contributed to the writing of the paper and overall analysis.

**Competing interests** The authors declare no competing interests.

**Additional information**
**Correspondence and requests for materials** should be addressed to Barnaby Fryer.

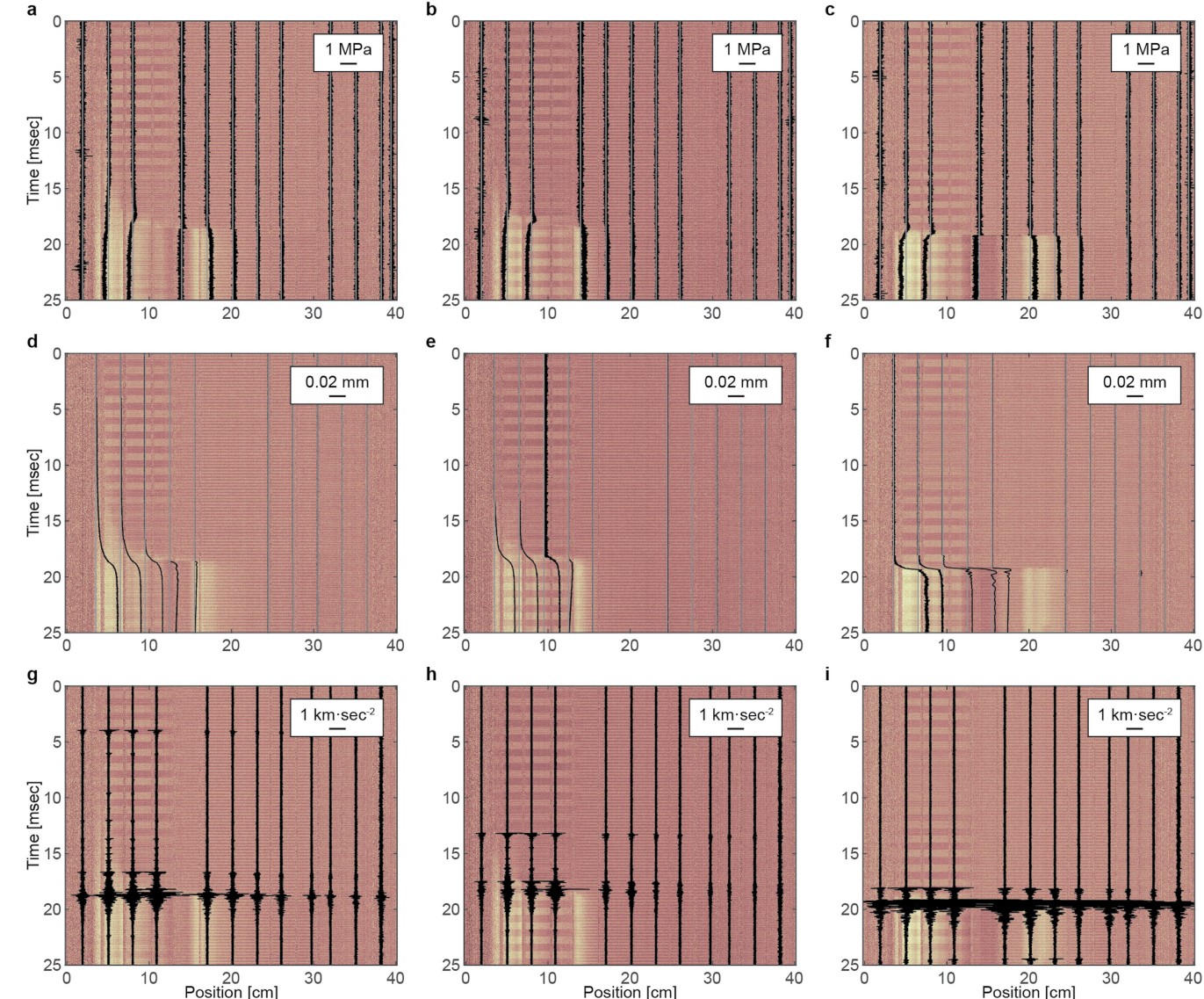

**Extended Data Fig. 1 | Overview of collected data during events. a–i,** The videograms of the events in Fig. 1 overlain with variations in shear stress (**a–c**), slip displacement (**d–f**) and acceleration (**g–i**). In all sub-figures, the reading is zeroed to its position on the experimental fault and deviation from this position corresponds to a change in the measured parameter in accordance with the scale given in the top-right of each videogram.

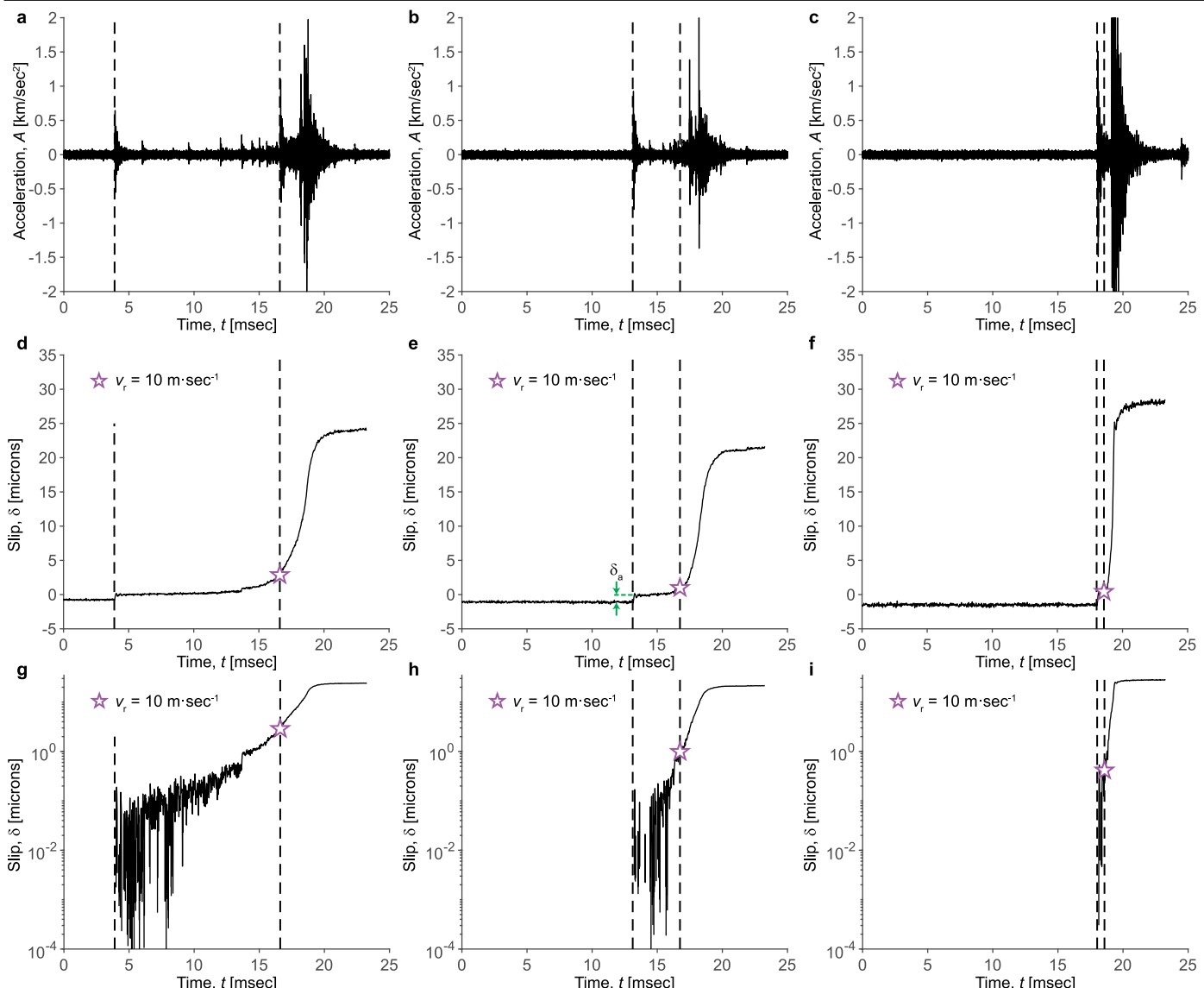

**Extended Data Fig. 2 | Accumulated slip between foreshock and mainshock. a–c**, The accelerations recorded by the three closest accelerometers to the foreshock plotted one on top of the other for the events shown in Fig. 1. **d–f**, The local slip recorded by the closest displacement sensor to the nucleating patch. The moment at which the rupture front passes 10 m s⁻¹ is indicated with a star. The nucleation time, $\Delta t_c$, is the time between the foreshock and the rupture velocity reaching 10 m s⁻¹. Both of these moments are indicated with dashed vertical lines. The slip during the foreshock, $\delta_a$, is taken from the displacement sensors as illustrated. **g–i**, The same as in **d–f** but with slip shown on a log scale to illustrate that the slip is occurring. The slip is assumed to evolve exponentially in time. This assumed evolution is fit and the derivative evaluated at the time of the foreshock, yielding $V_{min}$.

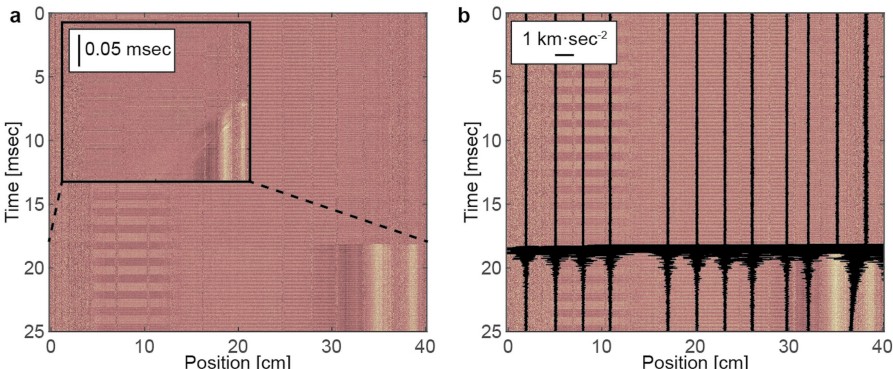

**Extended Data Fig. 3 | An example of an event with no discernible nucleation phase. a**, The videogram of the event with an inset showing a zoom-in on the start of the dynamic event. **b**, The same videogram overlain with the accelerometer data, showing no discernible foreshock before the event.

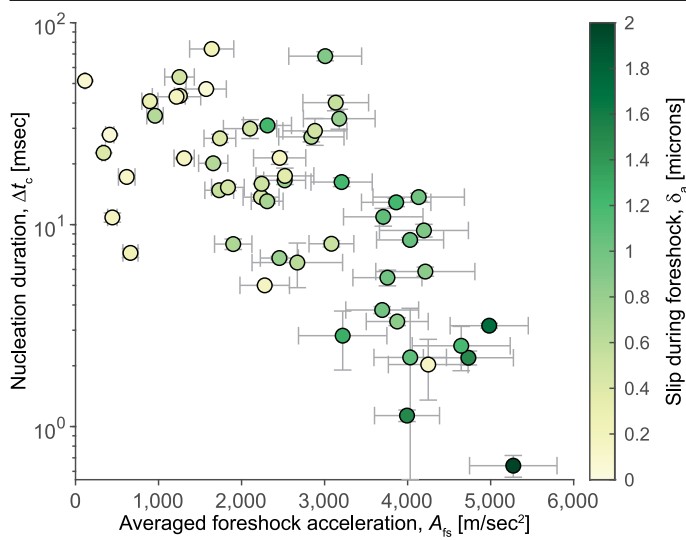

**Extended Data Fig. 4 | Larger foreshocks correlate to shorter nucleation phases.** The nucleation duration versus foreshock size (as measured by the accelerometers). The colour bar shows the slip during the foreshock (as measured by the displacement sensors).

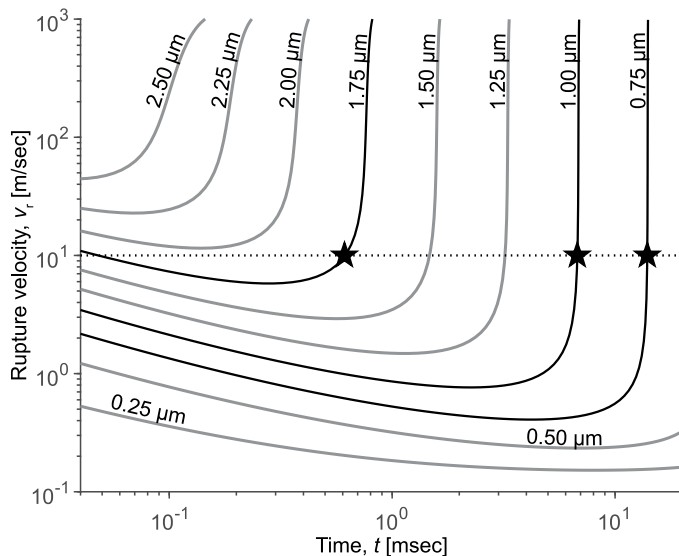

**Extended Data Fig. 5 | The rupture velocity corresponding to the simulations shown in Fig. 3b.** The overstress is $\Delta f_0 = 0$ and the asperity slip, $\delta_a$, is shown on the curve of each simulation. Dark lines correspond to the cases most closely resembling the asperity slip of the experiments shown in Fig. 3a. Stars indicate the nucleation of dynamic rupture.

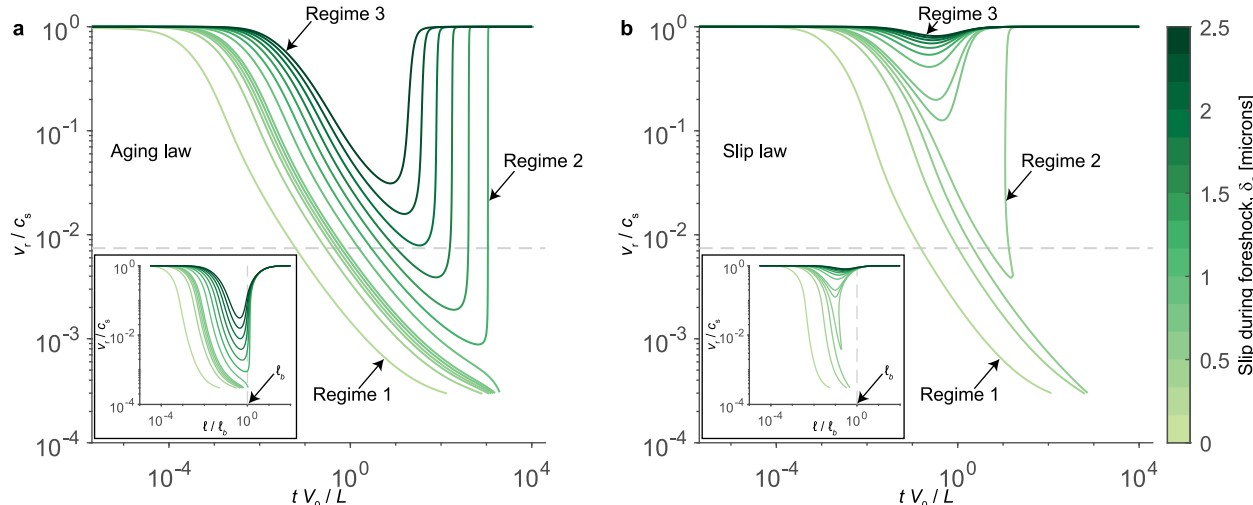

**Extended Data Fig. 6 | EoM simulations illustrating the three nucleation regimes. a,b,** EoM simulations for an understressed case, $\frac{\Delta f_0}{b} = -1$, for the ageing (**a**) and slip (**b**) laws. The asperity slip, $\delta_a$, during the foreshock for each simulation is given by the colour bar. The other parameters are the same as those used in Fig. 3b. The three regimes discussed in the main text are present for both laws. Regime 1: transient deceleration followed by arrest without the nucleation of dynamic rupture. Regime 2: transient deceleration to a minimum sliding velocity, $V_{min}$, followed by accelerating slip and rupture velocity leading to dynamic rupture. Regime 3: near-immediate acceleration into dynamic rupture, without a quasi-static phase. The horizontal dashed lines show $v_r = 10 \text{ m s}^{-1}$ considering $c_s = 1{,}345 \text{ m s}^{-1}$. The insets show crack half-length, with the vertical dashed line indicating $\ell_b$.

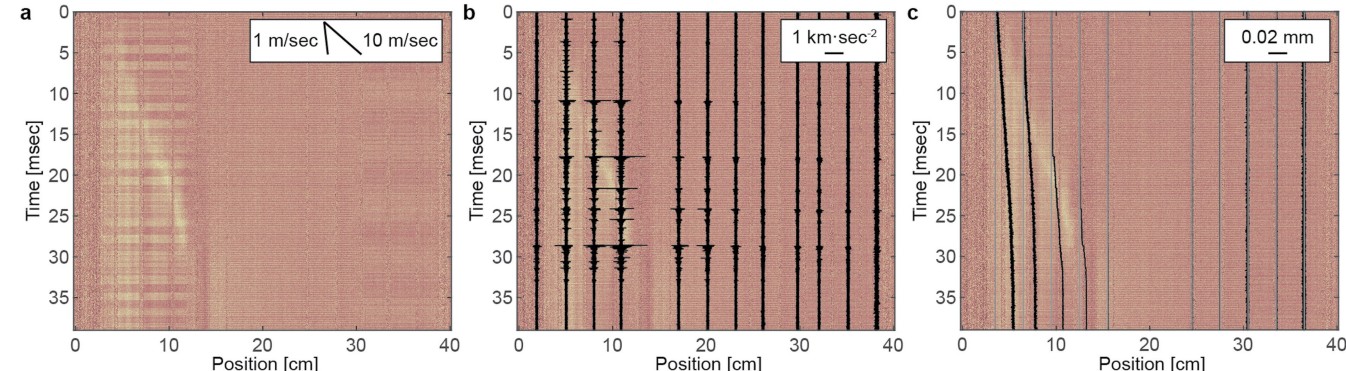

**Extended Data Fig. 7 | Frustrated nucleation.** An example of an event with an extended quasi-static slip phase but without the nucleation of dynamic rupture (maximum rupture velocity approximately 4 m s⁻¹). **a**, The videogram of the event. **b**, The same videogram overlain with the accelerometer data, showing the initiation of the quasi-static slip phase with a foreshock. **c**, The videogram overlain with the local slip displacement data. Note that, although this event can be interpreted to not have had an appropriate hypocentral force and overstress to develop into dynamic rupture as predicted by the EoM, an alternative interpretation could be that the crack tip propagated into an understressed region (barrier), resulting in its arrest.

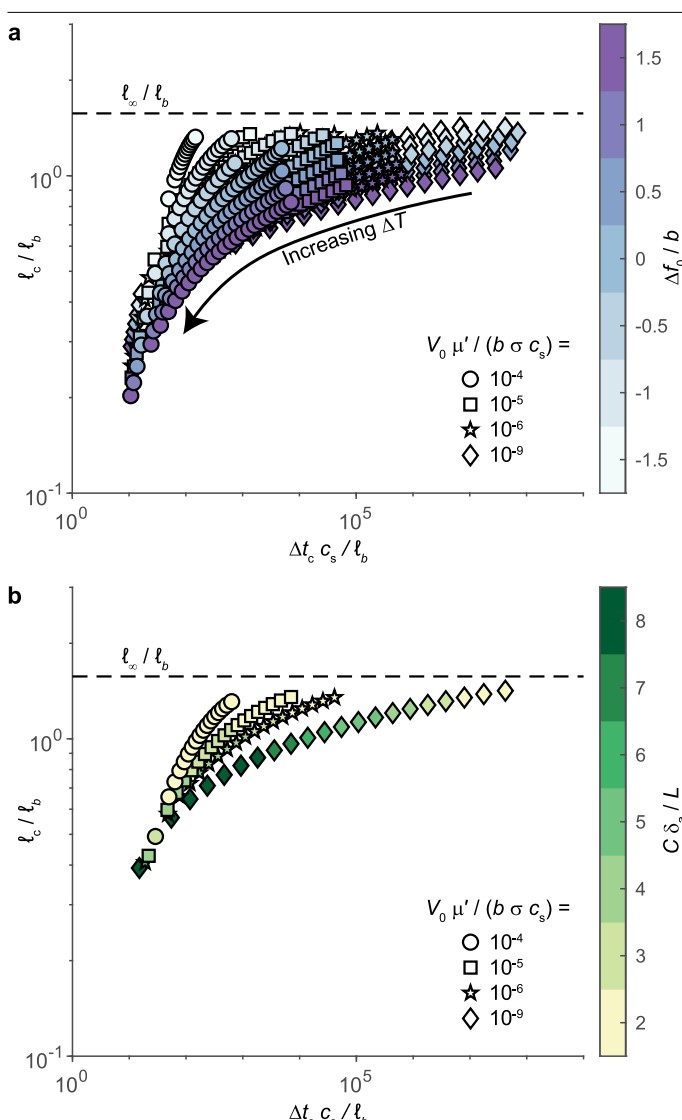

**Extended Data Fig. 8 | Overstress and foreshock impulse reduce nucleation length. a**, EoM solution for the nucleation length $\ell_c$ versus time $\Delta t_c$ for varying foreshock force $\Delta T$ (implicit in this plot); various values of the overstress $\frac{\Delta f_0}{b}$ given by the colour bar and ambient sliding velocity $V_0$ shown by symbols. **b**, The same plot but only showing cases in which $\frac{\Delta f_0}{b} = -1$, with the foreshock size shown by the colour bar. Both overstress and foreshock impulse reduce the nucleation length $\ell_c$ up to an order of magnitude below the theoretical maximum length $\ell_\infty$ indicated by the dashed line.

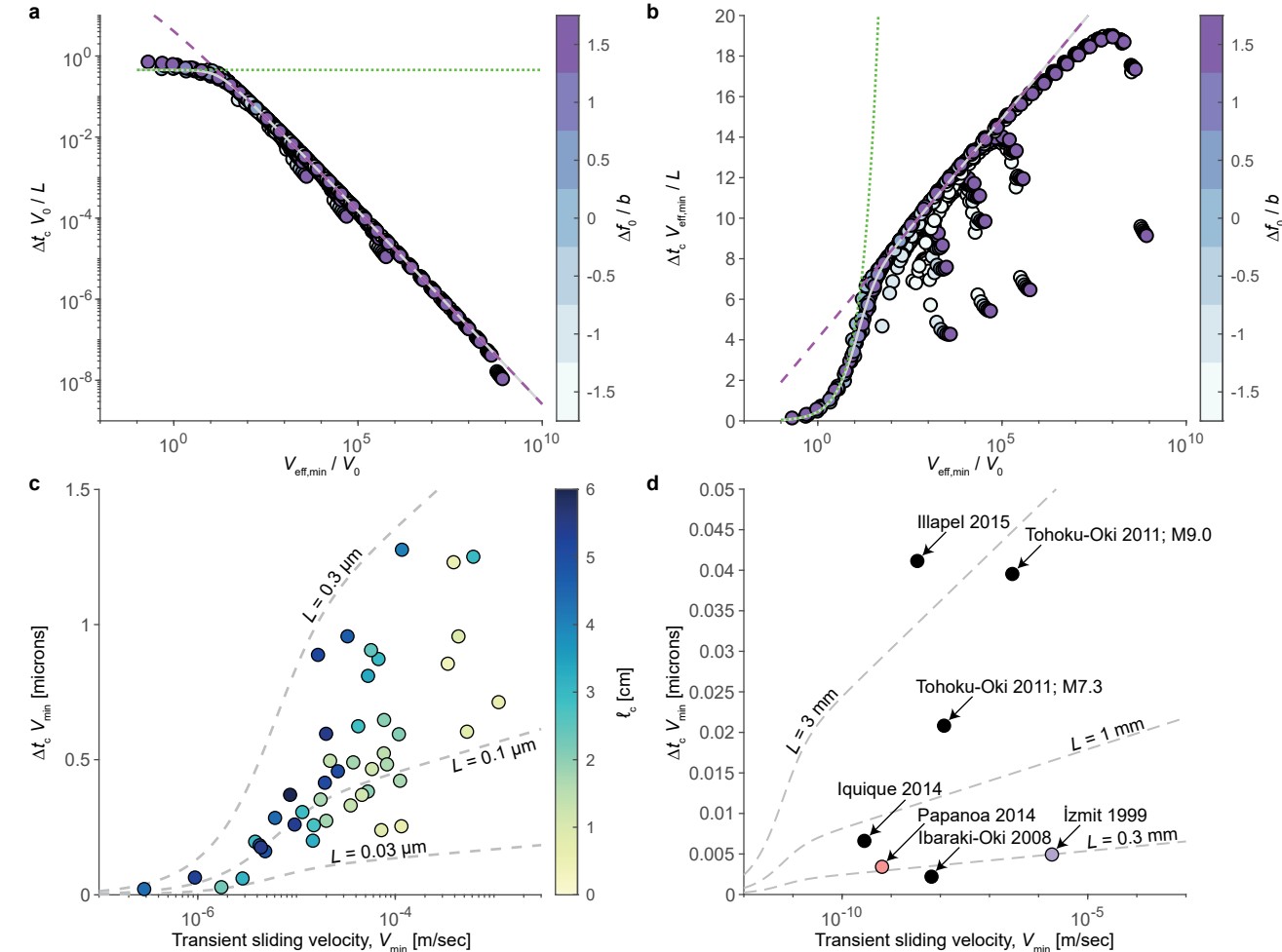

**Extended Data Fig. 9 | $V_{min}$ controls nucleation duration. a**, EoM simulations on which equation (17) is based. The data show a collapse, regardless of overstress, onto the primary $\Delta t_c \approx \frac{L}{V_{eff,min}}$ dependence with the low-velocity cut-off at $V_{eff,min} \approx V_0$. **b**, By removing the primary $\Delta t_c \approx \frac{L}{V_{eff,min}}$ dependence, the second-order trend is visible. In **a** and **b**, the overstress is given by the colour bar. The first-order trend is shown with the dashed magenta line. The second-order trend is shown with the dotted green line. The combined trend is shown with the grey line. **c**, Fig. 4c detrended to show the second-order dependence of $\Delta t_c$ on $V_{min}$. The collapse (for varying $\Delta T$ and $\Delta f_0$) for $L = 0.03$, 0.10 and 0.30 µm are shown with $a/b = 0.55$ and $V_0 = 10^{-6}$ m s$^{-1}$. **d**, The equivalent to **c** for natural earthquakes and $a/b = 0.75$, $V_0 = 10^{-12}$ m s$^{-1}$ and $L = 0.3$, 1.0 and 3.0 mm.

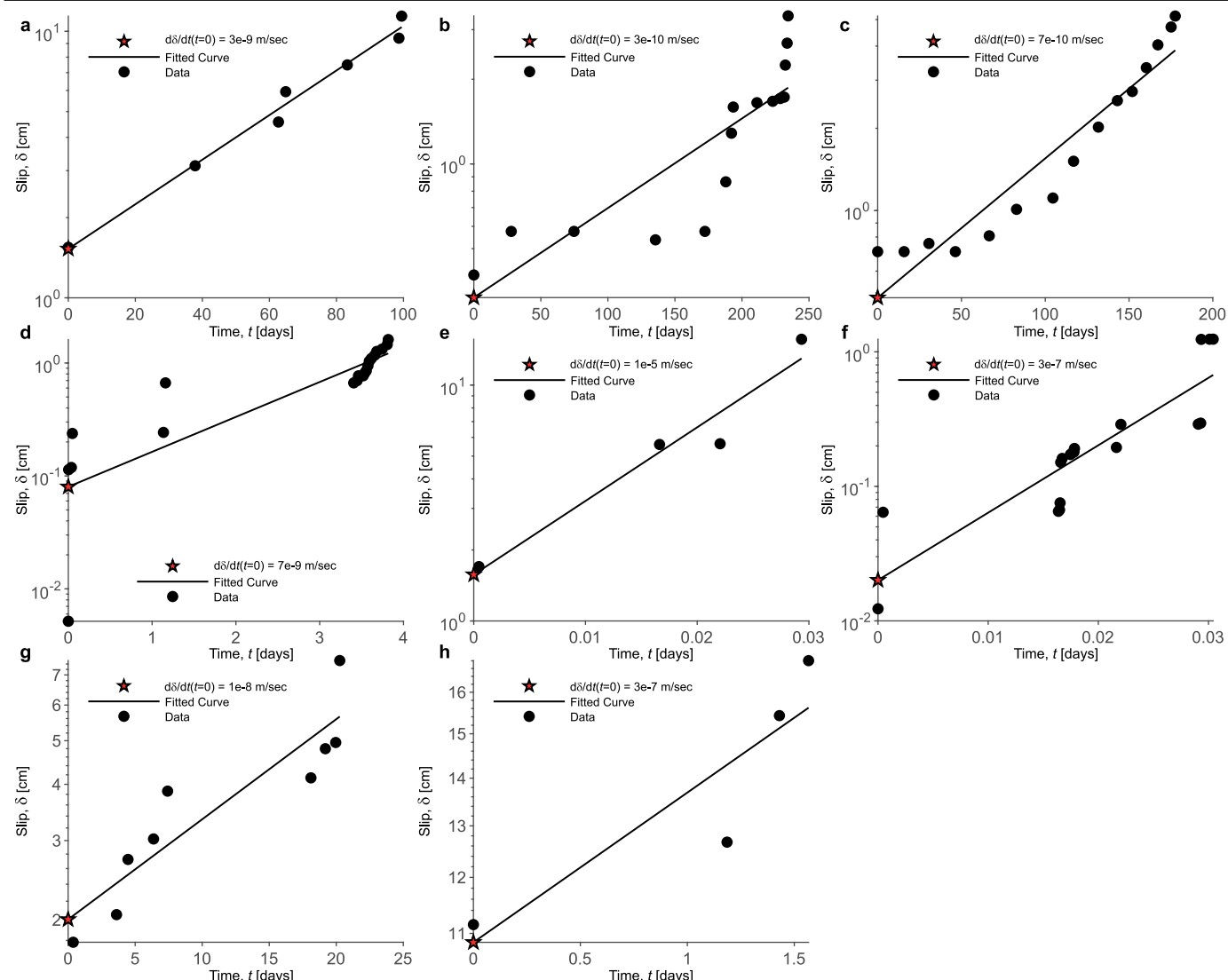

**Extended Data Fig. 10 | The fitting of natural earthquakes. a–h,** Demonstration of how the transient sliding velocities of the Illapel 2015 (**a**, ref. 9), Iquique 2014 (**b**, ref. 4), Papanoa 2014 (**c**, ref. 40), Ibaraki-Oki 2008 (**d**, ref. 10), İzmit 1999 (**e**, ref. 11; **f**, ref. 1), Tohoku-Oki 2011 M7.3 (**g**, ref. 2) and Tohoku-Oki 2011 M9.0 (**h**, ref. 2) earthquakes were estimated. The data (Extended Data Table 1) were fit with an exponential function. The derivative of this exponential function was then evaluated at the start of the sequence, yielding an estimate of transient sliding velocity. The geometric average of **e** and **f** was taken for İzmit 1999, with the two values providing a crude error estimate.

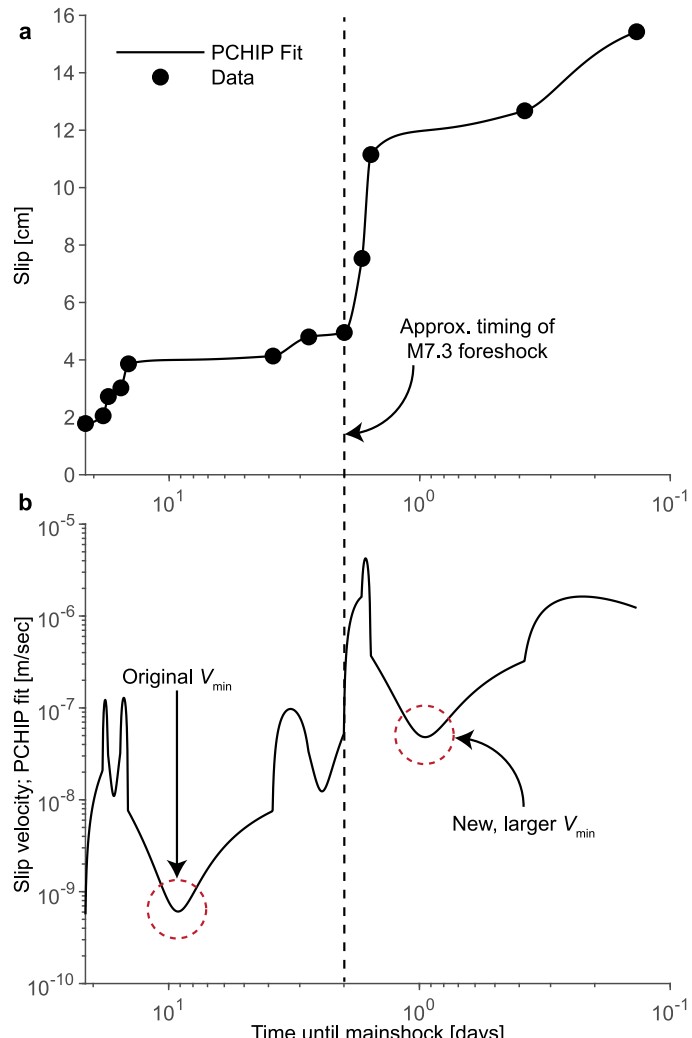

**Extended Data Fig. 11 | Slip development before Tohoku-Oki 2011. a**, The slip data taken from Kato et al.[2]; their Fig. 3Bc. The slip data are fit with a monotonic shape-preserving cubic interpolation (PCHIP). **b**, The derivative of the slip data is taken with respect to time. An initial spike in slip velocity can be seen after the first foreshocks approximately 23 days before the mainshock. The slip velocity then decreases to a minimum before accelerating. Approximately 2 days before the mainshock, the slip velocity accelerates to a peak associated with the M7.3 foreshock. The slip velocity then decreases to a new minimum, larger than the previous minimum, before accelerating again towards the M9.0 mainshock.

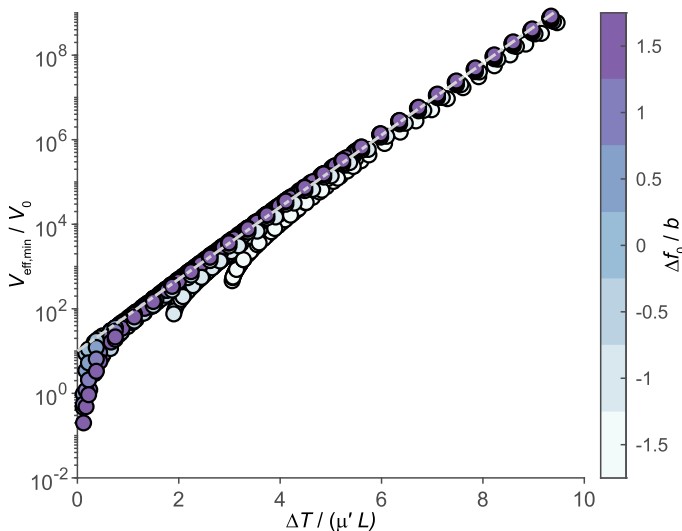

**Extended Data Fig. 12 | EoM simulations showing the relationship between foreshock size and initial minimum sliding velocity.** The EoM only predicts collapse for large overstress cases. $V_{\text{eff,min}}$ (and $V_{\text{min}}$ by association) depends not only on foreshock size but also overstress. The overstress is given by the colour bar. The collapsed portion of the data is fit with $\frac{V_{\text{eff,min}}}{V_0} \approx 10 \times e^{\frac{\Delta T}{0.51\mu' L}}$; this trend is shown with a dashed grey line.

**Extended Data Table 1 | Laboratory and natural earthquake data from the literature**

| Earthquake | Lab-quake | Year | $\Delta t_c$ [sec] | Quasi-static half length [m] | Quasi-static slip [m] | $V_{min}$ [m/sec] |
|---|---|---|---|---|---|---|
| Tohoku-Oki M7.3[*] | - | 2011 | 1.75e6 | - | 0.075 | 1.2e-8 |
| Tohoku-Oki M9.0[*] | - | 2011 | 1.35e5 | 37500 | 0.2 | 2.9e-7 |
| Illapel[†] | - | 2015 | 1.21e7 | 25000 | 0.113 | 3.4e-9 |
| Iquique[‡] | - | 2014 | 2.33e7 | 55000 | 0.09 | 2.8e-10 |
| Papanoa[§] | - | 2014 | 5.26e6 | 125000 | 0.05 | 6.5e-10 |
| Ibaraki-Oki[‖] | - | 2008 | 3.33e5 | 22300 | 0.017 | 6.6e-9 |
| İzmit[¶] | - | 1999 | 2.64e3 | 150 | 0.0124 | 2.7e-7 |
| İzmit[#] | - | 1999 | 2.64e3 | 150 | 0.160 | 1.3e-5 |
| Yanbgi[☆] | - | 2021 | 3.24e3 | 6750 | 0.2 | 6.17e-5[‡‡‡] |
| Valparaíso[**] | - | 2017 | 3.46e5 | 45000 | 0.09 | 2.6e-7[‡‡‡] |
| Kumamoto[††] | - | 2016 | 1.00e5 | 10000 | 0.255 | - |
| Silivri[‡‡] | - | 2019 | 3.72e5 | 3500 | - | - |
| Duvernay[§§] | - | 2016 | 1.90e4 | 1000 | - | - |
| - | Ohnaka and Shen, 1999[‖‖] | - | 9.00e0 | 0.12 | 1.5e-5 | 3.6e-7 |
| - | McLaskey, 2019[¶¶] | - | 4.50e0 | 0.38 | 5.8e-5 | 1e-7 |
| - | McLaskey, 2019[##] | - | 4.85e-2 | 0.94 | - | 1e-5 |
| - | Xu et al., 2023[☆☆] | - | 1.22e-1 | 0.635 | - | - |
| - | Latour et al., 2013[***] | - | 1.2e-2 | 0.027 | - | - |
| - | Yamashita et al., 2021[†††] | - | 3.3e-1 | 0.55 | - | - |

Details on data collection are provided with footnotes

[*]$\Delta t_c$ from Fig. 3. Quasi-static length from Fig. 2. $V_{min}$ from Fig. 3Bc (ref. 2).

[†]Quasi-static length from Fig. 3b. Slip and $\Delta t_c$ from page 3,918. $V_{min}$ from Fig. 4c evaluated at −140 days (ref. 9).

[‡]$\Delta t_c$ from abstract. Slip from Fig. 3b. Quasi-static length from Fig. 4. $V_{min}$ from Fig. 3b (ref. 4).

[§]Quasi-static length from Fig. 2. $V_{min}$ estimated from Fig. 2c assuming mainshock at slip of 5 cm and constant patch size (ref. 40).

[‖]Quasi-static length and $\Delta t_c$ from Fig. 5a. $V_{min}$ from Fig. 5d (ref. 10).

[¶]Slip and $V_{min}$ from Supplementary Table 1 assuming shear modulus of 30 GPa. Patch size from page 879 (ref. 1).

[#]Slip and $V_{min}$ from Supplementary Table 2 (ref. 11). Patch size assumed the same as [¶].

[☆]Quasi-static length and slip from Fig. 10b. $\Delta t_c$ from Stage III in Fig. 10a (ref. 13). $V_{min}$ taken as the average slip velocity (overestimate).

[**]Quasi-static length from Fig. 5. $\Delta t_c$ from text page 10,293. Slip from Fig. 3 (ref. 6). $V_{min}$ taken as the average slip velocity (overestimate).

[††]Quasi-static length and $\Delta t_c$ from Fig. 3. Slip from page 8,951 (ref. 5).

[‡‡]Quasi-static length from abstract. $\Delta t_c$ from Fig. 5 (ref. 42).

[§§]$\Delta t_c$ from Fig. 5d. Quasi-static length from Fig. 3 (ref. 41).

[‖‖]All data from Figs. 20 and 21 (ref. 39).

[¶¶]$\Delta t_c$ and quasi-static length from Fig. 2c. Slip and $V_{min}$ from Fig. 2b (ref. 14).

[##]Quasi-static length from Fig. 14a. $\Delta t_c$, slip and $V_{min}$ from Fig. 14b (ref. 14).

[☆☆]Quasi-static length and $\Delta t_c$ from Fig. 2d. Figure 2c considered dynamic (ref. 44).

[***]Data from Fig. 3 (ref. 27).

[†††]Quasi-static length and $\Delta t_c$ from Fig. 2a (ref. 43).

[‡‡‡]The time series for these data is not available. Instead, an average slip velocity is used for $V_{min}$, resulting in probable overestimation.

**Extended Data Table 2 | Tabulation of the principal variables used in this study**

| Variable | Name |
|---|---|
| $a$ | Direct effect |
| $A_{fs}$ | Averaged foreshock acceleration |
| $b$ | Evolution effect |
| $c_s$ | Shear wave speed |
| $\delta$ | Slip |
| $\delta_a$ | Foreshock slip |
| $\Delta f_0$ | Initial overstress |
| $\Delta f_p$ | Maximum departure of friction from steady state |
| $\Delta t_c$ | Nucleation duration |
| $\Delta T$ | Hypocentral Coulomb force |
| $\Delta \tau$ | Shear-stress drop |
| $\Delta \tau_{eff}$ | Effective (uniform) shear-stress drop |
| $f_0$ | Initial stress ratio |
| $f_{ss}$ | Steady-state level of friction |
| $g$ | Dynamic prefactor |
| $G$ | Energy release rate |
| $G_c$ | Fracture energy |
| $k$ | Wave-mediated dynamic prefactor |
| $K$ | Stress intensity factor |
| $K_{bg}$ | Background stress contribution to stress intensity factor |
| $K_c$ | Fault toughness |
| $K_{fs}$ | Foreshock contribution to stress intensity factor |
| $L$ | Characteristic slip distance |
| $\ell$ | Crack half length |
| $\ell_b$ | Elasto-frictional length scale |
| $\ell_c$ | Nucleation half length |
| $\ell_\infty$ | Maximum nucleation half length (aging law) |
| $\mu'$ | Apparent shear modulus |
| $\sigma$ | Normal stress |
| $t$ | Time |
| $\tau$ | Shear stress |
| $\tau_p$ | Peak shear stress |
| $\bar{V}_0$ | Characteristic rupture velocity scale |
| $v_r$ | Rupture velocity |
| $V$ | Fault sliding velocity |
| $V_0$ | Ambient fault sliding velocity |
| $V_{eff,min}$ | Effective transient sliding velocity (model) |
| $V_{min}$ | Transient sliding velocity |
| $W$ | Slip-weakening rate |

The list is non-extensive.