## [Peer Review File · Nature]

Foreshock-induced slip transients set mainshock nucleation timing

Corresponding Author: Dr Barnaby Fryer

Version 0:

Reviewer comments:

Referee #1

(Remarks to the Author)

Please refer to the PDF file attached.

Referee #2

(Remarks to the Author)

I co-reviewed this manuscript with one of the reviewers who provided the listed reports.

Referee #3

(Remarks to the Author)

Please see attached pdf.

Version 1:

Reviewer comments:

Referee #1

(Remarks on code availability)

(Remarks to the Author)

Please see the PDF file attached.

Referee #2

(Remarks on code availability)

(Remarks to the Author)

I co-reviewed this manuscript with one of the reviewers who provided the listed reports.

Referee #3

(Remarks on code availability)

(Remarks to the Author)
See attachment

Reviewers #1 and 2

10. *In this manuscript, the authors investigated how stick slip events initiate and grow for the dynamic rupture during PMMA friction experiments, aiming to understand the nucleation process of large earthquakes. They considered that an impulse generated by a foreshock initiates the nucleation slip and modulate the time required for a fault to evolve into dynamic rupture. The principal findings are as follows:*
- *An impulsive perturbation generated by a foreshock directly influences the subsequent transient sliding velocity, which controls the nucleation duration.*
 - *This behavior is supported by the rate-dependent equation of motion (EoM) framework formulated by Garagash, indicating that both overstress and foreshock impulse jointly shape rupture evolution.*

These results suggest that impulsive perturbations at the onset of quasi-static nucleation, typically absent from canonical preslip models, may serve as a first-order control on nucleation duration in natural earthquakes. Accordingly, the study may expand our current understanding of the realistic nucleation process of both seismic ruptures and more general frictional failure processes.

This manuscript consists of comprehensive views drawn from experiments, theory, and natural observations. The comparison between experimental/theoretical insights and seismological observations is leveraged to propose scaling relationships between foreshock size, nucleation duration, and transient sliding velocity.

The argument is quite impactful for understanding earthquake nucleation, and it is innovative to address this problem through combined experimental and theoretical approaches. I agree with the interpretations of the experimental data by theoretical analysis of rupture nucleation based on the EoM framework. However, I have several concerns regarding the assumptions of the modeling, analysis of the experimental results, and the interpretation of natural earthquakes. Therefore, I think significant revisions could be needed if this manuscript is considered to be accepted for publication. I hope that the comments listed below will be useful in further improving the manuscript.

Response: We thank the reviewers for their thorough evaluation of our manuscript and for their constructive and insightful comments. Their careful reading and detailed analysis of our experimental approach, theoretical framework, and compar-

ison with natural earthquakes have been invaluable. The points they raise have helped us reassess several aspects of our work, refine our interpretations, and improve the clarity and robustness of the manuscript. We have carefully addressed each of their comments below and believe that the revisions have substantially strengthened the paper.

Major Comments

11. • *Logical collapse of the nucleation mechanism*

In the proposed scenario of the nucleation model proposed here, the stimulation by the stress impulse seems a key point, which the authors thought to be generated by the foreshock. This stimulation initiates the nucleation of the rupture and control the nucleation duration. However, if this stimulation only comes from a foreshock, one cannot explain the cases when no foreshock was observed at the beginning of rupture nucleation. In addition, that foreshock itself should have its own nucleation process which should be triggered by another small foreshock following the authors' scenario. which apparently requires another nucleation mechanism and the proposed nucleation mechanism is considered as a case under very limited conditions. Then, the audience would wonder how to apply the proposed method to apply the natural earthquakes or other frictional phenomena. I think it quite important to clarify the condition where the proposed model can apply.

Response: We thank the reviewers for raising this important point. While our study focuses on cases in which nucleation is initiated by a foreshock, because this is the behavior consistently observed in our experimental dataset, this is not a requirement of the equation of motion (EoM), nor do we claim that nucleation is systematically triggered by a foreshock. We clarify this more explicitly in the revised manuscript (please see lines 137-146; 343-345, 479-489). Previous laboratory studies show that nucleation can also grow quasi-statically without any observable foreshock. Indeed, with sufficient overstress, any perturbation (be it from a foreshock or from loading) will trigger nucleation. Our present work purposely isolates the subset of events in which a foreshock-generated impulse modifies the ensuing nucleation dynamics, because this regime exhibits a systematic and reproducible scaling that we consider important to understand.

The EoM framework itself, however, remains fully applicable to more classical nucleation sequences. In such cases, a steadily increasing background loading (see Garagash 2021[1], p. 14) leads to gradual increases in $\Delta f_0/b$ and to quasi-static crack growth. We do not explicitly account for this loading in our analysis because (i) the background loading rate is identical across all experiments, (ii) the quasi-static nucleation phases in our dataset are short compared to the imposed loading timescale, and (iii) our aim here is to isolate the effects of foreshock-generated impulses on subsequent nucleation. Quantitatively, the characteristic timescale

$t_0 = L/V_0 \approx 0.1407$ s exceeds all observed quasi-static nucleation durations, meaning that the background evolution of V_0 and Δf_0 can be neglected. On natural faults, however, tectonic loading operates over much longer timescales and can significantly increase $\Delta f_0/b$, thereby enhancing K_{bg} and driving crack growth even in the absence of foreshocks. Exploring this classical nucleation regime is beyond the scope of the present study, and in natural settings it is also more difficult to identify because reliably measuring quasi-static nucleation duration without accompanying foreshocks remains challenging and is seldom documented.

Similarly, the foreshock that initiates the mainshock in our experiments does not require a secondary, smaller foreshock for its own initiation. As described in the Methods, the coexistence of foreshocks and macroscopic nucleation arises from substantial stress heterogeneity at scales ranging from asperities to centimetres. Under such heterogeneous conditions, a foreshock may nucleate either from localized overstress or from background loading, depending on the relative timescales. When the characteristic timescale t_0 is short relative to the foreshock’s quasi-static nucleation interval, background loading can meaningfully contribute to its initiation. In these cases, no additional foreshock is required to trigger the foreshock that ultimately initiates the mainshock.

We have added this discussion to the section “Foreshocks and nucleation length scaling”. Please see lines 137-146; 330-355, 479-489.

12. • *Uncertain evaluation of quasi-static length on natural earthquakes*

The authors listed the quasi-static length, quasi-static slip, and transient sliding velocity V_{min} on the 10 natural earthquakes from the published literature as listed in Supplementary table S1. The coherence between these parameters supports the argument of the correlation between the nucleation duration and quasi-static length, and the transient sliding velocity and nucleation duration observed in the laboratory experiment. However, the quasi-static length is generally difficult to estimate, even in the order sense, and it looks like different measurements are mixed in the table. At least, the authors should describe how these values are estimated and evaluate their uncertainties in the supplementary.

Response: We appreciate and are grateful of the reviewer’s careful examination of the natural-earthquake dataset, and we agree that estimating quasi-static nucleation length from seismological observations is inherently challenging. The values reported in Methods Table S1 indeed come from different types of measurements (e.g., repeating earthquakes, slow-slip–induced precursors, foreshock migration distances, and geodetic measurements), because no single method is uniformly applicable across all events. In the revised Methods, we now clearly describe, for each earthquake, the observational method used to estimate the quasi-static length and the underlying assumptions. Please see lines 363-391.

Importantly, our interpretation relies only on the order-of-magnitude consistency between quasi-static length, nucleation duration, and V_{min} , and not on precise values of any single parameter. Even with these uncertainties, the natural events

[REDACTED]

Figure 1: *Stress changes due to foreshocks on the 1999 İzmit earthquakes (Ellsworth and Bulut, 1999) [2].*

exhibit the same systematic relationships observed in the laboratory and predicted by the EoM framework. We have added a dedicated subsection in the Methods detailing the estimation procedures for all events listed in Methods Table S1 and state how the estimates are performed in the main text and the caption of Figure 4. We address all the individual comments in the following answers and in the revised version of the manuscript, for each individual earthquake.

Note that these changes have also resulted in changes to Figure S9d (previously Figure S5d).

- a) *The 1999 İzmit earthquake is interpreted in the manuscript as having a quasi-static nucleation length of approximately 300 m, based on the documentation by Bouchon et al. (2011). However, this value corresponds to the size of the foreshock-ruptured patch, not the quasi-static nucleation length. Subsequent analysis by Ellsworth and Bulut (2018) reported even smaller dimensions (<100 m) for these foreshock patches (Figure R1 [here Figure 1]). Moreover, Ellsworth and Bulut explicitly stated that “The absence of any other repeating foreshocks argues against a process driven by aseismic slip (fault creep) as envisioned previously.” Thus, there is no observational evidence linking the foreshock patch size to a quasi-static nucleation process for this earthquake.*

Response: We thank the reviewer for bringing attention to the interpretation of the 1999 İzmit foreshock sequence. We agree that Ellsworth and Bulut (2018) [2] argued that the absence of additional repeating foreshocks disfavors a quasi-static slip front, and we now cite their viewpoint explicitly. However, this conclusion is not universally accepted. In contrast, Bouchon et al. (2011)

[3] interpreted the presence of repeaters as evidence for an underlying quasi-static or slow-slip front during the nucleation of the İzmit earthquake. These two studies therefore draw different conceptual inferences from similar observations, reflecting a broader philosophical divergence in how precursory repeaters are interpreted.

That said, we acknowledge that our estimate of the minimal transient velocity for the İzmit nucleation sequence is necessarily uncertain, because the observations provide only an upper bound on the spatial scale of precursory slip. In addition, we agree with the reviewers that the occurrence of precursory slip in this sequence is questionable. Despite these points, we decided to retain İzmit in our figures but treat its values conservatively, highlighting the point with a different color and making a remark in the caption. Importantly, we were able to identify an additional natural case with a similar nucleation duration involving an uncontested slow-slip event triggered by a foreshock, which further supports the scaling we observe.

To provide full transparency, we have revised Fig. 4d to clearly distinguish (i) events preceded by slow-slip transients (Papanoa 2016), (ii) events unequivocally preceded by foreshocks, and (iii) cases such as İzmit where the interpretation remains debated. This clarification makes explicit which natural events most robustly support the proposed scaling relationships. Please see Figure 4, Figure 4 caption, and lines 382-384.

- b) *The 2014 Papanoa earthquake is discussed with reference to Radiguet et al. (2016). However, this study did not report any foreshocks; it only documented the preceding slow slip event (SSE). In addition, the metric used in the manuscript to infer a quasi-static nucleation length of 250 km from Figure R2 [here Figure 2] appears uncertain. The scale of the SSE does not directly translate into a quasi-static nucleation length, and the interpretation presented in the manuscript lacks sufficient justification.*

Response: We thank the reviewer for this observation. It is correct that, in contrast to several other natural examples and to our laboratory experiments, the slow slip event (SSE) preceding the 2014 Papanoa earthquake was not triggered by a seismic foreshock. However, there are two important points to consider. As can be seen in Figure S8 (old Figure S2) in the Methods, when the foreshock size ΔT tends to zero the EoM retrieves ℓ_∞ as the nucleation length limit. This means that for vanishingly small seismic foreshocks (such as those too small to be detected) the trend seen in laboratory and earthquake data between Δt_c and V_{\min} is still predicted by the EoM. We have added a subfigure to Figure S8 (old Figure S2) illustrating this more clearly. Secondly, while the EoM in this form requires a localized acceleration, this does not have to come from a seismic source. Therefore, aseismic asperity yielding accelerating slip would suffice to provide a ΔT .

For these reasons, we retain the Papanoa event in Fig. 4d, while clearly

[REDACTED]

Figure 2: *Spatio-temporal distribution of SSE on the 2014 Papanoa earthquake (Radigue et al., 2016) [4].*

distinguishing it from cases where nucleation is initiated by a recorded seismic foreshock by applying a different color in Figure 4. In the revised figure and caption, we now explicitly identify Papanoa as an event with a well-documented slow-slip nucleation phase without a clear seismic foreshock triggering the onset of the slow slip itself. We also now explicitly discuss the vanishingly small foreshock limit. Please see Figure 4, Figure 4 caption, and lines 343-345, 382-384, and 479-489.

- c) *The locations of the Mw 6.2 foreshock and the subsequent mainshock of the 2016 Kumamoto earthquake are shown in Figure R3 [here Figure 3] (Kato et al., 2016). To my knowledge, it remains controversial whether these events occurred on the same fault plane. While I agree that this sequence is broadly analogous to the scenario proposed in the present study, where the foreshock may promote quasi-static slip leading to mainshock initiation, a more careful discussion is required to evaluate the specific role of the foreshock in this earthquake.*

Response: We thank the reviewers for this insightful comment. Off-fault foreshocks introduce additional complexity compared to the on-fault point-force idealization used in Garagash (2021) [1]. In particular, their effects depend on geometric factors such as the distance to the main fault, the relative orientation of the secondary fault, and the possibility of creating shadow zones where static stress changes may inhibit rather than promote propagation [6, 7]. A full treatment of these effects is certainly possible, but it is well beyond the scope of this manuscript. To provide some detail, in order to perform this analysis, one would replace the shape function taken from Tada et al., 2000 [8], replacing it with the shape function appropriate for the foreshock-main-fault geometry.

[REDACTED]

Figure 3: *Cross-sections of spatio-temporal evolution of the seismicity towards the main-shock (Kato et al., 2016) [5].*

Figure 4: Taken from Figure 4b in the manuscript.

However, ultimately, in the case of a well-oriented off-fault foreshock, a contribution will be made by the foreshock via stress redistribution to the crack tip. This contribution will ultimately decrease as the crack grows. Therefore, the same characteristic behaviour of initial acceleration, a velocity minimum, and finally reacceleration to seismic rupture will still occur (in the case the foreshock is the appropriate size to result in Regime ii, considering the background stress, $\frac{\Delta f_0}{b}$).

Thus, although the full physics of off-fault foreshocks is considerably richer and more interesting than what we model here, the essential mechanism we describe remains valid. We therefore feel it is appropriate to retain the Kumamoto 2016 sequence as an illustrative example, even if its initiating foreshock was located slightly off the main fault.

Overall, I find that the evaluation of the quasi-static nucleation process is overly generalized when compared with the laboratory and theoretical insights into foreshock-to-mainshock sequences. It would greatly strengthen the manuscript if the authors could refine the discussion presented in the final section.

In addition, I was unable to identify the original data sources for the natural-earthquake markers plotted in Figure 4b (See annotation of Figure R4 [here Figure 4]). It would be helpful if the authors could explicitly specify the datasets used to construct these points.

We thank the reviewer for this comment. We hope that the clarifications provided throughout our responses address the reviewers' concerns regarding the interpretation of natural foreshock, mainshock sequences. Our experimental observations and theoretical development are intended to apply only to the limited number of

earthquakes for which a resolvable nucleation phase has been documented, whether triggered by a foreshock or accompanied by aseismic precursory slip. Because such cases remain rare, we consider it important to discuss all well-constrained examples available in the literature, and we believe that our framework provides a useful way to interpret these observations and motivates further analyses of similar sequences.

To strengthen this section, we have expanded the natural-earthquake dataset to include two additional recent events. The 2021 Yangbi (Mw 6.1) earthquake experienced approximately 200 mm of aseismic slip during the ≈ 54 minutes preceding the mainshock (Wang et al., 2024 [9]), and the onset of this slip was associated with a foreshock. Although the full time series is unavailable, we now include this event in Figure 4d using a distinct color and compute a conservative overestimate of its minimal velocity using the simple average $V_{\min}^* = 200 \text{ mm}/3240 \text{ s}$. From the same study, we estimate $\ell_c \approx 13.5 \text{ km}/2$.

We have also added the 2017 Valparaíso (Mw 6.9) earthquake, for which Ruiz et al. (2017) [10] reported $\sim 0.09 \text{ m}$ of precursory aseismic slip beginning four days before the mainshock. Because the full time series is again unavailable, we compute a conservative average $V_{\min}^* = 0.09 \text{ m}/346000 \text{ s}$, and from their geodetic inversion we estimate $\ell_c \approx 90 \text{ km}/2$. As can be seen in Figure 5 (this document), both events likely fall within the order-of-magnitude trends highlighted in our revised Figure 4d when the overestimation associated with the simple averaging is taken into account.

Finally, we have added an additional point for the Tohoku-Oki earthquake. It is likely that the M7.3 event two days before the M9.0 mainshock had a significant effect on the nucleation dynamics. We have added a figure to illustrate this (Figure S11 of the main text). We further explain our use of two points for Tohoku-Oki in the added text of the “Transient sliding velocity” section. Please see Figures 4 and S11, Table S1, and lines 374-382, 384-391.

Finally, to address the reviewer’s question regarding the data sources for Figure 4b, many of the previously unlabeled points originate from Whitcomb et al. (1973) [11], and these have now been assigned a unique color for clarity.

13. • *Material dependent frictions: PMMA bare contact has similar features to those of rock and fault gouge?*

The stick-slip experiments were conducted on PMMA samples with bare contact and without gouge. I think this condition is crucial for reproducing a quasi-static nucleation process that is initiated by a foreshock. In contrast, experiments using rock samples have demonstrated a different scenario, in which preslip develops around asperity patches and subsequently generates foreshocks as a byproduct (e.g., McLaskey and Kilgore, 2013). While I agree that such foreshocks can influence the transient sliding velocity and the nucleation duration, the correlation between these quantities is far less clear in rock experiments than in the present study. The presence of fault gouge also affects foreshock generation (Yamashita et al., 2021).

Figure 5: An alternate version of Figure 4d where in addition to the estimates of V_{\min} (dark circles) an estimate using a simple average (i.e., total aseismic slip divided by time) is shown (white circles). Additionally, Papanoa 2015 (pink) was a slow slip event not triggered by a foreshock [4]. İzmit 1999 (purple) is a debated case whose repeaters may or may not be linked to aseismic slip [3, 2]. Time series data for Yangbi 2021 [9] and Valparaiso 2017 [10] (white) are not available such that only a simple average is used to estimate V_{\min} . This figure shows that the overestimation of V_{\min} due to the simple average for Yangbi 2021 and Valparaiso 2017 likely corresponds to the overestimation V_{\min} for the earthquakes where time series data is available. Therefore, these two earthquakes likely support the trend shown by the dark circles.

Therefore, although I agree that the laboratory observations are broadly consistent with the EoM framework, extending these findings to natural earthquakes requires further discussion that accounts for differences in frictional conditions.

Response: We thank the reviewer for raising this important point. We agree that frictional conditions, whether bare PMMA interfaces, bare rock surfaces, or gouge-rich faults, strongly influence the generation of foreshocks and the character of the quasi-static nucleation phase. Our use of PMMA was motivated by the need to isolate and quantify the effect of impulsive perturbations on nucleation under controlled and well-characterized conditions. The framework we develop does not rely on the specific material, but on the dimensionless parameter $\Delta T/(\mu' L) \sim \delta_a/L$, which captures the amplitude of the foreshock-induced perturbation relative to the characteristic slip distance L .

In this regard, PMMA offers a key scaling advantage. In our experiments, $L \sim 0.1 \mu\text{m}$, whereas bare-surface rock experiments typically have L values between 1 and $10 \mu\text{m}$ (Kilgore et al., 1993 [12]). Regarding gouge, quartz gouge's L might be between 2-25 μm (e.g., Marone and Kilgore, 1993 [13]). To produce a comparable perturbation in rock, the foreshock slip δ_a would therefore need to be 10 to 100 times larger in rock experiments, or even larger in gouge experiments. As documented in McLaskey (2019) [14], such slip amplitudes in rock samples are often comparable to the mainshock slip itself, indicating that true foreshock-induced perturbations may be difficult to generate at laboratory scale. In contrast, natural faults are much larger, and foreshocks on such faults can accumulate slip amplitudes that are significant relative to their effective L , allowing them to contribute meaningfully to the equation of motion.

The contribution of background loading enters through the parameter $\Delta f_0/b$, which does not depend on the absolute value of L . This makes it possible to compare the effects of impulsive perturbations and background loading across different materials.

As detailed in the Supplementary Material, the use of PMMA also provides a second scaling advantage. The cohesive zone and the characteristic slip distance L are proportionally much smaller relative to the 40 cm experimental fault length than in rock samples. A 40 cm rock interface remains a 40 cm interface when extrapolated to natural conditions. In contrast, a 40 cm PMMA interface, with a cohesive zone and L one to two orders of magnitude smaller, more closely reproduces the natural ratio of cohesive-zone size to fault length. This ratio is essential for capturing the physics of rupture nucleation in the laboratory.

We further wish to clarify the definition of foreshocks in the context of our study. In many natural and experimental systems, a nucleation phase initiated by stable slip can generate small foreshocks or acoustic emissions within the slipping patch. These events reflect the onset of nucleation, and they are valuable for detecting this phase, but they do not contribute to the nucleation dynamics because they do not generate a significant impulsive perturbation. In our framework, a fore-

shock is considered dynamically relevant only when its slip, δ_a , is large enough relative to L to increase the transient slip velocity and thereby shorten the nucleation duration. The foreshocks described in several of the rock experiments cited by the reviewer appear to fall into the former category, namely small-scale acoustic emissions produced by slip at small asperities, presenting small rupture area, and consequently small slip, and which do not modify the nucleation dynamics for the reasons explained above. We now refine this distinction in the revised manuscript and explicitly state our definition of dynamically-relevant foreshocks in the conclusion.

For these reasons, although we agree that gouge-rich or asperity-controlled rock experiments may complicate the observed correlations between foreshock slip and nucleation duration, we believe that our PMMA experiments capture the fundamental influence of impulsive perturbations on nucleation. We have expanded the discussion in the final section of the manuscript to clarify these points and to better articulate the role of frictional conditions in the applicability of our framework. Please see lines 158-161, 277-285, and 335-345.

Other Comments

14. *First line of the abstract: The statement “Coseismic rupture is commonly preceded by stable fault slip, which may accelerate into dynamic rupture as the slip front reaches critical velocities” may not be acceptable for some readers, given that the nature of the nucleation process remains controversial. Cascade-up scenarios without substantial quasi-static preslip are also plausible for large earthquakes (e.g., Ide, 2019). Therefore, the introductory assertion may need to be reframed to acknowledge this variability.*

Response: We have rewritten the abstract and this phrase no longer appears. Instead, we say that “Classical models often assume that nucleation arises from slow, quasi-static slip governed primarily by fault weakening”. Please see line 9.

15. *I think the role of foreshocks in earthquake nucleation is extensively discussed in McLaskey (2019). It would be helpful for the authors to clearly distinguish the foreshock-driven nucleation concepts presented in that work from the framework used in the present study, particularly by emphasizing the role of foreshocks in initiating quasi-static slip that evolves toward the mainshock.*

Response: We have added two sentences to the “Transient sliding velocity controls earthquake nucleation” section which we feel addresses this concern. In particular, we clearly define which earthquakes we feel should be designated as foreshocks with a clear connection to the work of McLaskey (2019) [14].

“Based on these considerations, we suggest the use of the word foreshock should be restricted to earthquakes which either initiate or tangibly influence the quasi-static nucleation phase of another, larger earthquake. This definition would exclude earthquakes occurring during preslip-like[14] but not necessarily those during rate-dependent-cascade-up[14] nucleation phases.” Please see lines 158-161.

16. *Figure 1 Caption: Please provide more detailed information on "videogram". What is the physical quantity the videogram shows? Which is bigger in dark/light color? better to provide color scales.*

Response: The videogram measures light intensity. At each instance of time (with the camera recording at 1 MHz) 1280 pixels along the fault are recorded. Each of these pixels is associated to one location along the fault ($\sim 40 \text{ cm} / 1280 \text{ pixels} \approx 0.03125 \text{ cm/pixel}$) and records one value of light intensity. The light intensity that the camera records changes as the stress in the PMMA changes because PMMA is a photoelastic material. In more detail, PMMA is initially isotropic but has stress-induced birefringence. As the stress in the material changes, the principal stress directions cause a phase shift between the two orthogonal polarization components, altering the polarization state of the transmitted light. A second linear polarizer, placed before the camera, converts these polarization changes into measurable intensity variations. This means that the light intensity recorded is a reflection of the local shear stress along the fault. As propagating ruptures result in stress changes, they can be visualized with this method. As far as the authors are aware, this technique was first applied to shear ruptures by Rosakis et al., 1999 [15], with the following references applying it for spontaneous ruptures [16, 17, 18, 19, 20, 21].

Regarding color scales, calibration can be performed to relate the light intensity changes to specific values of stress. Our interest is not to use this method to have the precise value of stress along the fault. We instead use strain gauges for this and therefore do not calibrate the light intensity to precise stress values. As the calibration depends on the strain gauges, we prefer to just simply use the strain gauges for stress measurements. Instead, we use the videograms to track the propagating rupture, a procedure which does not require the calibration to stress. For these reasons a color bar is of limited interest.

We have added a reference to the Methods where we explain this approach. We have also added two references who were among the original pioneers for this approach. Finally, we added some text to this part of the Methods section to improve clarity. Please see lines 283-285, 297-299, and 321-325.

17. *Figure 1b: The shear stress τ appears to exceed the normal stress σ . Does this imply that the friction coefficient becomes exceptionally large (>1.0) under these conditions? If so, this point is directly related to my major comment #3 regarding material differences and their implications for the nucleation process.*

Response: It is true that locally the ratio of shear stress to normal stress can become large (>1) in our experiments. However, macroscopically the ratio between these two stresses is less than one, as in rocks. To illustrate this, we show the force readings of the load cells, which yield the macroscopic stress ratio, Figure 6 (this document). This figure shows that all events occur at a stress ratio less than 1. Note that recently the applicability of a macroscopic friction coefficient has come into question (Ben-David and Fineberg, 2011) [22]. While this is not the focus of our manuscript, the results shown in Figure 6 (this document) are in agreement that macroscopic friction coefficient is not a material parameter.

Regarding the local stress ratio, large local stress ratios are often observed along long (> 20 cm) frictional surfaces. For example, Xu et al., 2023 [23] show local shear-to-normal stress ratios exceeding 1 on a long (1.5 m) fault made of Indian metagabbro samples in their Figure 1d. Indeed, local stress ratios often exceed macroscopic stress ratios. For example, Yamashita et al., 2018 [24], their Fig. 11b shows local stress ratios between 0.4 and 0.5 while the macroscopic stress ratio is as low as 0.25 (also metagabbro). For these reasons, the nucleation of dynamic events with macroscopic stress ratios between 0.4 and 0.7 while the local stress ratio exceeds one at certain positions along the fault is not something that could not be observed in rock.

18. *Figure 1c-e: Please add the amplitude scale for accelerograms.*

Response: Done. Please see Figure 1.

19. *Figures 1c-e & S1: I was wondering how the authors determine the nucleation time. It was difficult for me to identify the starting point of the nucleation visually from the videogram since the color change was quite tiny there. Did the authors confirmed if no slip generated before the foreshock (i.e., the foreshock occurred slightly earlier than the start of nucleation)? And did the foreshock occurred exactly the same location as that the nucleation started, or it occurred slightly different location? Such information should be important to support the authors scenario proposed here.*

Response: In fact the white star is the located foreshock (in time and space) from the accelerometers. Therefore, the located foreshock and the results of the videogram agree very well in terms of the foreshock location. It is true that it is hard to see the foreshock just from the videogram, but actually it is visible. We have included a zoom on the foreshock location for these three examples, Figure 7(d-f) (this document). In this figure the stress change due to the foreshock is visible. The foreshocks were localized based on the accelerometer arrival times. While there is uncertainty in these measurements, due to, for example, an assumed shear wave velocity, the events were localized to 6.5, 5.5, and 8.7 cm agreeing with the videogram results to within ± 3 cm. In time, these events were localized to 3.9165, 13.1365, and 17.9640 msec on these figures. These located times are

Figure 6: The load cells measurements for an experiment performed at 150 bar. **a**, The voltage readings for the normal and shear forces. **b**, The macroscopic stress ratio. Stars indicate a dynamic event.

Figure 7: Unannotated videograms for the three events shown in Figure 1 of the main text. (a-c) show the unannotated version of Figure 1 (note that c is already zoomed in here; in the main text 25 msec is shown). (d-f) show a zoom. The foreshocks, while faint, can be seen in the zoomed images. For reference, the foreshock in (a,d) was localized by the accelerometers to 6.5 cm; the foreshock in (b,e) was localized to 5.5 cm; the foreshock in (c,f) was localized to 8.7 cm. These values are in reasonable agreement to locations seen visually on the videograms.

taken as the start of nucleation. The locations are not used explicitly anywhere in the manuscript (crack lengths are based on the traced videogram), but they are inherent in the calculation of foreshock magnitude A_{fs} .

Regarding the slip occurring before the foreshock, we have plotted the slip for a significant amount of time before the foreshock, Figure 8 (this document). There is no distinguishable slip on this time scale.

20. *Figure 2: Each point corresponds to measured data? Since original data was sampled at 2MHz, is this data averaged at constant slip (or time) window? Can you show the location of the sensors? It seems better to use smaller symbols to see more details.*

Response: The location is 5.2 cm for all events. This corresponds to the approximate position of the events as seen on the videograms. The results of the two nearest strain gauges (5.1 and 8.0 cm) and displacement sensors (3.5 and 6.5 cm) are linearly interpolated to this position. In this sense, each point corresponds to interpolated measured data. The data are filtered with a low-pass filter at 10 kHz. We have added the sensor positions as well as an improved description to the caption. The symbol size has been reduced. Please see Figure 2 and Figure 2

Figure 8: The videograms of the three events shown in Figure 1 of the main text overlain with the displacement data from the displacement sensors. The time is shown up to 80 msec earlier than in Figures 1 and S1 to show that displacement is not occurring prior to the foreshocks. Note that the videogram is not recorded up to -80 msec due to a lack of storage space on the camera. Note that a closer view of the displacement data during nucleation is shown in Figure S1.

caption.

21. *Line 45-46: According to Uenishi and Rice (2003), a sharper weakening rate leads to a shorter critical nucleation length. However, this does not necessarily imply that the weakening rate directly controls the nucleation duration, which is influenced not only by the weakening rate but also by the loading rate. Therefore, this section requires further clarification to avoid overinterpreting the role of the weakening rate in determining the temporal evolution of nucleation.*

Response: We agree that the model of Uenishi and Rice (2003) [25] is dependent on the loading rate (their R) as demonstrated by their Equation 15. However, in this equation the weakening rate (their W) appears via their variable a_c . In this sense, we agree with the reviewer that the model of Uenishi and Rice (2003) has a dependence on both weakening and loading rate. We also note that the effect of loading rate has been previously studied experimentally (e.g., McLaskey 2019; Guérin-Marthe et al., 2019) [14, 26]. We have added clarification to the sentence indicated by the reviewer to show that weakening rate is not the only parameter to be considered, and now cite Guérin-Marthe et al., 2019. Please see line 46.

22. *Figure 3c: Several data points exhibit one-sided error bars. While asymmetric error bars are understandable given the log scale, the meaning of the one-sided representation is unclear. Please clarify how these uncertainties were calculated and why a one-sided error bar is appropriate for these particular data points.*

Response: This occurs because the error associated with the data point is larger than the value of the data point. The left side of the error bar does not plot on the log scale as the log scale is not appropriate for negative values. The error

for the slip during the foreshock is taken as the difference between the maximum and mean values read by the displacement sensor in the first 0.5 msec after the nucleation phase initiates. This represents a conservative estimate. We now limit the lower bound error to 50% of the measured value, with an explanation in the caption. Please see Figure 3c and Figure 3 caption.

23. *Line 107: The statement “The advancing crack tip draws energy both from the foreshock and from the inherent stress state along the fault” is intuitively understandable. However, some readers may be concerned by this phrasing, because a foreshock dissipates stored elastic energy and therefore cannot literally serve as an energy source for the system. It would be better to rephrase this sentence to avoid implying that foreshocks contribute energy rather than perturbing the stress field.*

Response: This sentence has been removed and replaced with “The advancing crack tip draws energy from both the inherent stress state along the fault and the stress concentration (or redistribution) generated by the foreshock.” Please see lines 106-108.

24. *Line 142: The cascade-like process is generally associated with nucleation without a significant quasi- static slip phase, as illustrated in the schematic by McLaskey (2019). Therefore, it may be helpful to clarify how the cascade-like behavior described here relates to the quasi-static nucleation framework used in this study.*

Response: The reviewer is right to point out that our wording was not precise here. The cascade end-member indeed does not generally have significant quasi-static slip. We have updated the text to reflect that we mean that our results are compatible with a spectrum of behaviours between the preslip and cascade models, corresponding instead to the rate-dependent cascade up model of McLaskey (2019) [14]. We also emphasize that the framework allows for cases where the foreshocks are the driving mechanism of the main shock; essentially the case on the spectrum of behaviours most similar to the original cascade model. Please see lines 150-152.

25. *Although it would require additional effort, it would strengthen the manuscript if all videograms with the annotation like Figures 1d could be summarized into a single supplementary figure to reinforce the argument regarding the relationship between foreshock size and nucleation duration*

Response: We have prepared a summary figure showing the relationship between foreshock size and nucleation duration, Figure 9 (this document). We have also added it to the Methods section, referencing it in the main text. Please see Figure S4 and line 53.

Figure 9: The nucleation duration versus foreshock size; showing that foreshock size (as measured by the accelerometers) directly relates to nucleation duration. The color bar shows the slip during the foreshock (as measured by the displacement sensors).

Reviewer #3

26. *I like this paper. It attempts to make sense of laboratory friction experiments in which the time and length scales of contained earthquake nucleation are quite variable, despite the frictional properties remaining nominally constant (e.g., as suggested by Fig. 2). To a large extent I think it succeeds. The main conclusion is that the foreshock(s) that often precede laboratory earthquakes provide an impulsive stress change to the fault, a stress change that manifests itself as a perturbed sliding velocity that ultimately affects the time and length scales of nucleation. Similar observations have been made previously, but not really addressed theoretically; I imagine this behavior was considered too complicated to analyze theoretically. Numerous simplifying assumptions are made in the theoretical treatment, but I believe it's reasonable to give the first paper to really tackle this issue some slack. The theoretical results allow the authors to parameterize the nucleation behavior in terms of the minimum slip speed reached on the fault between the times of the (largest?) foreshock and the ensuing main shock, and this treatment does a respectable job of collapsing the lab data onto interpretable trends. The authors go on to apply the same theoretical framework to natural earthquakes, and it appears that they follow the same trend to larger length and time scales. Even more assumptions are required to interpret the natural earthquakes in this way (mostly in determining the minimum fault slip speed, achieved just after the foreshock), but the attempt certainly provides food for thought, and appears to be successful, so it adds a lot to the paper.*

In a bit more detail: The theoretical analysis relies on previous work by one of the authors (Garagash, 2021), who developed an “equation of motion” for the tip of the nucleation/rupture front, by matching the rupture-velocity-dependent stress intensity factor K at the tip to the fracture energy required for growth of the rupture. In the absence of foreshocks, K for a steady-state velocity-weakening surface increases as the nucleation zone grows in size, while a foreshock provides a separate contribution to K that decreases as the nucleation zone grows. The result is that 3 nucleation regimes arise: (1) The foreshock is so weak that the nucleation attempt dies without generating a main shock; (2) The fault slip speed initially decreases due to the reduction in the foreshock contribution to K , but subsequently accelerates as the background, velocity-weakening contribution takes over; or (3) Slip accelerates monotonically and rapidly toward failure. Most of the paper deals with nucleation in the presence of foreshocks, regime (2), in which larger foreshocks lead to larger minimum slip speeds, shorter nucleation times, and smaller nucleation lengths. I think a fair amount of insight was required to recognize that the important behaviors could be parameterized in terms of the minimum fault slip speed, which is influenced by both the foreshock size and the fault pre-stress. One of the strengths of the paper, I think, is that regardless of the details of the equations used, the idea of a competition between two contributions to K , one of which increases as the nucleation zone grows and one of which decreases, seems robust, and a sensible way of thinking about the large differences in nucleation time on what is nominally the same fault.

I think the combination of a realistic model for the laboratory observations of earthquake nucleation, and a plausible extrapolation to natural earthquakes, makes this certainly suitable for publication in Nature. The model makes sense of observations that previously have not really been addressed. Some fitting parameters/model simplifications are involved, so it’s hard to guarantee that the model is the appropriate one, but I’m fine with that. Larger rate-state slip distances are required for the natural examples, using the same model, but that’s plausible. I have the following, mostly minor, comments, in chronological order:

Response: We thank the reviewer for their thoughtful and constructive evaluation of our manuscript. We appreciate the clear summary of the study, the recognition of the conceptual framework we developed, and the careful consideration given to both the laboratory and natural-earthquake applications. We are grateful for the reviewer’s positive assessment of the paper’s contributions as well as for the detailed comments, which have helped us improve the clarity and robustness of the manuscript. We have addressed all points raised and believe the revisions have strengthened the work. In particular, the reviewer’s comments have helped us improve the aging vs. slip law discussion.

27. *Figure 1 caption: I think this should read “lengths, widths, and heights of (x ,*

$y, z) = (40, 10, 1) \text{ cm}$ ", assuming that y is height and z is width (front-to-back).

Response: We have changed this to read "lengths, heights, and widths". Please see Figure 1 caption.

28. *The short-duration nucleation event (0.6 msec, triangles) in Figure 2 looks surprising, in that there appears to be very little stress drop before the nucleation phase is over/rupture speed of 10 m/s is reached. There are some triangles at higher and lower stress levels than the black triangle at 0.6 msec, so perhaps that variation is indicative of noise in the measurements, but from the symbol colors there is good temporal consistency between the measurements. Making this harder to understand is that the data seem to show a non-intuitive relation between time and slip: The darker green colors, which are earlier in time, occur at greater slip distances than the pale green colors, which occur later. So taken literally, the earliest measurements (dark green) show more slip and smaller stresses, followed by (light green) less slip and larger stresses, with the end of the nucleation phase (black symbol) having larger slips but intermediate stresses. Is this apparent inconsistency the result of stress, slip, and rupture speed being point measurements made at different locations? In any case, some comment might be in order, because otherwise it's hard to understand what is going on.*

Response: Indeed, the reviewer is correct. There are two things happening which result in, for example, certain data from the 0.6 msec event (triangles) occurring at larger slip and lower stress than the dark triangle (the moment nucleation is considered to be achieved), despite these points being colored in green and therefore occurring earlier in time than the black triangle. First, there is noise in the data both in the strain gauge (stress drop) and displacement sensor (slip) data. Second, as the data are coming from two different sensors, the measurements are not at exactly the same location. The rupture speed is then measured with a third instrument (the high-speed camera) and used to establish a common (zero) slip reference state for the other two measurements. This indeed results in slight mismatch between the measurements. We have expanded the figure caption to explain this. Please see Figure 2 caption.

29. *On line 40 I would add the word "detectable", as in "without the intermediate stage of detectable quasi-static slip-front propagation".*

Response: Added. Please see line 40.

30. *The Figure 3 caption refers to the foreshock slip distances δa , and cites the Methods for determining them. But I can't find how this was done; perhaps I read the relevant Methods section too quickly. There is a "Foreshock magnitude" section that gives equation (S6) for how the averaged acceleration A_{fs} is determined, but*

I don't see how that is converted to slip distance. Source-station distances in (S6) are normalized by a reference sphere of radius 1 cm; is that an estimated source size, and the same size is used to convert accelerations to slip distances? But that would also require an estimate of slip speed. Or is that 1 cm the footprint of the accelerometer? I confess I didn't follow up with reference 44. Or does the estimate of δa just come from the match between the data in Fig. 3a and the model runs in 3b, with the relation between ΔT (the centrally-applied force on the crack from the foreshock) and δa coming from equation (S10) and the empirical C ? It would be good to clarify this.

Response: In fact the foreshock slip distances δ_a are directly measured by the displacement sensors. They are not found using the accelerometers. We have added an illustration of this procedure to Figure S2 (old Figure S3) and have updated the captions of Figures 3 (referencing S2) and S2 (old Figure S3).

31. *line 136: Again, I would insert the qualifier “observable” here: “This suggests that while an observable nucleation phase often occurs, it is not a strict prerequisite for rupture”.*

Response: Added, although note this sentence has been reformulated. Please see lines 137-146.

32. *The data for natural earthquakes in Figure 4d, plotting the estimated nucleation duration (easy to determine, when defined as the time between the largest foreshock and the main shock) as a function of the minimum slip speed (presumably very difficult to determine), seems almost too good to be true. There's a statement in the Methods that “these data rely on either inversion of the fault slip based on geodetic data[,] or the supposition that the slip during repeater earthquakes is representative of the aseismic slip of the patch”. In addition, use is made of the modeling result that slip accumulates exponentially with time, so that data at multiple times can be fit with a straight line on a semi-log plot, as shown in Figure S4. This information is too important to leave for the Methods only. Right now there's just a statement in the main text that “Additionally, part of the discrepancy may stem from the uncertainty in estimating transient sliding velocities for natural earthquakes; our values, based on indirect constraints, likely overestimate the V_{min} during nucleation”. I don't think there's even a citation to Figure S4. Please include something like the quoted sentence from the Methods, plus the theoretical result and a mention of S4, in the main text.*

Response: We thank you for this comment, which was also raised by the two other reviewers in comment #12. We have now included a description of how this data is estimated in the main text, including a reference to Figure S10 (old Figure S4). Please see lines 130-133, 363-391, and Figure 4's caption.

33. *Foreshocks and nucleation length scaling: Starting on line 320 of Methods is written “The coexistence of foreshocks and a larger nucleation front can be rationalized by considering the scale dependence of nucleation. At the scale of micron-sized asperities, local stress concentrations lead to significantly higher effective normal stresses than the macroscopically-applied value. Because the theoretical nucleation size scales inversely with normal stress, ...”. I’m wondering if two concepts are being conflated here. I could easily imagine that at small scales, but still encompassing many asperities, rate-state properties, normal stress, and (for non-planar surfaces) the slip-dependence of normal stress can be variable, and produce foreshocks on length scales smaller than the main shock nucleation zone. But the statement “At the scale of micron-sized asperities, local stress concentrations lead to significantly higher effective normal stresses than the macroscopically-applied value” makes it seem that the intent might be to appeal to local stresses at the single-asperity scale that exceed the remotely-applied normal stress by one over the fractional contact area, where the latter could be less than 1%, which would then drop the nucleation zone size by a factor of more than 100. I’m not sure the standard RSF equations can be applied at this scale, but are the authors proposing that the laboratory foreshocks represent rapid slip of a single asperity? Typically we think of the characteristic asperity size as being approximately L in the RSF equations, which would be about $0.2 \mu\text{m}$ for these experiments. But the inferred foreshock slip distances are several times this value, which is difficult to reconcile with single-asperity-scale sources (contact stresses might be 10% of the shear modulus, so a complete stress drop to zero on an asperity might give rise to displacements of a few times 10% of the asperity dimension, or less than say $0.05 \mu\text{m}$, which is too small, and complete stress drop is outside the standard RSF equations anyway). So please clarify that (I think) you are not proposing that the foreshocks represent slip of a single asperity, with friction on that single asperity being governed by the standard RSF equations. This might be as simple as writing “At the scale of multiple micron-sized asperities, local stress concentrations...”.*

Response: We agree with the reviewer. We were aiming to offer an explanation for how a foreshock can co-exist with a larger nucleation front. We agree that these foreshocks would likely require multiple asperities for the reasons the reviewer outlined. We have changed the text as the reviewer suggested. Please see line 347.

34. *Slip- vs. Aging-law simulations: Line 366 of Methods says “Considering the apparent invariance of the weakening rate W in our experiments (Fig. 2), we adopt the aging law for the remainder of the analysis”. But this argument could equally well favor the Slip law – the Slip law predicts a weakening rate that is larger for larger peak-residual stress drops, not seen in Fig. 2, while the Aging law predicts larger slip-weakening distances for larger stress drops, also not seen in Fig. 2. What you can say is that Fig. 2 shows a quasi-exponential decay to the new friction level, which is a hallmark of the Slip law (also the Aging law, but only for very small*

velocity increases). The fact is that the range of stress drops in Fig. 2 is small enough that one could reasonably not expect to see either an increase in weakening rate (if the Slip law were appropriate) or a longer effective slip-weakening distance (if the Aging law were appropriate), but the exponential stress decay looks a lot like the Slip law. Line 368 continues “Note that the model presents a qualitatively-similar behaviour using the slip law”, but I imagine this must be for values of a/b not too close to 1. In any case, use the Aging law if you wish, but I’m not sure it’s fair to cite Fig. 2 in support of this.

Response: We agree with the reviewer that this is not an appropriate argument for the adoption of the aging law. Indeed, as the reviewer points out, the range of stresses of these events is too small to reasonably distinguish between the two based on their slip-weakening slopes alone. We now cite Berthoud et al., 1999 [27] who perform friction experiments in PMMA and find that the aging law is appropriate. Note that this is from the same series of papers as the Baumberger et al., 1999 [28] on which our friction parameters are based in the methods.

Regarding the qualitatively similar behaviour between the aging and slip law, we have performed slip law simulations and included them in Figure S6b, updating Figure S6a (old Figure S9) for comparison. These simulations show the same characteristic behaviour as the original aging law simulations (initial impulse-driven fast rupture velocity, slow-down as crack length extends, reacceleration to dynamic rupture). The main qualitative difference between EoM predictions for the aging and slip laws, respectively, is the non-physical ‘switch-back’ behavior of the latter, also commented on by Garagash [1], when in the reacceleration stage the EoM solution passes through a point of infinite acceleration while still in the phase of slow quasi-static slip (the point with a vertical tangent in Fig S6b), which then necessitates an instantaneous jump of the rupture speed to the seismic range to regain the EoM solution there. This can be a reflection of a non-trivial (not-crack-like) actual development of later stages of nucleation process for the slip-law rate-and-state friction studied in Ampuero and Rubin, 2008 [29], when spatial localization of slip leads to the transition from crack-like to pulse-like slip development. We can speculate that it is this transition that can lead to a non-physical switch-back behavior of the EofM solution predicated on the crack-like mode of slip. Notwithstanding these differences, our simulations show the same three regimes for the slip law simulations as the aging law simulations (arrested rupture, slow-down to quasi-static before reacceleration, and direct transition into dynamic rupture with no quasi-static phase).

Finally, in Figure 2 we can observe a slip weakening distance, δ_c , of approximately 1-2 microns. We note that what appears as the continuing weakening with slip past 1-2 microns of the nucleation slip on Fig. 2 likely represents a near steady-state evolution of friction with sliding velocity away from the accelerating rupture front, and thus it is not expected to reflect the near-front breakdown process with slip (exponential for slip-law and approximately linear for aging law). Considering

that, for the slip law, $L \sim \delta_c$, and the largest possible nucleation crack length $2\ell_c \sim \frac{2\ell_b}{1-a/b} = \frac{2\mu' L}{(1-a/b)b\sigma}$, the slip law would predict a nucleation length of approximately 28 cm, since the local normal stress $\sigma \sim 1$ MPa, $\mu' = 1.24$ GPa, $L \sim 1.5$ microns and $a = 0.016$ and $b = 0.029$. Note that this is significantly longer than any nucleation lengths observed either in this study or the previous study on the same apparatus in PMMA [21]. Conversely, for the aging law $L \sim \delta_c / \frac{\Delta f_p}{b}$. As $\frac{\Delta f_p}{b} \sim 5 - 10$, $L \sim 0.2$ microns (note this value also agrees with our inverted values of L in Figure 4, as well as with the value 0.4 microns inferred for L by Baumberger et al., 1999 [28] from velocity step experiments on a PMMA fault). By then calculating, for the aging law, $2\ell_c = 2\ell_\infty \approx \frac{2}{\pi} \frac{\ell_b}{(1-a/b)^2} = \frac{2}{\pi} \frac{\mu' L}{(1-a/b)^2 b\sigma} \sim 3$ cm, which is in much closer agreement with our results, in particular when considering uncertainty and variations in L and σ . Therefore, the slip law predicts unrealistically large ℓ_c for our experiments, whereas the aging-law-predicted nucleation lengths more closely agree with experiment.

We recognize that, while the aging law may be more appropriate for our PMMA experiments, a recent study with bare-surface granite has indicated that the slip law may be more appropriate for rock bare surfaces (Bhattacharya et al., 2022 [30]). Conversely, other authors have found the aging law appropriate for quartz gouge (Mei and Wang, 2024 [31]; their figure 6). At any rate, it is not our goal to advocate for use of one law over another or to systematically explore their differences (while we feel such an analysis would be outside of the scope of this study, we do feel it would be an interesting avenue of research). Instead, we use the aging law for the natural events for simplicity and continuity with our experiments and because use of the slip law, if we were to employ it for natural earthquakes, would provide qualitatively similar results as we now show with Figure S6 (old Figure S9).

We thank the reviewer for this helpful comment. We have updated Figure S6 (old Figure S9) and improved the justification given for the aging law in the methods. Please also see lines 415-418.

35. *line 464-65 in Methods: This must be μm , not m .*

Response: Correct, changed. Please see lines 526-527.

36. *Supplementary Figure S5: I find it difficult to distinguish between red and magenta in panels a and b. Should it be “red dotted line”, rather than red dashed line?*

Response: We have changed the caption to say “dashed” for this line and have changed its color to green. Please see caption for Figure S9 (old Figure S5).

Additional changes

37. **Additional change:** We have reformulated the response to our question (i) in the “Transient slip rate controls nucleation” section. We now emphasize that, not only are not all earthquakes initiated with a quasi-static nucleation phase, but not all quasi-static nucleation phases are initiated by a foreshock. Please see lines 137-146.

38. **Additional change:** We have changed the color bar from σ to ℓ_c in Figure 4c. We feel the independence relative to normal stress is still understood with Figure 4b and that the control of V_{\min} on ℓ_c is too important to be only shown with modelling results (Figure S8; old Figure S2). Note that this also results in the same change to Figure S9 (old Figure S5). We then also changed the color bar for the EoM supplementary figures to avoid a clash.

39. **Additional change:** Supplementary Table S1 listed “Quasi-static length”; however, ℓ is defined as a half length. Therefore, these values have been divided by two. This has also resulted in a factor 2 shift in Figure 4b.

40. **Additional change:** We have changed the order of the figures in the Methods to match their order of appearance in the text of the manuscript.

Yours sincerely on behalf of all authors,

Barnaby Fryer

References

- [1] D. Garagash. Fracture mechanics of rate-and-state faults and fluid injection induced slip. *Philosophical Transactions of the Royal Society A*, 379, 2021. <https://doi.org/10.1098/rsta.2020.0129>.
- [2] W. Ellsworth and F. Bulut. Nucleation of the 1999 Izmit earthquake by a triggered cascade of foreshocks. *Nature Geoscience*, 11:531–535, 2018. <https://doi.org/10.1038/s41561-018-0145-1>.
- [3] M. Bouchon, H. Karabulut, M. Aktar, S. Özalaybey, J. Schmittbuhl, and M. Bouin. Extended nucleation of the 1999 M_w 7.6 Izmit Earthquake. *Science*, 331:877–880, 2011. <https://doi.org/10.1126/science.1197341>.

- [4] M. Radiguet, H. Perfettini, N. Cotte, A. Gualandi, B. Valette, V. Kostoglodov, T. Lhomme, A. Walpersdorf, E. Cabral Cano, and M. Campillo. Triggering of the 2014 M_w 7.3 Papanoa earthquake by a slow slip event in Guerrero, Mexico. *Nature Geoscience*, 9:829–833, 2016. <https://doi.org/10.1038/NGEO2817>.
- [5] A. Kato, J. Fukuda, S. Nakagawa, and K. Obara. Foreshock migration preceding the 2016 M_w 7.0 Kumamoto earthquake, Japan. *Geophysical Research Letters*, 43: 8945–8953, 2016. <https://doi.org/10.1002/2016GL070079>.
- [6] J. Laures and M. Kachanov. Three-dimensional interactions of a crack front with arrays of penny-shaped microcracks. *International Journal of Fracture*, 48:255–279, 1991. <https://doi.org/10.1007/BF00012916>.
- [7] Alice-Agnes Gabriel, Dmitry I Garagash, Kadek H Palgunadi, and P Martin Mai. Fault size-dependent fracture energy explains multiscale seismicity and cascading earthquakes. *Science*, 385(6707):eadj9587, 2024.
- [8] H. Tada, P. Paris, and G. Irwin. *The Stress Analysis of Cracks Handbook, Third Edition*. ASME Press, 2000.
- [9] K. Wang, Z. Peng, S. Liang, J. Luo, K. Zhang, and C. He. Migrating foreshocks driven by a slow slip event before the 2021 Mw 6.1 Yangbi, China earthquake. *Journal of Geophysical Research: Solid Earth*, 129, 2024. [10.1029/2023JB027209](https://doi.org/10.1029/2023JB027209).
- [10] S. Ruiz, F. Aden-Antoniow, J. Baez, J. Otarola, B. Potin, F. del Campo, P. Poli, C. Flores, C. Satriano, F; Leyton, R. Madariaga, and P. Bernard. Nucleation phase and dynamic inversion of the Mw 6.9 Valparaíso 2017 earthquake in central Chile. *Geophysical Research Letters*, 44:10290–10297, 2017. <https://doi.org/10.1002/2017GL075675>.
- [11] J. Whitcomb, J. Garmany, and D. Anderson. Earthquake prediction: Variation of seismic velocities before the San Francisco Earthquake. *Science*, 180:632–635, 1973. [10.1126/science.180.4086.632](https://doi.org/10.1126/science.180.4086.632).
- [12] B. Kilgore, M. Blanpied, and J. Dieterich. Velocity dependent friction of granite over a wide range of conditions. *Geophysical Research Letters*, 20:903–906, 1993. <https://doi.org/10.1029/93GL00368>.
- [13] C. Marone and B. Kilgore. Scaling of the critical slip distance for seismic faulting with shear strain in fault zones. *Nature*, 362:618–621, 1993. <https://doi.org/10.1038/362618a0>.
- [14] G. McLaskey. Earthquake initiation from laboratory observations and implications for foreshocks. *Journal of Geophysical Research: Solid Earth*, 124:12882–12904, 2019. <https://doi.org/10.1029/2019JB018363>.
- [15] A. Rosakis, O. Samudrala, and D. Coker. Cracks faster than the shear wave speed. *Science*, 284:1337–1340, 1999. <https://doi.org/10.1126/science.284.5418.1337>.

- [16] S. Nielsen, J. Taddeucci, and S. Vinciguerra. Experimental observation of stick-slip instability fronts. *Geophysical Journal International*, 180:697–702, 2010. <https://doi.org/10.1111/j.1365-246X.2009.04444.x>.
- [17] A. Schubnel, S. Nielsen, J. Taddeucci, S. Vinciguerra, and S. Rao. Photo-acoustic study of subshear and supershear ruptures in the laboratory. *Earth and Planetary Science Letters*, 308:424–432, 2011. <https://dx.doi.org/10.1016/j.epsl.2011.06.013>.
- [18] S. Latour, A. Schubnel, S. Nielsen, R. Madariaga, and S. Vinciguerra. Characterization of nucleation during laboratory earthquakes. *Geophysical Research Letters*, 40:5064–5069, 2013. <https://doi.org/10.1002/grl.50974>.
- [19] A. Gounon, S. Latour, J. Letort, and S. El Alem. Rupture nucleation on a periodically heterogeneous interface. *Geophysical Research Letters*, 49, 2022. <https://doi.org/10.1029/2021GL096816>.
- [20] F. Paglialunga, F. Passelègue, S. Latour, A. Gounon, and M. Violay. Influence of viscous lubricant on nucleation and propagation of frictional ruptures. *Journal of Geophysical Research: Solid Earth*, 128, 2023. <https://doi.org/10.1029/2022JB026090>.
- [21] B. Fryer, M. Lebihain, C. Noël, F. Paglialunga, and F. Passelègue. The effect of stress barriers on unconventional-singularity-driven frictional rupture. *Journal of the Mechanics and Physics of Solids*, 193, 2024. <https://doi.org/10.1016/j.jmps.2024.105876>.
- [22] O. Ben-David and J. Fineberg. Static friction is not a material constant. *Physical Review Letters*, 106, 2011. <https://doi.org/10.1103/PhysRevLett.106.254301>.
- [23] S. Xu, E. Fukuyama, F. Yamashita, H. Kawakata, K. Mizoguchi, and S. Takizawa. Fault strength and rupture process controlled by fault surface topography. *Nature Geoscience*, 16:94–100, 2023. <https://doi.org/10.1038/s41561-022-01093-z>.
- [24] F. Yamashita, E. Fukuyama, S. Xu, K. Mizoguchi, H. Kawakata, and S. Takizawa. Rupture preparation process controlled by surface roughness on meter-scale laboratory fault. *Tectonophysics*, 733:193–208, 2018. <https://doi.org/>.
- [25] K. Uenishi and J. Rice. Universal nucleation length for slip-weakening rupture instability under nonuniform fault loading. *Journal of Geophysical Research*, 108, 2003. <https://dx.doi.org/10.1029/2001JB001681>.
- [26] S. Guérin-Marthe, S. Nielsen, R. Bird, S. Giani, and G. Di Toro. Earthquake nucleation size: Evidence of loading rate dependence in laboratory faults. *Journal of Geophysical Research: Solid Earth*, 124:689–708, 2019. <https://doi.org/10.1029/2018JB016803>.
- [27] P. Berthoud, T. Baumberger, C. G’Sell, and J. Hiver. Physical analysis of the state- and rate-dependent friction law: Static friction. *Physical Review B*, 59:14313–14327, 1999. <https://doi.org/10.1103/PhysRevB.59.14313>.

- [28] T. Baumberger, P. Berthoud, and C. Caroli. Physical analysis of the state- and rate-dependent friction law. II. Dynamic friction. *Physical Review B*, 60:3928–3939, 1999. <https://doi.org/10.1103/PhysRevB.60.3928>.
- [29] J. Ampuero and A. Rubin. Earthquake nucleation on rate and state faults: Aging and slip laws. *Journal of Geophysical Research*, 113, 2008. <https://dx.doi.org/10.1029/2007JB005082>.
- [30] P. Bhattacharya, A. Rubin, T. Tullis, N. Beeler, and K. Okazaki. The evolution of rock friction is more sensitive to slip than elapsed time, even at near-zero slip rates. *Proceedings of the National Academy of Sciences*, 119, 2022. <https://doi.org/10.1073/pnas.2119462119>.
- [31] C. Mei and L. Wang. Velocity dependence of rate-and-state friction in granular fault gouge and implications for slow-slip events. *Journal of Geophysical Research: Solid Earth*, 129, 2024. <https://doi.org/10.1029/2024JB029393>.

Reviewers #1 and 2

12. *Thank you for the revision in response to our comments. We think most part becomes clear now. We still have the following comments and suggestions to make the manuscript more understandable.*

Response: We thank the reviewers for their time and rigor. We feel the manuscript, and in particular the natural earthquakes section, has greatly benefited from their critique.

13. *Comment # 12: We visited Whitcomb et al (1973) and thought it inappropriate to cite these data. Whitcomb et al (1973) analyzed the 1971 San Fernando earthquake (M6.4) and compared with other data from Aggarwal et al (1973) and Semenov (1969) with a conversion of magnitude to characteristic length. The serious issue is that this characteristic length in Whitcomb et al (1973) could not be considered as the nucleation length but the mainshock fault area estimated from aftershock distribution of the San Fernando earthquake. Other lengths were converted from magnitudes using an empirical relation by Whitcomb et al. using published data. Therefore, the citation of the data in Whitcomb et al (1973) should not be used. If*

the authors still wish to use these data, please detail how Dtc and lc are estimated. And these values should be listed in Table S1.

Response: We have completely removed Whitcomb et al., 1973 from Figure 4. Please see Figure 4 and Figure 4's caption.

14. *And the followings are what we noticed after reading the revised manuscript. Since many parameters are used and some of the definitions appeared after their usage, I would recommend to tabulate the definitions of the parameters as a supplementary, which would help the audience.*

Response: We have added a table at the end of the manuscript defining the main variables used. We did not list every single variable (e.g., we did not add the strain gauge voltage readings, U); instead, we defined the most important variables and in particular ones that are used outside of the part of the text where they are first defined. Please see Table S2. We have also added this phrase so that we reference all tables in the text:

"See Extended Data Tab. S2 for a tabulation of the main variables." Please see line 880.

15. *Figure 4a. The label "Foreshock size, Afs [m/sec^2]" is misleading. It should be "Averaged foreshock acceleration, Afs [m/sec^2] as described at L386.*

Response: This has been changed. Please see Figures 3, 4a, and S4.

16. *Figure 4b. It seems convenient for the readers to swap the axes. Vertical axis is Dt and horizontal axis is lc .*

Response: This change has been made. Note that this changed which earthquakes we indicated with labels due to the reorientation making certain labels inconvenient. See Figure 4b.

17. *Figures 4a-b. The quasi-static nucleation phases reported in the referenced laboratory earthquakes (Ohnaka and Shen, 1999; Latour et al., 2013; McLaskey, 2019; Yamashita et al., 2021; Xu et al., 2023) do not appear to have been initiated by foreshocks. While I acknowledge the authors' reply to Comment # 11 that the nucleation discussed here is not necessarily caused by foreshocks, the laboratory data shown in Figure 4, which are used to discuss the scaling of nucleation duration, should be more clearly described in the main text.*

Response: We agree that these events do not appear to have been initiated by (distinguishable) foreshocks. We have therefore adjusted the text to reflect this. We now remove the text regarding these laboratory events in relation to the idea that large slip is accumulated when foreshocks are larger:

“This trend is not only evident in the present experimental dataset, but also appears consistent with observations from ~~larger-scale laboratory experiments and~~ natural earthquakes” Please see line 123.

We further indicate that the laboratory events are not preceded by a foreshock in the caption of Figure 4. Note that this addition resulted in the caption surpassing the 300-word limit. We have therefore made some light modifications to meet the word limit:

“Transient sliding velocity predicts nucleation duration and length. a, Prolonged nucleation phases accrue more slip. Laboratory and natural earthquakes are included with black dots. The foreshock amplitude is given by the color bar. ~~The error for the slip during nucleation~~ Slip-during-nucleation error is estimated by evaluating the slip within 0.1 ± 0.05 msec of v_r reaching 10 m/sec. **b,** The nucleation ~~duration, Δt_c~~ length, ℓ_c , as a function of the nucleation ~~length, ℓ_c~~ duration, Δt_c . The error of ~~both ℓ_c and Δt_c~~ was found by reassessing both quantities with $v_r = 20$ m/sec. Natural and laboratory earthquakes are plotted with black dots ~~(Whitecomb et al., 1973 data in grey)~~. The local value of normal stress is given by the color bar. In a, b the laboratory events are not preceded by a distinguishable foreshock. **c,** The nucleation duration as a function of V_{\min} . Predictions from the EoM are shown in grey dashed lines based on state-evolution aging law slip distances, L , of 0.03, 0.1, and 0.3 microns, $\frac{a}{b} = 0.55$, and an ambient sliding velocity of 10^{-6} m/sec. The color bar indicates the quasi-static half length at nucleation. **d,** The estimated duration and transient sliding velocity of the nucleation phases of natural earthquakes. These data are estimated from inverted GNSS data or repeater earthquakes. However, Papanoa 2015 (pink) was a slow slip event not triggered by a foreshock. İzmit 1999 (purple) is a debated case whose repeaters may or may not be linked to aseismic slip. Time series data for Yangbi 2021 and Valparaíso 2017 (white) are not available such that a simple average is used to estimate V_{\min} . EoM fits in dashed lines for state-evolution aging law slip distances of 0.3, 1.0, and 3.0 mm using $\frac{a}{b} = 0.75$ and an ambient sliding velocity of 10^{-12} m/sec. Fits with $L = 1.0$ mm and $\frac{a}{b} = 0.30, 0.75, 0.90$ are shown in pink.”

Finally, we distinguish between laboratory events without a foreshock and natural events with a foreshock when discussing the correlation between nucleation time and length:

“This trend holds not only in our experiments, but also in larger-scale (~ 1 m) rock friction experiments without a distinguishable foreshock and for quasi-statically

slipping patches of natural earthquakes, where the onset of nucleation is identified by a foreshock.” Please see line 208.

18. *Supplementary Table S1: To improve clarity and facilitate cross-referencing, it would be preferable to separate natural and laboratory earthquakes under distinct column headers and list the authors’ names and publication year for laboratory earthquakes.*

Response: This change has been made. Please see Table S1.

Reviewer #3

19. *Having read the revised paper and the authors’ response, I’m satisfied that my comments have been addressed. I have the following minor, comments, mostly related to the changes from the first draft.*

Response: We thank the reviewer for their insight and time. We feel their comments have helped us be more precise and balanced in our wording.

20. *Fig. 3: I’m uncertain how the velocities are determined in panel (a). The new caption for panel (c) says “After the foreshock, the slip evolves as an exponential function of time (Fig. S2), yielding the initial minimum sliding velocity of the patch at the onset of nucleation.” I found this a little hard to understand; the deleted portion “This evolution is fit and its derivative is evaluated just after the foreshock, yielding ...” makes it much clearer, and I would even write “This assumed evolution ...”. But this still doesn’t explain Fig. 3a. From the slip-vs-time data in Fig. S2 e–f, it’s obvious that one can’t choose the time of minimum velocity by eye. Fig. S2 panels g–i show the numerical derivative of this slip, but it looks very noisy, as expected when differentiating a real time series (I imagine that a point-by-point differentiation could lead to negative velocities sometimes, consistent with the data gaps in Fig. S2h). But the velocity prior to 0.2×10^{-1} s in Fig. 3a, which clearly includes the local maximum in velocity prior to deceleration, looks very smooth. It looks like the displacement time series must have been smoothed prior to differentiation. If so, this procedure should be outlined. Including the details in the Fig. S2 caption might be enough.*

Response: We have added back the previous phrase to Figure 3’s caption. Note that we did have to cut out an additional phrase in order to meet the 300 word limit for the captions:

“Foreshock size predicts the transient sliding velocity of the nucleating patch. a, The sliding velocity of the three events shown in Fig. 2. The

black symbol shows $v_r = 10$ m/sec. Note the initial sliding velocity due to the foreshock, its reduction after the foreshock, and the reacceleration up to dynamic rupture propagation. The foreshocks of these three events have estimated slips, δ_a , of 0.72, 0.99, and 1.86 microns (Extended Data Fig. S2e) in order of increasing foreshock size, A_{fs} (Methods). **b**, EoM simulation runs, characterized by an asperity slip, δ_a ; $\Delta f_0 = 0$. The simulations with δ_a most closely corresponding to the experimental events in **a** are shown with dark lines with $v_r = 10$ m/sec indicated by a star (Extended Data Fig. S5). **c**, After the foreshock, the slip evolves ~~as an exponential function of~~ exponentially with time (Extended Data Fig. S2). This assumed evolution is fit and its derivative is evaluated at the foreshock, yielding the initial minimum sliding velocity of the patch at the onset of nucleation. The slip accrued during the foreshock, δ_a , is taken as the jump in slip at the moment of the foreshock (Extended Data Fig. S2). Larger foreshocks yield larger ~~V_{min} transient sliding velocities.~~ This result is predicted by the EoM, considering $\frac{a}{b} = 0.55$, state-evolution slip distances, $L = 0.03, 0.1, 0.3$ microns, and ambient fault sliding velocity, $V_0 = 10^{-6}$ m/sec. The dashed lines represent EoM predictions for the case with non-negative overstress. Events with negative overstress are predicted to fall below these lines. For **a** and **c**, the foreshock amplitude is given by the color bar. The error of δ_a is estimated as the difference between the maximum and mean displacement recorded within 0.5 msec after the foreshock, with the lower bound limited to 50% of the measured value. The error of V_{min} is estimated by propagating the 95% confidence interval of the exponential fit to the slip, time data.”

We have also added details to Fig. S2’s caption as requested by the reviewer:

“The slip is assumed to evolve exponentially in time. This assumed evolution is fit and the derivative evaluated at the time of the foreshock, yielding V_{min} .”

21. *The new question (i) on line 137 (line numbers from the “tracked changes” version): I’m not convinced that this improves the paper, at least as currently written. The question is: “Are all earthquakes preceded by a quasi-static nucleation phase?”, and the one-word answer is ‘No.’. This is then qualified as “observable quasi-static nucleation is not a prerequisite for dynamic rupture”, which I agree with, but the qualifier “observable” isn’t part of the question. The answer goes on to describe the regime where a “foreshock” leads directly to dynamic rupture. But in this case the expectation is that the “foreshock” (although no longer really a foreshock) itself had a quasi-static nucleation phase. I think an acceptable fix is to just get rid of the “No,” at the start of this sentence.*

Response: We have made the change the reviewer suggested.

“Are all earthquakes preceded by a quasi-static nucleation phase? ~~No,~~ While most

of the dynamic events in these experiments exhibit a quasi-static nucleation phase, observable quasi-static nucleation is not a prerequisite for dynamic rupture.” Please see line 241.

22. *line 149: “under sufficiently stressed conditions”*

Response: Changed.

“Larger foreshocks shrink the time scale and reduce the stress criticality required for nucleation. The largest foreshocks can themselves merge into a main shock under sufficiently stressed conditions.” Please see line 258.

23. *lines 148-152: This is a really long sentence.*

Response: We have divided this sentence into two.

“Yet, observations of quasi-static nucleation phases in foreshock to main shock sequences are relatively rare in nature, ~~1.~~ This is likely because the impulses of most foreshocks are either too insignificant to lead to nucleation after the initial slowdown (Regime 1), too large to not directly merge into a main shock (Regime 3), or because quasi-static phases are simply too difficult to identify in observations.” Please see line 260.

24. *Fig. 4 caption: “The error for the slip during nucleation is estimated by evaluating the slip within 0.1 msec of v_r reaching 10 m/sec.” Meaning ± 0.1 msec?*

Response: In fact we mean ± 0.05 msec. We have updated the caption:

“The error for the slip during nucleation is estimated by evaluating the slip within ± 0.05 msec of v_r reaching 10 m/sec.”

25. *line 158: “Based on these considerations, we suggest the use of the word foreshock should be restricted to earthquakes which either initiate or tangibly influence the quasi-static nucleation phase of another, larger earthquake.” I think this is not a good idea, and that whether a good idea or not, it has approximately zero chance of being adopted by the seismological community. I think it is not a good idea because it requires evaluating a potential foreshock using a model that could easily be incorrect or incomplete. I think its chance of being implemented is near zero because in many cases the data necessary to evaluate the potential foreshock (location relative to the mainshock; focal mechanism) may be unavailable or inadequate.*

Response: We agree with the reviewer and have removed these two sentences.

~~“Based on these considerations, we suggest the use of the word foreshock should be restricted to earthquakes which either initiate or tangibly influence the quasi-static nucleation phase of another, larger earthquake. This definition would exclude earthquakes occurring during preslip-like but not necessarily those during rate-dependent cascade-up nucleation phases.”~~ Please see line 263.

26. *line 335: I think this explanation is an explanation that doesn't really explain. The real question is if/why the “dynamic range” of potential observable foreshocks on PMMA samples, rock samples, and natural faults differ, and their statistical likelihood, meaning the relative abundance of heterogeneities on different length scales. But nothing the authors say here strikes me as wrong, so they can keep it if they like.*

Response: We understand the reviewer’s point. To briefly recall some points to aid in the response below, we state, from scaling arguments, that the “size” of the foreshock in terms of its influence on nucleation dynamics is related to $\frac{\delta_a}{L}$, or the slip during the foreshock divided by the characteristic slip distance. As L is larger in rock samples, the slip during any foreshocks must be 10-100 times larger in rock samples than in PMMA to have a similar influence on nucleation dynamics compared to what we see in our data set. Yet, these slip values correspond to *main-shock* slip events seen during laboratory experiments (e.g., McLaskey 2019 [1]). Therefore, any foreshocks (with smaller slip values than this) should not be expected to have a significant influence.

As the reviewer suggests, this still misses one piece of the puzzle. Why are foreshocks in rock samples so “small”? We believe that the reason foreshocks in large-scale (20 cm +) laboratory rock samples do not exhibit enough slip to sufficiently influence nucleation dynamics has to do with the larger shear modulus in rock compared to PMMA (≈ 24 GPa and 1.24 GPa, respectively). Considering that most large-scale apparatus like these are limited in the amount of stress they can apply (typically 5-20 MPa, e.g., McLaskey 2019 [1] and our experiments), the amount of slip associated with a given stress drop is much smaller in rock samples than in PMMA as slip can be expected to scale as $\sim \frac{\Delta\tau}{\mu_r}$. As the available stress drop is similar, the larger shear modulus means the slip in rock samples cannot be much larger than in PMMA as it would be required to be for the foreshocks to significantly influence nucleation dynamics (i.e., have a large enough $\frac{\delta_a}{L}$).

In fact, if one uses the scalar seismic moment of a circular fault [2, 3], $\Delta\tau = \frac{7}{16} \frac{M_0}{r^3}$, where $\Delta\tau$ is the stress drop, $M_0 = \mu\pi r^2 \delta_a$ [4] is the seismic moment of the foreshock, and r is the source radius, one can estimate the source radius necessary to produce the kinds of slips that would be required in rock for foreshocks to influence

nucleation dynamics as $r = \frac{7\pi}{16}\mu\frac{\delta_a}{\Delta\tau}$. In order to then achieve the 10-100 microns of slip required to produce a significant $\frac{\delta_a}{L}$, the source radius, r , would need to be (assuming $\delta_a = 10$ microns) 0.33 m for a stress drop of 1 MPa (a typical stress drop for a main shock in McLaskey 2019 [1]). This value of rupture radius would not be considered a foreshock. In order for a foreshock radius to be approximately 1 cm, the stress drop (assuming $\delta_a = 10$ microns) would need to be 33 MPa, a value which is clearly unrealistic.

Additional changes

27. We went back and reverified how we selected the aseismic slip data used to make Figure 4d. To improve consistency between the different earthquakes, we slightly adjusted the picking window for Iquique 2014 and Ibari-Oki 2008. This resulted in anecdotal changes to these two points in Figure 4d.

28. We updated equation 6 as we realized k was defined twice. We replaced it with N_a :

$$A_{fs} = \sqrt{\frac{1}{N_a} \sum_{i=1}^{N_a} \left(\frac{r_i}{0.01} A_i^{\max} \right)^2} \quad (1)$$

29. Similarly, we changed the gauge amplification and gauge factors, G_{amp} and G_f , to J_{amp} and J_f to avoid overlap with fracture energy, G_c . Please see Equation 2.

30. We have added a sentence to the Methods section so that the reader has an indication of where Equation (17) came from:

“The asymptotes follow by considering the time required to achieve the maximum stable crack length, ℓ_∞ , at the minimum rupture velocity, $v_{r,\min}$, itself dependent on $V_{\text{eff},\min}$ by Equation (15).” Please see line 929.

31. We corrected a citation which was referencing the wrong Kato paper:

“**Iquique 2014** exhibited repeater earthquakes starting around 270 days prior to the mainshock (42).” Please see line 673.

32. We corrected the year of the Papanoa earthquake in Figure 4d. “Papanoa 20142015”

Yours sincerely on behalf of all authors,

Barnaby Fryer

References

- [1] G. McLaskey. Earthquake initiation from laboratory observations and implications for foreshocks. *Journal of Geophysical Research: Solid Earth*, 124:12882–12904, 2019. <https://doi.org/10.1029/2019JB018363>.
- [2] K. Aki. Scaling laws of seismic spectrum. *Journal of Geophysical Research*, 72: 1217–1231, 1967. <https://doi.org/10.1029/JZ072i004p01217>.
- [3] R. Madariaga. Seismic Source Theory. In *Treatise on Geophysics, Second Edition*, chapter 4, pages 51–71. Elsevier B.V., 2015.
- [4] K. Aki. Generation and propagation of G waves from the Niigata Earthquake of June 16, 1964. Part 2. Estimation of earthquake moment, released energy, and stress-strain drop from the G wave spectrum. *Bulletin of the Earthquake Research Institute*, 44: 73–88, 1966.

Review report for “From foreshock to mainshock: Transient sliding velocity sets nucleation time” by Barnaby Fryer, Dmitry Garagash, Mathias Lebihain, and François Passelègue.

In this manuscript, the authors investigated how stick slip events initiate and grow for the dynamic rupture during PMMA friction experiments, aiming to understand the nucleation process of large earthquakes. They considered that an impulse generated by a foreshock initiates the nucleation slip and modulate the time required for a fault to evolve into dynamic rupture. The principal findings are as follows:

- An impulsive perturbation generated by a foreshock directly influences the subsequent transient sliding velocity, which controls the nucleation duration.
- This behavior is supported by the rate-dependent equation of motion (EoM) framework formulated by Garagash, indicating that both overstress and foreshock impulse jointly shape rupture evolution.

These results suggest that impulsive perturbations at the onset of quasi-static nucleation, typically absent from canonical preslip models, may serve as a first-order control on nucleation duration in natural earthquakes. Accordingly, the study may expand our current understanding of the realistic nucleation process of both seismic ruptures and more general frictional failure processes.

This manuscript consists of comprehensive views drawn from experiments, theory, and natural observations. The comparison between experimental/theoretical insights and seismological observations is leveraged to propose scaling relationships between foreshock size, nucleation duration, and transient sliding velocity.

The argument is quite impactful for understanding earthquake nucleation, and it is innovative to address this problem through combined experimental and theoretical approaches. I agree with the interpretations of the experimental data by theoretical analysis of rupture nucleation based on the EoM framework. However, I have several concerns regarding the assumptions of the modeling, analysis of the experimental results, and the interpretation of natural earthquakes. Therefore, I think significant revisions could be needed if this manuscript is considered to be accepted for publication. I hope that the comments listed below will be useful in further improving the manuscript.

Major Comments:

- Logical collapse of the nucleation mechanism

In the proposed scenario of the nucleation model proposed here, the stimulation by the stress impulse seems a key point, which the authors thought to be generated by the foreshock. This stimulation initiates the nucleation of the rupture and control the nucleation duration. However, if this stimulation only comes from a foreshock, one cannot explain the cases when no foreshock was observed at the beginning of rupture nucleation. In addition, that foreshock itself should have its own nucleation process which should be triggered by another small foreshock following the authors' scenario. which apparently requires another nucleation mechanism and the proposed nucleation mechanism is considered as a case under very limited conditions. Then, the audience would wonder how to apply the proposed method to apply the natural earthquakes or other frictional phenomena. I think it quite important to clarify the condition where the proposed model can apply.

- Uncertain evaluation of quasi-static length on natural earthquakes

The authors listed the quasi-static length, quasi-static slip, and transient sliding velocity V_{min} on the 10 natural earthquakes from the published literature as listed in Supplementary table S1. The coherence between these parameters supports the argument of the correlation between the nucleation duration and quasi-static length, and the transient sliding velocity and nucleation duration observed in the laboratory experiment. However, the quasi-static length is generally difficult to estimate, even in the order sense, and it looks like different measurements are mixed in the table. At least, the authors should describe how these values are estimated and evaluate their uncertainties in the supplementary.

- I. The 1999 Izmit earthquake is interpreted in the manuscript as having a quasi-static nucleation length of approximately 300 m, based on the documentation by Bouchon et al. (2011). However, this value corresponds to the size of the foreshock-ruptured patch, not the quasi-static nucleation length. Subsequent analysis by Ellsworth and Bulut (2018) reported even smaller dimensions (<100 m) for these foreshock patches (Figure R1). Moreover, Ellsworth and Bulut explicitly stated that “The absence of any other repeating foreshocks argues against a process driven by aseismic slip (fault creep) as envisioned previously.” Thus, there is no observational evidence linking the foreshock patch size to a quasi-static nucleation process for this earthquake.

Figure R1. Stress changes due to foreshocks on the 1999 Izmit earthquakes (Ellsworth and Bulut, 1999).

- II. The 2014 Papanoa earthquake is discussed with reference to Radiguet et al. (2016). However, this study did not report any foreshocks; it only documented the preceding slow slip event (SSE). In addition, the metric used in the manuscript to infer a quasi-static nucleation length of 250 km from Figure R2 appears uncertain. The scale of the SSE does not directly translate into a quasi-static nucleation length, and the interpretation presented in the manuscript lacks sufficient justification.

Figure R2. Spatio-temporal distribution of SSE on the 2014 Papanoa earthquake (Radiguet et al., 2016).

- III. The locations of the Mw 6.2 foreshock and the subsequent mainshock of the 2016 Kumamoto earthquake are shown in Figure R3 (Kato et al., 2016). To my knowledge, it remains controversial whether these events occurred on the same fault plane. While I agree that this sequence is broadly

analogous to the scenario proposed in the present study, where the foreshock may promote quasi-static slip leading to mainshock initiation, a more careful discussion is required to evaluate the specific role of the foreshock in this earthquake.

Figure R3. Cross-sections of spatio-temporal evolution of the seismicity towards the mainshock (Kato et al., 2016).

Overall, I find that the evaluation of the quasi-static nucleation process is overly generalized when compared with the laboratory and theoretical insights into foreshock-to-mainshock sequences. It would greatly strengthen the manuscript if the authors could refine the discussion presented in the final section.

In addition, I was unable to identify the original data sources for the natural-earthquake markers plotted in Figure 4b (See annotation of Figure R4). It would be helpful if the authors could explicitly specify the datasets used to construct these points.

Figure R4. Taken from Figure 4b in the manuscript.

- Material dependent frictions: PMMA bare contact has similar features to those of rock and fault gouge?

The stick-slip experiments were conducted on PMMA samples with bare contact and without gouge. I think this condition is crucial for reproducing a quasi-static nucleation process that is initiated by a foreshock. In contrast, experiments using rock samples have demonstrated a different scenario, in which preslip develops around asperity patches and subsequently generates foreshocks as a byproduct (e.g., McLaskey and Kilgore, 2013). While I agree that such foreshocks can influence the transient sliding velocity and the nucleation duration, the correlation between these quantities is far less clear in rock experiments than in the present study. The presence of fault gouge also affects foreshock generation (Yamashita et al., 2021). Therefore, although I agree that the laboratory observations are broadly consistent with the EoM framework, extending these findings to natural earthquakes requires further discussion that accounts for differences in frictional conditions.

Other Comments:

1. First line of the abstract: The statement "*Coseismic rupture is commonly preceded by stable fault slip, which may accelerate into dynamic rupture as the slip front reaches critical velocities*" may not be acceptable for some readers, given that the nature of the nucleation process remains controversial. Cascade-up scenarios without substantial quasi-static preslip are also plausible for large earthquakes (e.g., Ide, 2019). Therefore, the introductory assertion may need to be reframed to acknowledge this variability.
2. I think the role of foreshocks in earthquake nucleation is extensively discussed in McLaskey (2019). It would be helpful for the authors to clearly distinguish the foreshock-driven nucleation concepts presented in that work from the framework used in the present study, particularly by emphasizing the role of foreshocks in initiating quasi-static slip that evolves toward the mainshock.
3. Figure 1 Caption: Please provide more detailed information on "videogram". What is the physical quantity the videogram shows? Which is bigger in dark/light color? better to provide color scales.
4. Figure 1b: The shear stress τ appears to exceed the normal stress σ . Does this imply that the friction coefficient becomes exceptionally large (>1.0) under these conditions? If so, this point is directly related to my major comment #3 regarding material differences and their implications for the nucleation process.
5. Figure 1c-e: Please add the amplitude scale for accelerograms.
6. Figures 1c-e & S1: I was wondering how the authors determine the nucleation time. It was difficult for me to identify the starting point of the nucleation visually from the videogram since the color change was quite tiny there. Did the authors confirmed if no slip generated before the foreshock (i.e., the foreshock occurred slightly earlier than the start of nucleation)? And did the foreshock occurred exactly the same location as that the nucleation started, or it occurred slightly different location? Such information should be important to support the authors scenario proposed here.
7. Figure 2: Each point corresponds to measured data? Since original data was sampled at 2MHz, is this data averaged at constant slip (or time) window? Can you show the location of the sensors? It seems better to use smaller symbols to see more details.
8. Line 45-46: According to Uenishi and Rice (2003), a sharper weakening rate leads to a shorter critical nucleation length. However, this does not necessarily imply that the weakening rate directly controls the nucleation duration, which is influenced not only by the weakening rate but also by the loading

rate. Therefore, this section requires further clarification to avoid overinterpreting the role of the weakening rate in determining the temporal evolution of nucleation.

9. Figure 3c: Several data points exhibit one-sided error bars. While asymmetric error bars are understandable given the log scale, the meaning of the one-sided representation is unclear. Please clarify how these uncertainties were calculated and why a one-sided error bar is appropriate for these particular data points.
10. Line 107: The statement “The advancing crack tip draws energy both from the foreshock and from the inherent stress state along the fault” is intuitively understandable. However, some readers may be concerned by this phrasing, because a foreshock dissipates stored elastic energy and therefore cannot literally serve as an energy source for the system. It would be better to rephrase this sentence to avoid implying that foreshocks contribute energy rather than perturbing the stress field.
11. Line 142: The cascade-like process is generally associated with nucleation without a significant quasi-static slip phase, as illustrated in the schematic by McLaskey (2019). Therefore, it may be helpful to clarify how the cascade-like behavior described here relates to the quasi-static nucleation framework used in this study.
12. Although it would require additional effort, it would strengthen the manuscript if all videograms with the annotation like Figures 1d could be summarized into a single supplementary figure to reinforce the argument regarding the relationship between foreshock size and nucleation duration.

I like this paper. It attempts to make sense of laboratory friction experiments in which the time and length scales of contained earthquake nucleation are quite variable, despite the frictional properties remaining nominally constant (e.g., as suggested by Fig. 2). To a large extent I think it succeeds. The main conclusion is that the foreshock(s) that often precede laboratory earthquakes provide an impulsive stress change to the fault, a stress change that manifests itself as a perturbed sliding velocity that ultimately affects the time and length scales of nucleation. Similar observations have been made previously, but not really addressed theoretically; I imagine this behavior was considered too complicated to analyze theoretically. Numerous simplifying assumptions are made in the theoretical treatment, but I believe it's reasonable to give the first paper to really tackle this issue some slack. The theoretical results allow the authors to parameterize the nucleation behavior in terms of the minimum slip speed reached on the fault between the times of the (largest?) foreshock and the ensuing main shock, and this treatment does a respectable job of collapsing the lab data onto interpretable trends. The authors go on to apply the same theoretical framework to natural earthquakes, and it appears that they follow the same trend to larger length and time scales. Even more assumptions are required to interpret the natural earthquakes in this way (mostly in determining the minimum fault slip speed, achieved just after the foreshock), but the attempt certainly provides food for thought, and appears to be successful, so it adds a lot to the paper.

In a bit more detail: The theoretical analysis relies on previous work by one of the authors (Garagash, 2021), who developed an “equation of motion” for the tip of the nucleation/rupture front, by matching the rupture-velocity-dependent stress intensity factor K at the tip to the fracture energy required for growth of the rupture. In the absence of foreshocks, K for a steady-state velocity-weakening surface increases as the nucleation zone grows in size, while a foreshock provides a separate contribution to K that decreases as the nucleation zone grows. The result is that 3 nucleation regimes arise: (1) The foreshock is so weak that the nucleation attempt dies without generating a main shock; (2) The fault slip speed initially decreases due to the reduction in the foreshock contribution to K , but subsequently accelerates as the background, velocity-weakening contribution takes over; or (3) Slip accelerates monotonically and rapidly toward failure. Most of the paper deals with nucleation in the presence of foreshocks, regime (2), in which larger foreshocks lead to larger minimum slip speeds, shorter nucleation times, and smaller nucleation lengths. I think a fair amount of insight was required to recognize that the important behaviors could be parameterized in terms of the minimum fault slip speed, which is influenced by both the foreshock size and the fault pre-stress. One of the strengths of the paper, I think, is that regardless of the details of the equations used, the idea of a competition between two contributions to K , one of which increases as the nucleation zone grows and one of which decreases, seems robust, and a sensible way of thinking about the large differences in nucleation time on what is nominally the same fault.

I think the combination of a realistic model for the laboratory observations of earthquake nucleation, and a plausible extrapolation to natural earthquakes, makes this certainly suitable for publication in Nature. The model makes sense of observations that previously have not really been addressed. Some fitting parameters/model simplifications are involved, so it's hard to guarantee that the model is the appropriate one, but I'm fine with that. Larger rate-state slip distances are required for the natural examples, using the same model, but that's plausible. I have the following, mostly minor, comments, in chronological order:

Figure 1 caption: I think this should read “lengths, widths, and heights of $(x, y, z) = (40, 10, 1)$ cm”, assuming that y is height and z is width (front-to-back).

The short-duration nucleation event (0.6 msec, triangles) in Figure 2 looks surprising, in that there appears to be very little stress drop before the nucleation phase is over/rupture speed of 10 m/s is reached. There are some triangles at higher and lower stress levels than the black triangle at 0.6 msec, so perhaps that variation is indicative of noise in the measurements, but from the symbol colors there is good temporal consistency between the measurements. Making this harder to understand is that the data seem to show a non-intuitive relation between time and slip: The darker green colors, which are earlier in time, occur at greater slip distances than the pale green colors, which occur later. So taken literally, the earliest measurements (dark green) show more slip and smaller stresses, followed by (light green) less slip and larger stresses, with the

end of the nucleation phase (black symbol) having larger slips but intermediate stresses. Is this apparent inconsistency the result of stress, slip, and rupture speed being point measurements made at different locations? In any case, some comment might be in order, because otherwise it's hard to understand what is going on.

On line 40 I would add the word “detectable”, as in “without the intermediate stage of detectable quasi-static slip-front propagation”.

The Figure 3 caption refers to the foreshock slip distances δ_a , and cites the Methods for determining them. But I can't find how this was done; perhaps I read the relevant Methods section too quickly. There is a “Foreshock magnitude” section that gives equation (S6) for how the averaged acceleration A_{fs} is determined, but I don't see how that is converted to slip distance. Source-station distances in (S6) are normalized by a reference sphere of radius 1 cm; is that an estimated source size, and the same size is used to convert accelerations to slip distances? But that would also require an estimate of slip speed. Or is that 1 cm the footprint of the accelerometer? I confess I didn't follow up with reference 44. Or does the estimate of δ_a just come from the match between the data in Fig. 3a and the model runs in 3b, with the relation between ΔT (the centrally-applied force on the crack from the foreshock) and δ_a coming from equation (S10) and the empirical C ? It would be good to clarify this.

line 136: Again, I would insert the qualifier “observable” here: “This suggests that while an observable nucleation phase often occurs, it is not a strict prerequisite for rupture”.

The data for natural earthquakes in Figure 4d, plotting the estimated nucleation duration (easy to determine, when defined as the time between the largest foreshock and the main shock) as a function of the minimum slip speed (presumably very difficult to determine), seems almost too good to be true. There's a statement in the Methods that “these data rely on either inversion of the fault slip based on geodetic data[,] or the supposition that the slip during repeater earthquakes is representative of the aseismic slip of the patch”. In addition, use is made of the modeling result that slip accumulates exponentially with time, so that data at multiple times can be fit with a straight line on a semi-log plot, as shown in Figure S4. This information is too important to leave for the Methods only. Right now there's just a statement in the main text that “Additionally, part of the discrepancy may stem from the uncertainty in estimating transient sliding velocities for natural earthquakes; our values, based on indirect constraints, likely overestimate the V_{min} during nucleation”. I don't think there's even a citation to Figure S4. Please include something like the quoted sentence from the Methods, plus the theoretical result and a mention of S4, in the main text.

Foreshocks and nucleation length scaling: Starting on line 320 of Methods is written “The coexistence of foreshocks and a larger nucleation front can be rationalized by considering the scale dependence of nucleation. At the scale of micron-sized asperities, local stress concentrations lead to significantly higher effective normal stresses than the macroscopically-applied value. Because the theoretical nucleation size scales inversely with normal stress, ...”. I'm wondering if two concepts are being conflated here. I could easily imagine that at small scales, but still encompassing many asperities, rate-state properties, normal stress, and (for non-planar surfaces) the slip-dependence of normal stress can be variable, and produce foreshocks on length scales smaller than the main shock nucleation zone. But the statement “At the scale of micron-sized asperities, local stress concentrations lead to significantly higher effective normal stresses than the macroscopically-applied value” makes it seem that the intent might be to appeal to local stresses at the single-asperity scale that exceed the remotely-applied normal stress by one over the fractional contact area, where the latter could be less than 1%, which would then drop the nucleation zone size by a factor of more than 100. I'm not sure the standard RSF equations can be applied at this scale, but are the authors proposing that the laboratory foreshocks represent rapid slip of a single asperity? Typically we think of the characteristic asperity size as being approximately L in the RSF equations, which would be about $0.2 \mu\text{m}$ for these experiments. But the inferred foreshock slip distances are several times this value, which is difficult to reconcile with single-asperity-scale sources (contact stresses might be 10% of the shear modulus, so a complete stress drop to zero on an asperity might give rise to displacements of a few times 10% of the asperity dimension, or less than

say $0.05 \mu\text{m}$, which is too small, and complete stress drop is outside the standard RSF equations anyway). So please clarify that (I think) you are not proposing that the foreshocks represent slip of a single asperity, with friction on that single asperity being governed by the standard RSF equations. This might be as simple as writing “At the scale of multiple micron-sized asperities, local stress concentrations...”.

Slip- vs. Aging-law simulations: Line 366 of Methods says “Considering the apparent invariance of the weakening rate W in our experiments (Fig. 2), we adopt the aging law for the remainder of the analysis”. But this argument could equally well favor the Slip law – the Slip law predicts a weakening rate that is larger for larger peak-residual stress drops, not seen in Fig. 2, while the Aging law predicts larger slip-weakening distances for larger stress drops, also not seen in Fig. 2. What you can say is that Fig. 2 shows a quasi-exponential decay to the new friction level, which is a hallmark of the Slip law (also the Aging law, but only for very small velocity increases). The fact is that the range of stress drops in Fig. 2 is small enough that one could reasonably not expect to see either an increase in weakening rate (if the Slip law were appropriate) or a longer effective slip-weakening distance (if the Aging law were appropriate), but the exponential stress decay looks a lot like the Slip law. Line 368 continues “Note that the model presents a qualitatively-similar behaviour using the slip law”, but I imagine this must be for values of a/b not too close to 1. In any case, use the Aging law if you wish, but I’m not sure it’s fair to cite Fig. 2 in support of this.

line 464-65 in Methods: This must be μm , not m.

Supplementary Figure S5: I find it difficult to distinguish between red and magenta in panels a and b. Should it be “red dotted line”, rather than red dashed line?

Allan Rubin

Thank you for the revision in response to our comments. We think most part becomes clear now. We still have the following comments and suggestions to make the manuscript more understandable.

a) Comment #12: We visited Whitcomb et al (1973) and thought it inappropriate to cite these data. Whitcomb et al (1973) analyzed the 1971 San Fernando earthquake (M6.4) and compared with other data from Aggarwal et al (1973) and Semenov (1969) with a conversion of magnitude to characteristic length. The serious issue is that this characteristic length in Whitcomb et al (1973) could not be considered as the nucleation length but the mainshock fault area estimated from aftershock distribution of the San Fernando earthquake. Other lengths were converted from magnitudes using an empirical relation by Whitcomb et al. using published data. Therefore, the citation of the data in Whitcomb et al (1973) should not be used. If the authors still wish to use these data, please detail how Δt_c and l_c are estimated. And these values should be listed in Table S1.

And the followings are what we noticed after reading the revised manuscript.

b) Since many parameters are used and some of the definitions appeared after their usage, I would recommend to tabulate the definitions of the parameters as a supplementary, which would help the audience.

c) Figure 4a. The label "Foreshock size, A_{fs} [m/sec²]" is misleading. It should be "Averaged foreshock acceleration, A_{fs} [m/sec²]" as described at L386.

d) Figure 4b. It seems convenient for the readers to swap the axes. Vertical axis is Δt and horizontal axis is l_c .

e) Figures 4a-b. The quasi-static nucleation phases reported in the referenced laboratory earthquakes (Ohnaka and Shen, 1999; Latour et al., 2013; McLaskey, 2019; Yamashita et al., 2021; Xu et al., 2023) do not appear to have been initiated by foreshocks. While I acknowledge the authors' reply to Comment #11 that the nucleation discussed here is not necessarily caused by foreshocks, the laboratory data shown in Figure 4, which are used to discuss the scaling of nucleation duration, should be more clearly described in the main text.

f) Supplementary Table S1: To improve clarity and facilitate cross-referencing, it would be preferable to separate natural and laboratory earthquakes under distinct column headers and list the authors' names and publication year for laboratory earthquakes.

Having read the revised paper and the authors' response, I'm satisfied that my comments have been addressed. I have the following minor, comments, mostly related to the changes from the first draft.

Fig. 3: I'm uncertain how the velocities are determined in panel (a). The new caption for panel (c) says "After the foreshock, the slip evolves as an exponential function of time (Fig. S2), yielding the initial minimum sliding velocity of the patch at the onset of nucleation." I found this a little hard to understand; the deleted portion "This evolution is fit and its derivative is evaluated just after the foreshock, yielding ..." makes it much clearer, and I would even write "This assumed evolution ...". But this still doesn't explain Fig. 3a. From the slip-vs-time data in Fig. S2 e-f, it's obvious that one can't choose the time of minimum velocity by eye. Fig. S2 panels g-i show the numerical derivative of this slip, but it looks very noisy, as expected when differentiating a real time series (I imagine that a point-by-point differentiation could lead to negative velocities sometimes, consistent with the data gaps in Fig. S2h). But the velocity prior to 0.2×10^{-1} s in Fig. 3a, which clearly includes the local maximum in velocity prior to deceleration, looks very smooth. It looks like the displacement time series must have been smoothed prior to differentiation. If so, this procedure should be outlined. Including the details in the Fig. S2 caption might be enough.

The new question (i) on line 137 (line numbers from the "tracked changes" version): I'm not convinced that this improves the paper, at least as currently written. The question is: "Are all earthquakes preceded by a quasi-static nucleation phase?", and the one-word answer is 'No.'. This is then qualified as "observable quasi-static nucleation is not a prerequisite for dynamic rupture", which I agree with, but the qualifier "observable" isn't part of the question. The answer goes on to describe the regime where a "foreshock" leads directly to dynamic rupture. But in this case the expectation is that the "foreshock" (although no longer really a foreshock) itself had a quasi-static nucleation phase. I think an acceptable fix is to just get rid of the "No," at the start of this sentence.

line 149: "under sufficiently stressed conditions"

lines 148-152: This is a really long sentence.

Fig. 4 caption: "The error for the slip during nucleation is estimated by evaluating the slip within 0.1 msec of v_r reaching 10 m/sec." Meaning ± 0.1 msec?

line 158: "Based on these considerations, we suggest the use of the word foreshock should be restricted to earthquakes which either initiate or tangibly influence the quasi-static nucleation phase of another, larger earthquake." I think this is not a good idea, and that whether a good idea or not, it has approximately zero chance of being adopted by the seismological community. I think it is not a good idea because it requires evaluating a potential foreshock using a model that could easily be incorrect or incomplete. I think its chance of being implemented is near zero because in many cases the data necessary to evaluate the potential foreshock (location relative to the mainshock; focal mechanism) may be unavailable or inadequate.

line 335: I think this explanation is an explanation that doesn't really explain. The real question is if/why the "dynamic range" of potential observable foreshocks on PMMA samples, rock samples, and natural faults differ, and their statistical likelihood, meaning the relative abundance of heterogeneities on different length scales. But nothing the authors say here strikes me as wrong, so they can keep it if they like.

Allan Rubin